# Seizure-Semiology-Suite($S^3$): A Clinically Multimodal Dataset, Benchmark, and Models for Seizure Semiology Understanding

**Lina Zhang** [1] **Tonmoy Monsoor** [1] **Peizheng Li** [2] **Jiarui Cui** [1] **Xinyi Peng** [1] **Chong Han** [1] **Prateik Sinha** [1]
**Siyuan Dai** [3] **Jessica Nichole Pasqua** [1] **Colin M McCrimmon** [1] **Weiting Liu** [4] **Hailey Marie Miranda** [1] **Bing Hu** [5]
**Xiangting Wu** [1] **Tengyou Xu** [1] **Chunhan Li** [6] **Jiaye Tian** [1] **Jiarui Tang** [1] **Detao Ma** [1] **Lingye Kong** [1]
**Junnan Lyu** [1] **Jungang Li** [6] **Yan Zan** [7] **Junhua Huang** [1] **Rajarshi Mazumder** [1] **Vwani Roychowdhury** [1]

## Abstract

While Multimodal Large Language Models (MLLMs) have demonstrated remarkable proficiency in general video understanding, their capacity to interpret involuntary, and spatio-temporally evolving pathologic motor behaviors such as seizure semiology remains largely untested. To address this gap, we introduce Seizure-Semiology-Suite (S³), a clinically grounded dataset and benchmark for fine-grained, structured seizure semiology understanding. The dataset includes 438 seizure videos annotated with over 35,000 dense labels covering 20 ILAE-defined semiological features. Building on this dataset, we propose a seven-task hierarchical benchmark that systematically evaluates MLLMs from low-level visual perception to temporal sequencing, narrative report generation, and seizure diagnosis. To enable clinically meaningful evaluation of generated reports, we further introduce the Report Quality Index for Seizure Semiology (Seizure-RQI). Extensive baselines across 11 open-weight MLLMs reveal systematic weaknesses in laterality reasoning, temporal localization, symptom sequencing, and clinically faithful reporting. We show that seizure-specific fine-tuning substantially improves performance across tasks, and that a two-stage neuro-symbolic framework achieves an F1 score of 0.96 on epileptic versus non-epileptic seizure classification. Seizure-Semiology-Suite establishes a rigorous benchmark for evaluating multimodal models in safety-critical medical video understanding and guides the development of clinically reliable, domain-adaptive multimodal intelligence. Our code is publicly available at SeizureSemiologySuite.

## 1. Introduction

Seizure semiology encapsulates the observable motor, behavioral, and autonomic manifestations of a seizure, and accurately recognizing these signs is essential for epilepsy diagnosis, seizure-type classification, and the localization of the seizure onset zone (McGonigal, 2021; Tufenkjian & Lüders, 2012). However, current semiology assessments rely heavily on manual video review by trained experts. This process is subjective and labor-intensive, making it difficult to scale in resource-limited clinical settings (Reuber & Elger, 2003; Chen & LaFrance Jr, 2016). While automated approaches exist, they have traditionally been limited to rigid discriminative tasks (Shoeb et al., 2004). Such categorical frameworks exhibit limitations in modeling the complex reality of epileptic events, where symptoms are often subtle, concurrent, and characterized by critical spatial laterality and temporal evolution (Loddenkemper & Kotagal, 2005; Blume et al., 2001). Consequently, existing systems lack the descriptive interpretability required for clinical decision-making.

Recent advances in MLLMs enable seizure understanding to move beyond simplistic discriminative tasks, embracing a more generative framework (Bai et al., 2025; Wang et al., 2025; Yang et al., 2025). This shift allows models to describe complex, evolving, and coexisting seizure signs in natural language. However, the current community suffers from a lack of large-scale datasets with intensive, descriptive expert annotations to realize the potential of MLLMs in this domain. Furthermore, there is no comprehensive task suite or evaluation protocol that quantitatively measures MLLMs' capabilities across clinically meaningful dimensions, such as laterality and temporal evolution.

[1]University of California, Los Angeles [2]Mercedes-Benz AG [3]University of Pittsburgh [4]Fudan University [5]University of California, Riverside [6]Hong Kong University of Science and Technology (Guangzhou) [7]Maharishi International University. Correspondence to: Rajarshi Mazumder <rmazumder@mednet.ucla.edu>, Vwani Roychowdhury <vwani@g.ucla.edu>.

*Proceedings of the $43^{rd}$ International Conference on Machine Learning*, Seoul, South Korea. PMLR 306, 2026. Copyright 2026 by the author(s).

To address these challenges, we introduce Seizure-Semiology-Suite (S³), a comprehensive dataset and benchmarking framework for fine-grained, structured seizure semiology understanding. Our main contributions:

- We construct the first large-scale, expert-supervised seizure video dataset with dense feature-level labels to address the data scarcity hindering MLLMs application in seizure.
- We build a seven-task hierarchical benchmark with clinically grounded evaluation metrics, including the Report Quality Index for Seizure Semiology (Seizure-RQI), to evaluate MLLMs' performance from perception to clinical reasoning.
- We conduct extensive experiments on 11 open-weight MLLMs to establish current baselines, identifying critical failure modes in temporal sequencing, spatial laterality reasoning and faithful report generation.
- We implement seizure-specialized finetuning and a two-stage classification framework, leading to significant performance improvements in seizure semiology perception and seizure classification.

## 2. Related work

**Medical Video Understanding.** Medical video understanding has primarily targeted structured, procedure-centric settings. MedVidQA (Gupta et al., 2023) advances instructional medical video classification and QA, while SV-RCNet (Jin et al., 2018) models workflow recognition in robotic surgery via phase identification. A shared limitation is the assumption of intentional, goal-directed actions with prescriptive temporal structure (e.g., surgery, first aid). In contrast, datasets for involuntary pathological motion remain scarce.

**General Video Motion Understanding.** Large-scale video-language benchmarks, including ActivityNet-QA (Yu et al., 2019) and MSRVTT-QA (Xu et al., 2016), have driven progress in general video understanding, while recent efforts such as MotionBench (Hong et al., 2025) emphasize fine-grained motion perception. However, these datasets predominantly feature everyday, goal-oriented activities and lack the clinical specificity required to assess diagnostically meaningful attributes. In particular, they do not capture key challenges central to seizure analysis, such as spatial laterality, concurrent symptoms, and precise temporal progression, which are critical for clinical interpretation.

**Seizure Semiology Analysis.** Existing computational approaches to seizure analysis have largely adopted narrow, task-specific pipelines targeting isolated symptoms or seizure types. Prior studies have explored 3D CNNs for tonic–clonic detection (Boyne et al., 2025), accelerometry-based methods (Poh et al., 2012), optical flow segmentation

(Kalitzin et al., 2012), and CNN-based seizure-type classifiers (Mehta et al., 2023). Although effective within their respective scopes, these systems typically produce coarse or weakly interpretable outputs and do not scale to comprehensive, feature-level semiology description. In contrast, our work targets structured, generative modeling of seizure semiology across a broad spectrum of clinically relevant features, aligning more closely with expert clinical reasoning.

**Clinical AI Evaluation.** High-stakes clinical evaluation requires metrics beyond surface-form overlap. N-gram metrics (e.g., BLEU, ROUGE) may correlate weakly with clinical factuality, motivating clinically grounded assessment for medical text generation (Ostmeier et al., 2024). Graph and structure aware resources such as RadGraph (Jain et al., 2021) evaluate entities and relations rather than token overlap. Nonetheless, existing schemes can undercapture discourse level coherence, narrative structure, and temporal logic, properties that are central to seizure reporting (Croxford et al., 2025).

## 3. Seizure-Semiology-Suite

### 3.1. Seizure-Semiology-Dataset Construction

**Dataset and Ethical Considerations.** Seizure-Semiology-Dataset is constructed from continuous seizure video recordings from patients who underwent monitoring at the tertiary Epilepsy Monitoring Unit (EMU) of Ronald Reagan UCLA Medical Center between 2019 and 2023. From patients' 24/7 recordings, expert reviewers first identified candidate seizure events. They then screened clips to ensure clear motor activity and unobstructed full-body views. The final dataset comprises 438 videos from 116 adult patients (ages 18–64) with resolutions of 1920×1080 (81.28%), 1280×720 (18.26%), and 640×480 (0.46%), and a diverse duration distribution (see Figure 1(a)). Each video is paired with its corresponding clinical semiology report and diagnostic record. All data collection was approved by the IRB-23-0054, and all recordings were fully de-identified to protect patient privacy (Section A).

**Annotation procedure and quality control.** Each seizure video was manually annotated by epileptologists and trained annotators for 20 ILAE-defined semiological features (e.g., automatisms, tonic, clonic; see Tables 6, 7, 8). For each feature, annotators recorded presence/absence, temporal boundaries (start/end times), and explanatory justifications, yielding over 35,000 semiology labels in total.

We employed a five-stage annotation workflow to ensure clinical correctness. (i) In an initial onboarding phase, epileptologists introduced the 20 ILAE-defined semiological features to the annotation team using representative video examples to calibrate visual understanding. (ii) Epileptologists selected 151 videos (∼ 34%) covering a full spec-

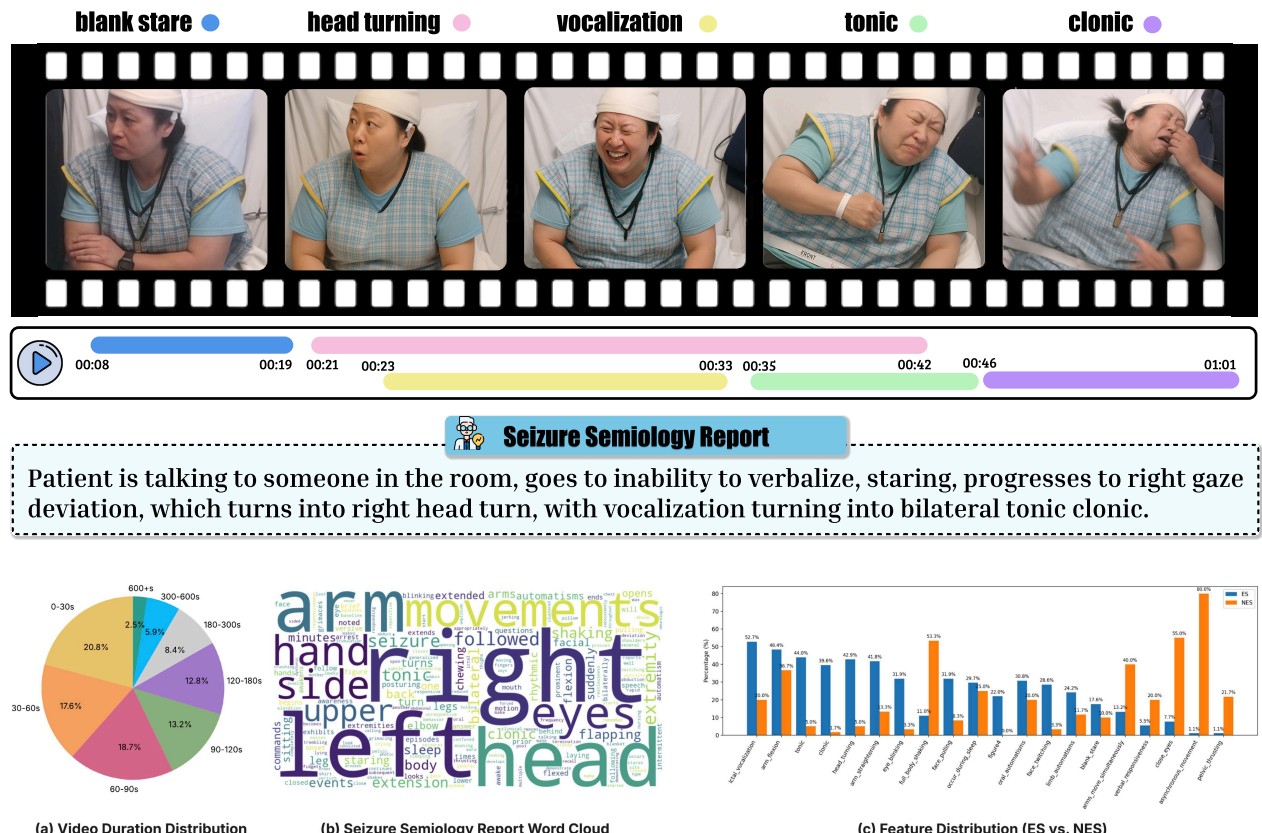

Figure 1. **Seizure Semiology Dataset Information.** (The patient depicted in this figure is AI-generated.)

trum of semiological features. They first trained the annotators using 76 videos through live annotations, detailed explanations, and real-time question and answer. To quantify reliability, the annotators independently labeled the other 75-video subset. As summarized in Table 31, comparison with the clinician annotations yielded an average Kappa score of 0.8395, demonstrating that the trained annotators can faithfully reproduce expert labels at the per-feature level. (iii) Trained annotators independently labeled the remaining 287 videos, with challenging cases presented to a panel of three experts in review meetings for consensus; any disagreements were resolved by a majority vote. (iv) Annotator outputs were iteratively verified using an adaptive sampling strategy. Discrepancies were continuously refined until the feature-level distributions from the annotator subset achieved statistical convergence with the expert-labeled subset. This convergence was confirmed across both seizure types, Epileptic Seizures (ES, pearson similarity score of 0.893) and Non-Epileptic Seizures (NES, pearson similarity score of 0.782) in Table 5. (v) Additional quality control measures were adopted, including using LLMs to correct grammatical errors and semantic inconsistencies in justifica-

tion texts, and implementing rule-based checks to validate temporal labels.

We partition the dataset 4:1 into training and test sets, splitting at the patient level with the ES:NES ratio approximately preserved. The test set comprises 82 videos from 24 patients, and the training set is divided into four folds for cross-validation. We release the dataset annotation part via GitHub and provide access to seizure videos under Data Use Agreement (see Section L).

### 3.2. Seizure-Semiology-Bench

We designed a seven-task hierarchy to systematically quantify MLLMs' capabilities, progressing from basic perception to integrated clinical reasoning (Figure 2).

#### 3.2.1. TASK 1 - SEIZURE SEMIOLOGY RECOGNITION

Semiological features provide critical diagnostic cues for seizure classification and for localizing the epileptogenic zone (EZ). Signs such as tonic posturing, clonic jerks, versive head/eye deviation, and figure-4 arm positioning are strong ES markers with prognostic implications (Lod-

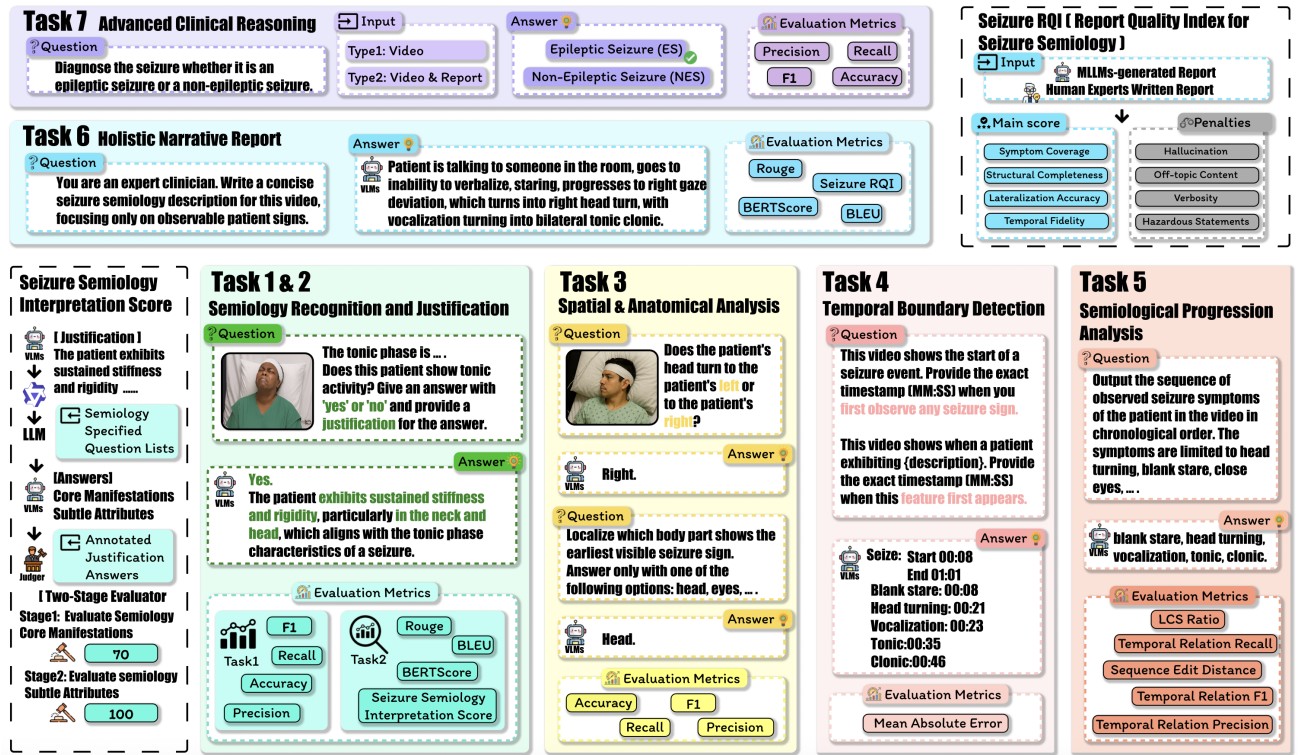

*Figure 2.* **Seizure Semiology Benchmark.** (The patients in this figure are AI-generated.)

denkemper & Kotagal, 2005), including risk of injury and sudden unexpected death in epilepsy (SUDEP) (Harden et al., 2017). In contrast, features like pelvic thrusting and asynchronous movements are more indicative of NES (Perez & LaFrance, 2016). This task evaluates whether an MLLM can detect the presence or absence of 20 core semiological features in a seizure video. Each feature is queried using a binary yes/no prompt, written by epileptologists based on ILAE definitions and clinical expertise, then translated into simpler language for use by general-purpose MLLMs. Predictions are evaluated using Precision, Recall, F1-Score, and Accuracy. The complete set of prompts is listed in Section B.

### 3.2.2. TASK 2 - FEATURE JUSTIFICATION

Beyond recognition, this task requires MLLMs to generate free-text justifications grounded in visual evidence to support their Task 1 predictions. Evaluation combines standard NLP metrics (BLEU, ROUGE-L, BERTScore) with a clinically grounded Seizure Semiology Interpretation Score, which measures semantic alignment between model explanations and expert annotations using a structured rubric. For each of the 20 symptoms, epileptologists define symptom-specific queries covering both core manifestations and subtle attributes. An external LLM (e.g., Qwen3-Plus (Yang et al., 2025)) extracts structured answers from both model justifications and ground truth reports, which are then scored

via a weighted rubric to compute semantic alignment (see Sections B, C).

### 3.2.3. TASK 3 - SPATIAL & ANATOMICAL ANALYSIS

Distinct spatial features, such as head/eye deviation and asymmetric tonic limb posturing, serve as established lateralizing signs in epilepsy diagnosis (Kernan et al., 1993; Kotagal et al., 2000). Furthermore, the anatomical location of the initial motor manifestation often correlates strongly with the seizure onset zone (SOZ), particularly in focal epilepsy (Chowdhury et al., 2021; Bonini et al., 2012). To gauge the model's capacity to discern laterality (left vs. right), we employed precise prompt tuning to mitigate potential viewpoint ambiguity. MLLMs are prompted with forced-choice questions that explicitly anchor directionality to the patient's perspective rather than the camera view, such as "Does the patient's head turn to the patient's left or to the patient's right?" Performance is evaluated using Precision, Recall, F1-Score, and Accuracy.

### 3.2.4. TASK 4 - TEMPORAL BOUNDARY DETECTION

Precise timing of ictal motor signs is clinically important for localizing SOZ and understanding propagation dynamics (Blair, 2012; Jaafar et al., 2024; Blume et al., 2001). Models are required to identify the onset times of individual semiological features, as well as the start and end times of the

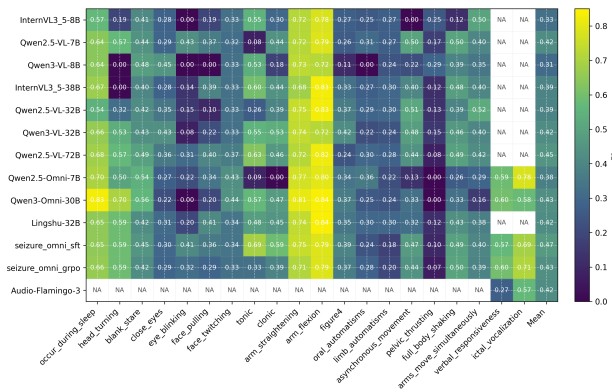

*Figure 3.* **Task 1 F1-scores across 20 semiological features.**

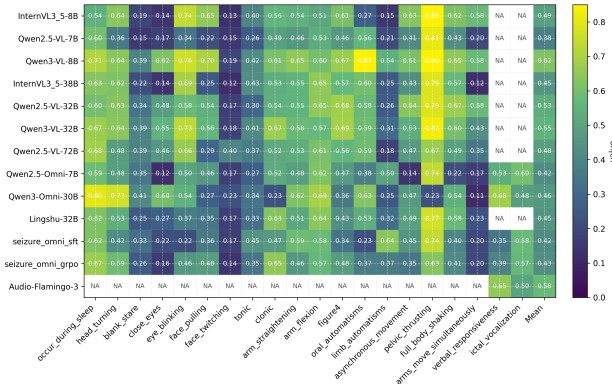

*Figure 4.* **Task 2 Semiology Interpretation Scores.**

seizure event. Prompts request exact timestamps (MM:SS) for symptom appearance. Temporal accuracy is evaluated using Mean Absolute Error (MAE) in seconds.

### 3.2.5. TASK 5 - SEMIOLOGICAL SEQUENCE ANALYSIS

The ordered unfolding of ictal signs, or the "ictal march," reflects the spatiotemporal spread of epileptic activity across the brain and can help infer both symptomatogenic and epileptogenic zones (Blume et al., 2001; Tufenkjian & Lüders, 2012). Temporal progression also distinguishes seizure types (e.g., focal vs. generalized) and supports functional mapping of cortical regions (Tufenkjian & Lüders, 2012). To assess the model's ability to capture the temporal evolution of seizure symptoms, we prompt MLLMs to list symptoms in chronological order from a constrained feature set. Specifically, concurrent symptoms are sorted by their onset time. We adopt three complementary sequence metrics: (i) Sequence Edit Distance (Levenshtein) penalizes insertions, deletions, and out-of-order predictions; (ii) Temporal-relation Precision/Recall/F1-Score evaluates pairwise ordering constraints (e.g., "tonic precedes clonic"); (iii) Longest Common Subsequence (LCS) ratio, defined as the LCS length normalized by the model's output length, measures how much of the correct progression is preserved. Together, these metrics assess both structural correctness and temporal coherence in symptom sequencing.

### 3.2.6. TASK 6 - NARRATIVE REPORT GENERATION

Seizure semiology reports summarize observable ictal manifestations to support diagnosis and localization (McGonigal, 2021). To evaluate whether MLLMs can generate clinically faithful reports from video, we develop clinician-guided prompts and evaluation metrics for narrative report generation (Table 9). Besides standard NLP metrics, we introduce the **Report Quality Index for Seizure Semiology (Seizure RQI)**, a clinically grounded metric designed to evaluate narrative report fidelity. Based on guidance from epilepsy spe-

cialists regarding clinical priorities, Seizure RQI comprises four weighted components: (i) *Structural Completeness S* (15%): Assesses whether the report contains expected 3 sections: onset, propagation, and postictal state. (ii) *Symptom Coverage C* (35%): Ratio of correctly extracted features from the generated report over all extracted features from the ground-truth report. (iii) *Key Localizing Features L* (25%): Extracts specific laterality values (left, right, or none) for four features from both generated and ground-truth reports. L is the proportion of matched features. (iv) *Temporal Fidelity T* (25%): Temporal-relation F1-Score computed between the chronologically ordered feature list extracted from the MLLM reports and the ground-truth feature list. Additional penalties are applied for hallucinated features $P_{\text{hall}}$, off-topic content $P_{\text{off}}$ (e.g., nursing interventions), overly verbose outputs $P_{\text{len}}$, or hazardous clinical statements $P_{\text{haz}}$. The detailed scoring rubric is mentioned in Section D. We also validate whether the Seizure RQI offers superior alignment with human expert judgment compared to NLP metrics.

$$
\begin{aligned}
\text{RQI} = &(0.15S + 0.35C + 0.25L + 0.25T) \\
&\times P_{\text{hall}} \times P_{\text{off}} \times P_{\text{len}} \times P_{\text{haz}}
\end{aligned}
\tag{1}
$$

### 3.2.7. TASK 7 - CLINICAL DIAGNOSIS AND REASONING

Semiology-based epilepsy diagnosis encompasses distinguishing ES from NES, classifying seizure types, and localizing the onset zone. Accurate ES vs NES differentiation represents a fundamental clinical challenge (Reuber & Elger, 2003; Chen & LaFrance Jr, 2016), as misdiagnosing NES as ES leads to inappropriate antiepileptic drug therapy, unnecessary surgical interventions, delayed psychiatric treatment, and substantial healthcare costs (Reuber & Elger, 2003; Chen & LaFrance Jr, 2016). We evaluate two MLLMs' diagnostic approaches: (i) Direct MLLMs Diagnosis, where the model observes the video and directly outputs a diagnosis; (ii) Report-Augmented Diagnosis, where the MLLMs have access to both the video and a clinical report to support contextual reasoning. Performance is evaluated

using Accuracy, Precision, Recall, and F1-Score.

## 3.3. MLLMs Baselines

### 3.3.1. BASELINE MODELS

We benchmark 11 state-of-the-art open-weight models spanning a range of parameter scales, modality coverage, and domain specialization: (i) **General-Purpose VLMs:** We evaluate models across three scales—*small* (InternVL3.5-8B (Wang et al., 2025), Qwen2.5-VL-7B-Instruct (Bai et al., 2025), Qwen3-VL-8B-Instruct (Yang et al., 2025)), *medium* (InternVL3.5-38B (Wang et al., 2025), Qwen2.5-VL-32B-Instruct (Bai et al., 2025), Qwen3-VL-32B-Instruct (Yang et al., 2025)), and *large* (Qwen2.5-VL-72B-Instruct (Xu et al., 2025a)). (ii) **Audio and Omni-Modality Models:** To capture auditory semiological features (e.g., ictal vocalizations, verbal responsiveness), we include the leading audio-language model Audio-Flamingo-3 (Ghosh et al., 2026), as well as omni-modality models Qwen2.5-Omni-7B (Xu et al., 2025a) and Qwen3-Omni-30B-A3B-Instruct (Xu et al., 2025b). (iii) **Medical VLMs:** We include Lingshu-32B (Xu et al., 2025c) to evaluate the added benefit of domain-specific medical pretraining for seizure semiology understanding.

### 3.3.2. IMPLEMENTATION DETAILS

To balance MLLMs context length limits with with the visual clarity and seizure high-frequency dynamics (e.g., rapid blinking, twitching), we use task-specific frame sampling protocols: (i) Uniform Sliding Window (Tasks 1, 2, 5, 6): Videos are segmented into 30s windows (60s for Qwen3 models) with 5s overlap at 2 FPS. Results are aggregated via logical union (Task 1), LLM summarization (Tasks 2, 6), or temporal alignment (Task 5). (ii) Event-Centric Clipping (Tasks 3, 4): We extract segments containing lateralizing features (Task 3) or 60s clips centered on symptom onset (Task 4) . Processing occurs at 1 FPS (2 FPS for Qwen3 models). (iii) Full Video Sampling (Task 7): To preserve global context for diagnosis, we apply uniform sparse sampling across the entire video, utilizing 60 frames (120 for Qwen3 models). (More details are provided in Section H.)

## 3.4. Improvement Strategies

### 3.4.1. SEIZURE-SPECIALIZED FINETUNING

Seizure semiology presents distinct, often subtle multimodal patterns that are sparsely represented in general-purpose pretraining corpora. To address this domain gap, we finetune models specifically for seizure understanding. We select Qwen2.5-Omni-7B as the base model due to its inherent capability to jointly process both visual and auditory modalities.

**Training Protocols.** We investigate two training paradigms: (i) Supervised Fine-Tuning (SFT) (Ouyang et al., 2022; Zhao et al., 2023), where the model is trained using video–prompt–answer triplets; and (ii) Group Relative Policy Optimization (GRPO) (Shao et al., 2024), a reinforcement learning method where the model is optimized using group-normalized rewards from multiple candidate samples. For our policy optimization, we designed task-specific reward functions: we used an accuracy-based reward for Tasks 1, 3, and 7; a composite reward derived from BLEU and ROUGE scores for Tasks 2 and 6; a temporal proximity reward function for Task 4; and the Longest Common Subsequence (LCS) ratio for Task 5. Complete details of hyperparameter choices and optimization settings are provided in Section E.

### 3.4.2. TWO-STAGE SEIZURE CLASSIFICATION

Directly diagnosing seizures from raw video (end-to-end MLLM inference) might be affected by their tendency to hallucinate clinical reasoning over long temporal contexts. To address this, we introduce a two-stage seizure classification strategy that decouples visual perception from diagnostic reasoning. Structured feature extraction (perception): we utilize the MLLMs as a feature extractor. For a given video, the model performs Task 1 to detect the presence or absence of the 20 core ILAE-defined semiological features. This process converts the high-dimensional, unstructured video data into a structured, interpretable binary feature vector $v \in \{0, 1\}^{20}$. In the second stage, we feed this feature vector into a Random Forest classifier. This neuro-symbolic approach leverages the MLLM's strength in visual recognition while offloading the logical classification to a reliable statistical model, thereby improving both accuracy and clinical trust.

## 4. Results and Discussion

The following analysis is based on the test set. Metrics over all 438 videos are provided in Section F, and additional test-set metrics in Section G.

## 4.1. Semiology feature understanding

As shown in Figures 3, 8, Task 1 results reveal:(i) Feature Visibility: Models excel at prominent, sustained features (e.g., tonic posturing, F1 0.6–0.8) but struggle with subtle, fast-evolving movements (e.g., oral automatisms, F1 < 0.4). This disparity stems primarily from the scarcity of fine-grained events in pre-training data.(ii) Scaling Limits: Scaling alone offers limited benefit: Qwen variants from 7B to 72B yield similar mean F1 scores (∼0.42–0.45), suggesting that size alone cannot overcome architectural bottlenecks.(iii) Multimodality fusion Advantage: Qwen3-Omni-30B outperforms both the vision-only Qwen2.5-VL-

32B and the audio-only Audio-Flamingo-3, particularly on sound-associated features. (iv) Architecture: The QwenVL family consistently surpasses InternVL across all scales, indicating superior visual-language grounding.

For Task 2, MLLMs perform best on questions tied to sustained, unambiguous motor behaviors such as "Which arm shows flexion?" or "Is the flexion sustained?", with top performing models reaching semiology interpretation scores around 0.7 (Figure 4). In contrast, questions requiring temporal reasoning (e.g., "Are the twitches brief and repetitive?"), or subtle motion detection (e.g., "Are there small muscle twitches observed on the patient's face?") yield significantly lower performance ($< 0.2$). Across model families, Qwen-VL consistently surpasses InternVL (16/18 features), further highlighting its advantage in grasping nuanced semiological details. To verify LLM sensitivity during evaluation, we conducted additional experiments incorporating two other LLMs: Kimi-K2.5 (Kimi Team, 2026) and GPT-5.4 (OpenAI, 2026) in Table 25. Across all models, Task 2 scores demonstrated minimal variance, ranging narrowly from 0.61 to 0.62.

## 4.2. Spatial and temporal reasoning

Models perform consistently poorly on spatial queries in Task 3, with mean F1 typically $< 0.2$ (Table 1), indicating weak spatial reasoning. Notably, performance remains low even with explicit prompts separating "patient's left" from "camera's left" (Sec. 3.2.3), implying prompt tuning alone is insufficient. Larger models (Qwen2.5-VL-72B, InternVL3.5-38B) are slightly more consistent but still inaccurate. Similar effects have been reported elsewhere (Hoehing et al., 2023; Kamath et al., 2023) which found that the evaluated VLMs perform poorly on spatial reasoning due to insufficient spatial relationship data in pretraining corpora like LAION-2B. We show they persist in clinical video, where lateralized motor signs are diagnostically critical yet remain unresolved for current MLLMs.

Temporal boundary detection is also variable and generally not clinically usable. Qwen2.5-VL-32B is best (MAE: 8.19 s onset, 12.72 s offset; Table 1), but still far from the precision needed for ictal localization (Shoeb et al., 2004). Lingshu-32B performs worse (MAE $> 50$ s), suggesting narrow or biased clinical finetuning can degrade temporal reasoning. Two factors likely drive these errors: (i) dense, concurrent seizure signs make timestamp assignment difficult; and (ii) unlike clinicians who use retrospective reasoning to pinpoint onset, current VLMs rely on a single feed-forward pass. This lack of iterative refinement limits precision, motivating agentic mechanisms for finer resolution.

Task 5 (temporal sequencing) further exposes ordering failures. Although Qwen3-Omni-30B attains the lowest Edit

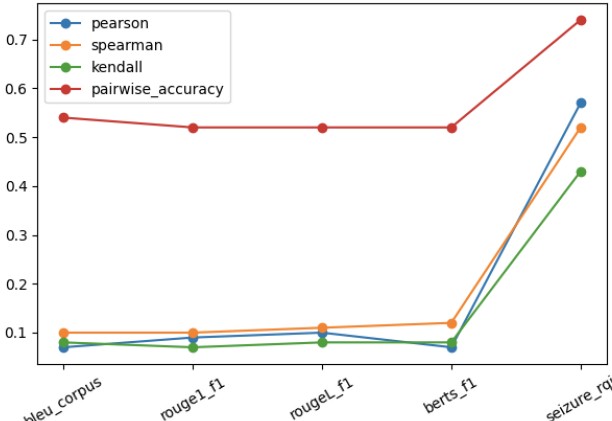

*Figure 5.* **Clinical Alignment of Evaluation Metrics.**

Distance (5.15) and highest LCS ratio (0.43), its Temporal F1 is low (0.18 vs. InternVL3.5-8B's 0.57), indicating poor pairwise order preservation (Table 1). This reflects cascading errors: misrecognition in Task 1 corrupts the "what," and mislocalization in Task 4 corrupts the "when," so sequence generation is conditioned on noisy content and timing. Joint optimization across recognition, localization, and sequencing is therefore likely necessary for clinically faithful ictal progression generation.

## 4.3. Holistic reporting and diagnosis

Lingshu-32B achieved the highest RQI (39.80), validating the benefit of medical pretraining; however, the comparable performance of Qwen3-VL (39.13) demonstrates that strong general baselines are equally competitive (Table 2). As illustrated in Figure 5, conventional metrics (BLEU, ROUGE-L, and BERTScore) exhibit negligible correlation with expert judgments (Pearson $r \leq 0.10$). In stark contrast, Seizure RQI demonstrates robust alignment, achieving a Pearson correlation of 0.57 and a pairwise accuracy of 0.74, significantly outperforming generic baselines (accuracy $\approx 0.54$) and highlighting its effectiveness. Similar to Task 2, we also evaluated LLM sensitivity, observing only small differences in Task 6 RQI scores (38.70–42.26), as shown in Table 25. (Several VLM generated report examples listed in Table 34.)

Report-augmented inputs yield an average F1 improvement of 21.12%, highlighting the value of accurate semiology understanding (Table 2). However, Lingshu-32B excelled (F1 0.84) in video-only setting but its performance dropped sharply with report augmentation (F1 0.60), suggesting limited language-reasoning capacity despite strong medical related visual priors. Qwen3-Omni-30B consistently surpasses vision-only baselines by effectively integrating audio and visual inputs, demonstrating the benefits of multimodal fusion again.

| Models | Task3 | | | | | | | | | | | | Task4 | | Task5 | | |
| --- | --- | --- | --- | --- | --- | --- | --- | --- | --- | --- | --- | --- | --- | --- | --- | --- | --- |
| | Head turning | | | | Arm movement | | | | Onset body part | | | | Start | End | | | |
| | P | R | F1 | A | P | R | F1 | A | P | R | F1 | A | MAE | MAE | EditDist | TempF1 | LCS |
| **Vision-LLM** | | | | | | | | | | | | | | | | | |
| InternVL3_5-8B | 0.22 | **1.00** | 0.36 | 0.22 | 0.88 | 0.88 | 0.88 | 0.78 | 0.09 | 0.12 | 0.08 | 0.14 | 19.14 | 24.33 | 874.66 | **0.57** | 0.02 |
| Qwen2.5-VL-7B | 0.23 | **1.00** | 0.37 | 0.25 | 0.60 | 0.12 | 0.21 | 0.15 | 0.19 | **0.28** | 0.21 | 0.28 | 19.57 | 14.69 | 34.72 | 0.51 | 0.12 |
| Qwen3-VL-8B | 0.14 | 0.14 | 0.14 | 0.62 | 0.50 | 0.04 | 0.08 | 0.11 | 0.08 | 0.09 | 0.06 | 0.16 | 16.17 | 29.06 | 18.86 | 0.45 | 0.19 |
| InternVL3_5-38B | 0.22 | **1.00** | 0.36 | 0.22 | 0.88 | **0.92** | **0.90** | **0.81** | 0.02 | 0.08 | 0.02 | 0.07 | 18.76 | 19.14 | 12.06 | 0.43 | 0.22 |
| Qwen2.5-VL-32B | **0.25** | 0.43 | 0.32 | 0.59 | 0.78 | 0.29 | 0.42 | 0.30 | **0.34** | **0.28** | **0.25** | 0.37 | **8.19** | **12.72** | 445.09 | 0.48 | 0.07 |
| Qwen3-VL-32B | 0.24 | 0.86 | **0.38** | 0.38 | 0.87 | 0.54 | 0.67 | 0.52 | 0.18 | 0.26 | 0.15 | 0.20 | 12.95 | 17.11 | 14.96 | 0.44 | 0.19 |
| Qwen2.5-VL-72B | 0.17 | 0.14 | 0.15 | 0.66 | **1.00** | 0.71 | 0.83 | 0.74 | 0.23 | 0.21 | 0.18 | 0.36 | 10.27 | 17.93 | 14.47 | 0.42 | 0.22 |
| **Omni-LLM** | | | | | | | | | | | | | | | | | |
| Qwen2.5-Omni-7B | 0.15 | 0.29 | 0.20 | 0.50 | 0.50 | 0.08 | 0.14 | 0.11 | 0.10 | 0.14 | 0.05 | 0.08 | 25.93 | 19.43 | 36.33 | 0.45 | 0.12 |
| Qwen3-Omni-30B | 0.00 | 0.00 | 0.00 | 0.75 | 0.50 | 0.08 | 0.14 | 0.11 | 0.15 | 0.18 | 0.14 | 0.22 | 36.21 | 45.37 | **5.15** | 0.18 | **0.43** |
| **Medical-VLM** | | | | | | | | | | | | | | | | | |
| Lingshu-32B | 0.22 | **1.00** | 0.36 | 0.22 | 0.94 | 0.67 | 0.78 | 0.67 | 0.15 | 0.19 | 0.17 | 0.40 | 50.48 | 64.47 | 29.92 | 0.46 | 0.15 |
| **SeizureMLLMs** | | | | | | | | | | | | | | | | | |
| seizure_omni_sft-7B | 0.00 | 0.00 | 0.00 | **0.78** | 0.00 | 0.00 | 0.00 | 0.11 | 0.08 | 0.14 | 0.10 | **0.43** | 19.05 | 22.33 | 7.47 | 0.32 | 0.18 |
| seizure_omni_grpo-7B | 0.00 | 0.00 | 0.00 | **0.78** | 0.00 | 0.00 | 0.00 | 0.11 | 0.18 | 0.14 | 0.11 | **0.43** | 15.66 | 19.45 | 12.24 | 0.32 | 0.18 |

*Table 1.* Performance of all MLLMs across Tasks 3–5.

| Models | Task6 | Task7 | | | | | | | | | | | |
| --- | --- | --- | --- | --- | --- | --- | --- | --- | --- | --- | --- | --- | --- |
| | | Precision | | | Recall | | | F1 | | | Accuracy | | |
| | RQI | w/o rpt | w/ rpt | 2-stages | w/o rpt | w/ rpt | 2-stages | w/o rpt | w/ rpt | 2-stages | w/o rpt | w/ rpt | 2-stages |
| **Vision-LLM** | | | | | | | | | | | | | |
| InternVL3_5-8B | 38.43 | 0.66 | 0.72 | 0.70 | 0.77 | 0.91 | 0.98 | 0.71 | 0.80 | 0.81 | 0.57 | 0.70 | 0.69 |
| Qwen2.5-VL-7B | 36.94 | 0.70 | 0.73 | 0.69 | 0.55 | 0.77 | 0.96 | 0.62 | 0.75 | 0.81 | 0.54 | 0.65 | 0.68 |
| Qwen3-VL-8B | 38.20 | 0.62 | 0.78 | 0.68 | 0.59 | 0.82 | 1.00 | 0.61 | 0.80 | 0.81 | 0.48 | 0.72 | 0.68 |
| InternVL3_5-38B | 36.14 | 0.65 | 0.72 | 0.69 | 0.46 | **0.96** | 0.95 | 0.54 | 0.82 | 0.80 | 0.46 | 0.72 | 0.67 |
| Qwen2.5-VL-32B | 37.48 | 0.64 | 0.91 | 0.73 | 0.75 | 0.70 | 0.96 | 0.69 | 0.79 | 0.83 | 0.54 | 0.74 | 0.73 |
| Qwen3-VL-32B | 39.13 | 0.73 | **0.98** | 0.71 | 0.62 | 0.80 | 0.98 | 0.67 | **0.88** | 0.83 | 0.59 | **0.85** | 0.72 |
| Qwen2.5-VL-72B | 37.37 | 0.71 | **0.98** | 0.72 | 0.70 | 0.75 | 0.91 | 0.70 | 0.85 | 0.80 | 0.60 | 0.82 | 0.69 |
| **Omni-LLM** | | | | | | | | | | | | | |
| Qwen2.5-Omni-7B | 35.91 | 0.67 | 0.72 | 0.88 | 0.93 | 0.79 | **1.00** | 0.78 | 0.75 | 0.93 | 0.63 | 0.65 | 0.88 |
| Qwen3-Omni-30B | 37.52 | 0.70 | 0.87 | 0.91 | 0.77 | 0.84 | 0.98 | 0.74 | 0.85 | 0.94 | 0.62 | 0.80 | 0.89 |
| **Medical-VLM** | | | | | | | | | | | | | |
| Lingshu-32B | **39.80** | 0.73 | 0.89 | 0.70 | **1.00** | 0.45 | 0.93 | **0.84** | 0.60 | 0.80 | **0.74** | 0.59 | 0.68 |
| **SeizureMLLMs** | | | | | | | | | | | | | |
| seizure_omni_sft-7B | 31.69 | 0.66 | 0.69 | **0.93** | 0.78 | 0.94 | 0.99 | 0.71 | 0.79 | **0.96** | 0.56 | 0.66 | **0.91** |
| seizure_omni_grpo-7B | 36.44 | **0.79** | 0.72 | 0.91 | 0.74 | 0.86 | 0.98 | 0.77 | 0.78 | 0.94 | 0.66 | 0.66 | 0.90 |

*Table 2.* Performance of all MLLMs across Tasks 6-7.

## 4.4. Impact of Video Sampling on VLM Performance

We compared clinician and MLLMs' performance under identical 2FPS sampled-frame constraints. As shown in Tables 26, 27, 28 clinicians consistently outperformed the best MLLMs across Task1, 3, 7, confirming the model limitations not solely preprocessing-induced information loss. Furthermore, we conducted an ablation study on sampling rate: increasing from 2 FPS to 4 FPS and 10 FPS yields average Task 1 F1 improvements of 0.06 and 0.08, respectively in Table 29. At last, we conducted an experiment for Task 4 on full videos by uniformly sampling 60 frames. As shown in Table 30, full-video evaluation degrades temporal localization performance, with average MAE increasing by 4.91s and the clinically critical feature clonic showing an MAE increase of 11.39s, which demonstrates that temporal grounding is a fundamental limitation of current MLLMs (Section J).

*Table 3.* Per-task comparison of the Qwen2.5-Omni baseline and two seizure-specialized fine-tuned variants. Improvement_Ave denotes the average percentage improvement across all tasks excluding Task 3.

| Model | Task 1 F1 | Task 2 F1 | Task 3 F1 | Task 4 MAE |
| --- | --- | --- | --- | --- |
| Qwen2.5-Omni | 0.38 | 0.42 | 0.13 | 25.50 |
| seizure_omni_sft | 0.47 | 0.42 | 0.03 | 23.02 |
| seizure_omni_grpo | 0.43 | 0.43 | 0.04 | 20.02 |

| Model | Task 5 LCS | Task 6 RQI | Task 7 F1 | Improvement_Ave |
| --- | --- | --- | --- | --- |
| Qwen2.5-Omni | 0.12 | 35.91 | 0.82 | – |
| seizure_omni_sft | 0.18 | 31.69 | 0.82 | 0.12 |
| seizure_omni_grpo | 0.18 | 36.44 | 0.83 | 0.15 |

## 4.5. Effect of seizure-specific finetuning

Compared to this baseline, excluding Task 3, Seizure-specific finetuning with SFT and GRPO achieve consistent improvements across all remaining six tasks, with an average relative improvement of 12% (SFT) and 15% (GRPO), as shown in Table 3, demonstrating the effectiveness of domain-adaptive training (Figures 3, 4, Tables 1, 2). In Task 1, our seizure_omni_sft model achieved the highest mean F1 score (0.47), outperforming even larger models like Qwen2.5-VL-72B. This highlights a core limitation of general-purpose MLLMs: their pretraining on everyday common activities leaves them poorly equipped to recognize pathological movements, a gap that targeted finetuning can effectively close. We also observed that improvements in feature recognition (Task 1) were associated with better temporal localization (Task 4), suggesting that accurate symptom detection facilitates temporal grounding. Across all 19 events, 18 show reduced MAE after finetuning. Average MAE decreases from 25.50s (baseline) to 23.02s (SFT) and 20.02s (GRPO), representing 9.7% and 21.5% relative improvement. For Task 5, we use LCS ratio as the primary metric because it accounts for output length, unlike Edit Distance and Temporal F1 which can be inflated by repetitive outputs. LCS ratio improves from 0.12 to 0.18 (50% gain) after finetuning. The Temporal F1 decrease (0.45→0.32) is an artifact of reduced repetition — the baseline generates substantial meaningless repetitions, and once eliminated, fewer pairwise comparisons lower the F1 numerically despite improved sequence quality.

In Task 3, the model exhibited catastrophic forgetting due to limited laterality data samples (See Table 12). Specifically, Task 3 has only 527 training samples ( 1/8 of Tasks 1–2), with head turning lateralization at 98 and arm movement lateralization at 83 samples — insufficient for fine-grained spatial reasoning. During SFT, the model collapses all predictions to a single direction ("left"), yielding F1 = 0.00 (positive label is "right"). The base model achieves F1 = 0.20 on head turning before finetuning, but narrow finetuning on limited laterality data erases this capability. Additionally, GRPO-based tuning exposed a key limitation: reward functions based on BLEU and ROUGE failed to reflect clinical relevance, resulting in repetitive outputs. This underscores the need for domain-specific optimization targets, such as seizure RQI for effective reinforcement learning in medical reporting tasks.

## 4.6. Two-Stage Seizure Classification Performance

As evidenced in Table 2, the proposed two-stage seizure classification consistently outperforms direct end-to-end MLLM classification. Quantitatively, this strategy yields a dramatic improvement over direct MLLM classification (w/o rpt), increasing the average F1 score from 0.70 to 0.86 and yield-

ing an average improvement of +0.16 F1 across models. Notably, even compared to report-augmented MLLM classification (w/ rpt), the two-stage strategy still delivers a clear gain, improving the average F1 from 0.79 to 0.86.

Table 32 shows that this improvement is robust across multiple second-stage classifiers trained on the same structured semiology representation. Random Forest achieves the highest mean F1, with SVM performing comparably, while KNN and Logistic Regression are somewhat lower, suggesting that the structured feature representation is the primary source of predictive gain and that classifier choice plays a secondary role. Consistent with this, Table 33 shows that the Random Forest classifier relies most strongly on clinically meaningful semiological cues such as head turning, tonic activity, rapid eye blinking, and occurrence during sleep, indicating that the second stage is leveraging interpretable seizure descriptors rather than opaque end-to-end correlations.

Most significantly, when coupled with our fine-tuned foundation model, seizure_omni_sft, the two-stage strategy achieves an F1 score of 0.96. To our knowledge, this is the first pure video-based seizure classification framework to achieve this level of performance on a large-scale dataset, demonstrating accuracy comparable to reported diagnostic benchmarks under controlled experimental settings.

## 5. Conclusion

Seizure-Semiology-Suite makes the following contributions: the first large-scale expert-supervised dataset supporting both domain-adaptive model development and future multimodal integration with EEG and MRI for comprehensive pathological analysis; the first clinically grounded hierarchical benchmark establishing quantifiable standards for AI-based seizure interpretation; extensive baseline evaluations exposing fundamental MLLMs limitations: while models recognize salient features, they fail critically at temporal reasoning, spatial localization, and clinical coherence; seizure-specific finetuning and a two-stage neuro-symbolic approach achieving near-clinical diagnostic accuracy, enabling automated assessment to reduce resource-intensive workflows and expand diagnostic access in underserved settings through mobile-captured video screening. Despite its contributions, this work is limited by its reliance on a single-center adult cohort, low-framerate processing constraints.

Future work will pursue: (1) agentic frameworks allowing models to iteratively refine temporal judgments through tool-assisted video analysis; (2) multimodal fusion of vision, audio, and 3D body pose to resolve left-right ambiguities; (3) extended-context architectures capable of tracking symptom progression throughout seizure events, advancing toward reliable clinical decision support systems.

## Impact Statement

This work aims to advance multimodal machine learning methods for structured understanding of seizure semiology from clinical video data. By introducing a carefully annotated dataset, clinically grounded benchmarks, and evaluation metrics aligned with expert reasoning, our goal is to facilitate research toward more interpretable and reliable AI systems that can support epilepsy diagnosis and analysis.

The proposed models and benchmarks are intended strictly as decision-support tools rather than replacements for trained clinicians. Automated interpretation of seizure videos carries inherent risks, including misrecognition of subtle symptoms, incorrect temporal localization, or over-reliance on model outputs in real-world clinical settings. To mitigate these risks, our study emphasizes fine-grained evaluation, explicit failure mode analysis, and clinically motivated metrics that penalize hallucinations, incorrect lateralization, and hazardous statements.

All data used in this work were collected under institutional review board (IRB) approval and fully de-identified to protect patient privacy. Visual examples shown in the paper are AI-generated and do not depict real patients. While our results demonstrate promising performance under controlled experimental conditions, any clinical deployment would require rigorous prospective validation, careful integration into clinical workflows, and oversight by qualified medical professionals.

We believe this work contributes positively by providing a transparent, standardized foundation for evaluating multimodal models in safety-critical medical video understanding, while highlighting the limitations and safeguards necessary for responsible future deployment.

## Acknowledgement

The extensive fine-grained annotation work required to construct the Seizure Semiology dataset was primarily completed by UCLA students. They underwent specialized medical training to ensure meticulous annotation, and all data annotators have been included in the author list. Additionally, we would like to extend our gratitude to Mehmet Efe Lorasdagi, Jayhee Seo, Ido Dukler, Chaya Manjeshwar, Ian Kwon, Medwin Yuming Zhu, and Yuanda Xu for their assistance with additional annotation and scripting tasks.

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

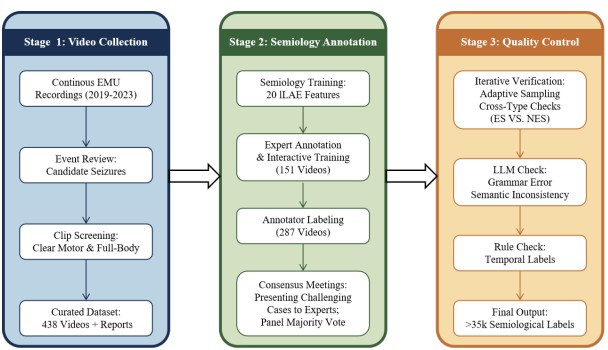

*Figure 6.* **Dataset Construction Pipeline**

*Table 4.* Demographic characteristics of patients with epileptic (ES) and nonepileptic seizures (NES).

| | ES (n=84) | | NES (n=32) | | Total (n=116) | |
|---|---|---|---|---|---|---|
| | n | % | n | % | n | % |
| Video samples | 300 | 68.5 | 138 | 31.5 | 438 | 100.0 |
| Sex: Female | 39 | 46.4 | 26 | 81.3 | 65 | 56.0 |
| Sex: Male | 45 | 53.6 | 6 | 18.8 | 51 | 44.0 |
| White/Caucasian | 46 | 54.8 | 18 | 56.3 | 64 | 55.2 |
| Hispanic/Latino | 17 | 20.2 | 4 | 12.5 | 21 | 18.1 |
| Asian | 7 | 8.3 | 2 | 6.3 | 9 | 7.8 |
| Black/African Am. | 3 | 3.6 | 1 | 3.1 | 4 | 4.8 |
| Am. Indian/AK Nat. | – | – | 3 | 9.4 | 3 | 2.6 |
| Multiracial | 1 | 1.2 | 1 | 3.1 | 2 | 1.7 |
| Declined | 10 | 8.4 | 3 | 9.4 | 13 | 11.2 |
| Age of onset (yrs) | 15.1 ± 12.2 | | 25.9 ± 16.6 | | 18.0 ± 14.3 | |
| Depression | 30 | 35.7 | 14 | 43.8 | 43 | 37.1 |
| Anxiety disorder | 23 | 27.4 | 12 | 37.5 | 34 | 29.3 |
| PTSD | 2 | 2.4 | 6 | 18.8 | 8 | 6.9 |

# A. Dataset and Annotation

The overall dataset construction pipeline is shown in Figure 6.

## A.1. Demographics

The study included 116 patients, comprising 84 individuals with epileptic seizures (ES) and 32 with nonepileptic seizures (NES). A total of 438 seizure videos were analyzed (300 ES, 138 NES). The ES group had a balanced sex distribution, whereas the NES group showed a higher proportion of female patients. Across racial and ethnic categories, White/Caucasian participants were the largest group (55%), followed by Hispanic/Latino (18%), Asian (8%), Black/African American (5%), and smaller proportions of American Indian/Alaska Native and multiracial individuals. NES patients tended to have a later age of onset and a higher prevalence of psychiatric comorbidities such as depression, anxiety, and PTSD compared with the ES group. Please see Table 4 for demographics details.

## A.2. Concordance Between Epileptologists and Trained Annotators labeling distributions

To quantitatively assess how closely trained annotators replicate expert-level semiology labeling, we computed three complementary similarity metrics across epileptologists and annotators: (i) Pearson correlation, (ii) Cosine similarity, and (iii) Earth Mover's Distance (EMD). These metrics capture linear correspondence, angular similarity in high-dimensional space, and distributional distance, respectively, providing a multi-faceted evaluation of agreement between the two groups.

**Pearson Correlation.** Pearson's $r$ measures the linear association between two real-valued distributions. Given expert feature frequencies $\mathbf{x} = (x_1, \ldots, x_d)$ and annotator frequencies $\mathbf{y} = (y_1, \ldots, y_d)$ for $d$ semiology features, Pearson correlation is computed as:

$$r = \frac{\sum_{i=1}^{d}(x_i - \bar{x})(y_i - \bar{y})}{\sqrt{\sum_{i=1}^{d}(x_i - \bar{x})^2}\sqrt{\sum_{i=1}^{d}(y_i - \bar{y})^2}},$$

where $\bar{x}$ and $\bar{y}$ denote the means. This metric captures how well variations in one distribution predict variations in the other.

For ES videos, we obtained a very strong correlation

$$r_{\text{ES}} = 0.893, \qquad p = 1.16 \times 10^{-7},$$

and for NES videos,

$$r_{\text{NES}} = 0.782, \qquad p = 4.70 \times 10^{-5}.$$

When concatenating ES and NES vectors into a unified 40-dimensional distribution, the correlation remained high:

$$r_{\text{combined}} = 0.836, \qquad p = 1.99 \times 10^{-11}.$$

**Cosine Similarity.** Cosine similarity evaluates the angular closeness between the expert and annotator frequency vectors:

$$\cos(\mathbf{x}, \mathbf{y}) = \frac{\mathbf{x} \cdot \mathbf{y}}{\|\mathbf{x}\|_2 \|\mathbf{y}\|_2}.$$

This metric is scale-invariant and frequently used in machine learning and computer vision to compare high-dimensional descriptors. We observed high cosine similarities: (i) $\cos_{\text{ES}} = 0.966$ (ii) $\cos_{\text{NES}} = 0.883$ (iii) $\cos_{\text{combined}} = 0.931$.

**Earth Mover's Distance (EMD).** To measure distributional dissimilarity, we computed the 1-Wasserstein distance between normalized feature frequencies:

$$\text{EMD}(\mathbf{p}, \mathbf{q}) = \inf_{\gamma \in \Gamma(\mathbf{p}, \mathbf{q})} \int |u - v| \, d\gamma(u, v),$$

*Table 5.* Similarity between epileptologists and trained annotators for ES, NES, and combined feature distributions using Pearson correlation, cosine similarity, and Earth Mover's Distance (EMD).

| Comparison | Pearson $r$ | $p$-value | Cosine | EMD |
|---|---|---|---|---|
| ES | 0.893 | $1.16 \times 10^{-7}$ | 0.966 | 0.00864 |
| NES | 0.782 | $4.70 \times 10^{-5}$ | 0.883 | 0.00752 |
| Combined | 0.836 | $1.99 \times 10^{-11}$ | 0.931 | 0.00288 |

where **p** and **q** are probability-normalized versions of expert and annotator distributions, and $\Gamma(\mathbf{p}, \mathbf{q})$ denotes all valid joint couplings. EMD quantifies the "cost" of transforming one distribution into the other and is sensitive to global structure. Results show very small distances: (i) $\text{EMD}_{\text{ES}} = 0.00864$ (ii) $\text{EMD}_{\text{NES}} = 0.00752$ (iii) $\text{EMD}_{\text{combined}} = 0.00288$.

**Computation Details.** All similarity metrics were computed directly from the raw feature frequency vectors derived from the annotated dataset. For each of the $d = 20$ semiology features, we computed the empirical percentage frequency of occurrence for epileptologists and trained annotators separately for ES and NES subsets. Vectors were normalized to probability distributions prior to EMD computation. Concatenated ES+NES distributions (40-dimensional) were evaluated identically.

**Interpretation.** Across all three metrics, trained annotators showed strong to extremely strong agreement with expert epileptologists. The ES distribution demonstrated near-identical shape across groups, and NES distributions, despite being more behaviorally diverse, still exhibited high similarity across all metrics. The very small EMD values indicate that only negligible global mass transfer is required to match annotator distributions to expert distributions. High cosine similarity across the ES, NES, and combined conditions further demonstrate that annotator-derived feature vectors preserve the geometric structure of expert feature profiles.

Overall, these results show that trained annotators reproduce expert-level seizure semiology patterns with exceptional fidelity across both epileptic and nonepileptic events. Their feature distributions are statistically and geometrically aligned with those of epileptologists, validating their use in scalable, high-throughput annotation pipelines for clinical seizure video datasets. Please see Figure 7 and Table 5 for group level annotation and statistical significance details.

## B. MLLM prompts for 7 hierarchichal tasks

### B.1. Task 1 MLLM Prompt and Task 2 Question list

Please see Tables 6, 7, 8 for the list of MLLM prompts for Task 1, Question list for Task 2, clinical relevance of the semiological features and the associated PubMed citations.

### B.2. MLLM Prompts for Tasks 3,4,5,6, and 7

Prompts are in Table 9.

## C. Semiology Interpretation Score

For each seizure semiological feature, the epileptologist provides a chained list of questions (see Tables 6, 7, 8 for the question list) moving from the core manifestation of the symptom to its subtle attributes. The Seizure Semiology Interpretation Score is also calculated according to this primary–secondary order. For each justification generated by the MLLM, we first check whether the MLLM's answer matches the ground truth; if it does not match, we immediately stop and assign a score of 0. Then we compare the core manifestation of the first symptom feature in the question list: if it matches, 70% of the score is awarded; if it does not match, we again directly assign 0. Finally, we compare the secondary attributes of the feature: if all details match, the score is 100%. If there are m secondary details and the MLLM correctly answers n of them, then the detail score is calculated as 30% × (n/m).

## D. Seizure RQI

### D.1. Seizure RQI Calculation

Standard NLP metrics are fundamentally misaligned with the requirements of clinical documentation, as they cannot penalize factually incorrect but linguistically plausible statements. To address this, we propose the Report Quality Index for Seizure Semiology (Seizure RQI), a comprehensive, multi-dimensional index designed to measure the gap between a generated report and a clinical gold standard.

The base score comprises four components—Structural Completeness, Symptom Coverage, Key Localizing Features, and Temporal & Quantitative Fidelity weighted 15%, 35%, 25%, and 25%, respectively. We briefly describe each of the component.

**Structural Completeness (S):** We compare the model-generated report with the ground-truth narrative across three sections: (i) Onset: concrete visible signs at the beginning (e.g., blank stare, lip smacking, versive head turn, unilateral arm flexion/extension). (ii) Propagation: how the seizure symptom evolves/spreads and laterality (e.g., fencer posturing with L flexion/R extension, transition to bilateral tonic–clonic). (iii) Postictal: the immediate state

*Table 6.* Semiological features: importance, citations, and MLLM prompts/questions for Tasks 1 & 2 (Part 1).

| Feature | Importance / Citation | MLLM Prompt / Questions / Context |
|---|---|---|
| **Face twitching** | **Importance:** Focal facial motor seizures lateralize to the contralateral hemisphere. **PubMedCitation:** (Wu & Nordli, 2023) | **MLLM Prompt:** Are there small muscle twitches observed on the patient's face? Answer with 'yes' or 'no'. **Questions:** (1) Are the twitches brief and repetitive? (2) Does the face twitch to the left or right? **Context:** N/A |
| **Face pulling** | **Importance:** Focal facial motor seizures lateralize to the contralateral hemisphere. **PubMed citation:** (Wu & Nordli, 2023) | **MLLM Prompt:** Does the patient exhibit unilateral sustained face-pulling movements? Answer with 'yes' or 'no'. **Questions:** (1) Does the face pull to the patient's left or to the patient's right? (2) Does the face pulling sustain for at least a few seconds? **Context:** N/A |
| **Tonic** | **Importance:** Hallmark of motor seizures; unilateral tonic extension/flexion patterns (e.g. "figure-4") aid lateralization. **PubMed citation:** (Kotagal et al., 2000) | **MLLM Prompt:** The tonic phase is marked by a sudden onset of sustained stiffness or rigidity, usually lasting 5–20 seconds. This stiffness may be generalized, with all limbs held in fixed extension or flexion posture and can include stiffening of the head and axial body. It may also be focal involving a subset of body parts or just one body part at a time. Does this patient show tonic? Give an answer with 'yes' or 'no'. **Questions:** (1) Does the patient exhibit sudden muscle stiffness or rigidity? (2) Which body parts are affected? (3) For each affected body part, specify which side of the patient is involved? **Context:** The patient is not having whole body shaking |
| **Clonic** | **Importance:** Repetitive, rhythmic jerking; contralateral localization when unilateral. **PubMed citation:** (Fotedar et al., 2024; Niaz et al., 1999) | **MLLM Prompt:** Do you observe rhythmic, repeated jerking movements in the patient's body? Answer with 'yes' or 'no'. **Questions:** (1) Which body parts are jerking? (2) Is the jerking unilateral or bilateral? (3) Is the jerking sustained or brief? **Context:** N/A |
| **Occur during sleep** | **Importance:** Nocturnal seizures (esp. frontal lobe epilepsy) often mimic parasomnias; sleep context raises suspicion for epilepsy. **PubMed citation:** (Derry et al., 2009; Krutoshinskaya et al., 2024) | **MLLM Prompt:** Is the patient sleeping at the beginning of the video? Answer with 'yes' or 'no'. **Questions:** N/A **Context:** N/A |
| **Arm flexion** | **Importance:** Part of asymmetric tonic posturing (e.g., "figure-4" sign) — flexion on the side ipsilateral to seizure onset. **PubMed citation:** (Kotagal et al., 2000) | **MLLM Prompt:** Does the patient exhibit sustained arm flexion? Answer with 'yes' or 'no'. **Questions:** (1) Is one arm flexed while the other is extended? (2) Which arm is flexed? **Context:** N/A |

after movements stop (e.g., confusion, unresponsiveness, amnesia, drowsiness).

**Symptom Coverage (C):** The ratio of correctly LLM-extracted features from the MLLM generated report over all LLM-extracted features from the ground-truth report.

**Key Localizing Features (L):** Extracting specific values for: direction of head turning, laterality of the first motor sign, type of asymmetric tonic arm posturing, and Figure-4 arm extension side. Example output: ($left$, $right$ or $none$ if not mentioned) and compare them to the ground-truth references for exact-match evaluation.

**Temporal Fidelity (T):** Given a chronologically ordered list of seizure features, calculate Temporal-relation F1-Score. Combining the 4 components, we get the base score as:

$$\text{Base} = 0.15\,S + 0.35\,C + 0.25\,L + 0.25\,T \qquad (2)$$

Besides the weighted base score, the penalty is composed of four parts: (i) Hallucination penalties: a multiplicative penalty based on features not present. For $n$ features in the MLLM generated report that are absent from the ground truth, multiply the weighted score by $0.95^n$,

$$P_{\text{hall}} = 0.95^{\,n}. \qquad (3)$$

(ii) Off-topic content or non-seizure disturbances (contrast/interference): nursing interventions, patient repositioning, camera obstruction, and device alarms must be excluded. If such content appears in the LLM-generated report, the final score is penalized by 20%,

$$P_{\text{off}} = 0.8. \qquad (4)$$

(iii) Length penalty: for any MLLM-generated report, every 50 words beyond 100 words length (W) reduces the final score by 20%,

$$P_{\text{len}} = 0.8^{\,\max\left(0,\,\lfloor (W-100)/50 \rfloor\right)}. \qquad (5)$$

(iv) Safety gate: if the report contains hazardous recommendations (e.g., incorrect medication dosage), the maximum achievable score is 50 points (i.e., the score is capped at 50).

$$P_{\text{haz}} = \begin{cases} 1, & \text{if no hazardous clinical statements are present,} \\ 0, & \text{otherwise.} \end{cases} \tag{6}$$

Combining the base scores and the weighted penalties, the final RQI is then given by,

$$\begin{aligned} \text{RQI} = (0.15S + 0.35C + 0.25L + 0.25T) \\ \times P_{\text{hall}} \times P_{\text{off}} \times P_{\text{len}} \times P_{\text{haz}} \end{aligned} \tag{7}$$

### D.2. Comparison of Task 6 Report Metrics with Expert Ratings

To compare the effectiveness of different report metrics in Task 6, we randomly selected 20 patients and three epileptologist manually scored 60 reports generated by three MLLMs based on the following criteria: 5 (Excellent): The MLLM-generated report accurately captures all critical semiological features mentioned in the ground truth. It is factually consistent and contains no hallucinations; 4 (Good): The MLLM-generated report captures the majority of the key semiological features but may miss minor details or use less precise clinical terminology; 3 (Moderate): The MLLM-generated report identifies some correct semiological features but misses significant ones or contains minor factual inaccuracies; 2 (Poor): The MLLM-generated report only describes general, non-specific events and fails to identify most of the crucial clinical signs mentioned in the ground truth; 1 (Failure): The MLLM-generated report is completely irrelevant, factually incorrect, or hallucinates information not supported by the ground truth. The details of each metric and the expert ratings are shown in Table 11.

We then computed the Pearson, Spearman, and Kendall correlation coefficients and the pairwise accuracy reflecting ranking consistency between five metrics and the expert scores. The results show that Seizure RQI is the metric most closely aligned with the expert ratings (Table 10).

### E. Finetune Details

**Finetuning dataset preparation.** To prevent data leakage and ensure generalization, we perform a strict patient-level split of the full video corpus, dividing samples into training and testing sets in a 4:1 ratio, grouped by patient ID. Stratified sampling maintains consistent ES to NES ratios across splits. Task-specific training samples are constructed as follows: (i) For Tasks 1–3, videos are trimmed to manually annotated start and end boundaries; (ii) For Task 4, 60-second clips are constructed such that the true semiological feature onset is randomly located within the window to avoid positional bias (e.g., center-frame shortcuts); (iii) For Tasks 5 and 6, both video inputs and corresponding answers are truncated at the 30-second level to preserve temporal alignment; (iv) For Task 7, we apply uniform sparse frame sampling across the full video to retain global context. The finetuning dataset consists of prompt–answer pairs and will be released publicly via Hugging Face to support transparent evaluation and community benchmarking.

SFT command:

```
export SAMPLING_RATE=16000
CUDA_VISIBLE_DEVICES=0,1,2,3,4,5,6,7 \
NPROC_PER_NODE=8 \
VIDEO_MAX_PIXELS=$((784*448)) \
FPS_MAX_FRAMES=60 \
FPS=1 \
MAX_PIXELS=$((784*448*60)) \
swift sft \
    --model "Qwen/Qwen2.5-Omni-7B"\
    --model_kwargs \
    '{"use_audio_in_video": true}' \
    --dataset $TASK_DATASET   \
    --load_from_cache_file true \
    --split_dataset_ratio 0.05 \
    --train_type lora \
    --torch_dtype bfloat16 \
    --attn_impl sdpa \
    --gradient_checkpointing true \
    --num_train_epochs 3 \
    --per_device_train_batch_size 1 \
    --per_device_eval_batch_size 1 \
    --learning_rate 5e-5 \
    --lora_rank 8 \
    --lora_alpha 32 \
    --target_modules all-linear \
    --freeze_vit true \
    --freeze_aligner true \
    --gradient_accumulation_steps 4 \
    --eval_strategy steps \
    --eval_steps 100 \
    --save_strategy steps \
    --save_steps 100 \
    --save_total_limit 3 \
    --logging_steps 20 \
    --max_length 32768 \
    --warmup_ratio 0.05 \
    --dataloader_num_workers 8 \
    --dataset_num_proc 1 \
    --load_best_model_at_end true \
    --metric_for_best_model eval_loss \
    --greater_is_better false \
    --deepspeed zero2 \
    --report_to 'wandb' \
    --run_name "$VERSION_NAME" \
    --enable_channel_loss 'True' \
    --output_dir $MODEL_SAVE_PATH
```

GRPO command:

```
export ENABLE_AUDIO_OUTPUT=False
VIDEO_MAX_PIXELS=$((784*448)) \
FPS_MAX_FRAMES=60 \
```

```
MAX_PIXELS=$((784*448*60)) \
NPROC_PER_NODE=8 \
ENABLE_AUDIO_OUTPUT=False \
CUDA_VISIBLE_DEVICES=0,1,2,3,4,5,6,7 \
FPS=1 \
swift rlhf \
    --rlhf_type grpo \
    --model $BASE_MODEL \
    --model_kwargs \
    '{"use_audio_in_video": true}' \
    --reward_funcs seizure \
    --reward_weights 1.0 \
    --train_type lora \
    --lora_rank 8 \
    --lora_alpha 32 \
    --target_modules all-linear \
    --torch_dtype bfloat16 \
    --dataset $TASK_DATASET  \
    --load_from_cache_file true \
    --external_plugins './plugin.py'\
    --max_steps 6000 \
    --per_device_train_batch_size 2 \
    --per_device_eval_batch_size 2 \
    --learning_rate 1e-6 \
    --gradient_accumulation_steps 1 \
    --eval_steps 300 \
    --save_steps 100 \
    --save_total_limit 10 \
    --logging_steps 5 \
    --max_new_tokens 1024 \
    --max_length 32768 \
    --warmup_ratio 0.05 \
    --dataloader_num_workers 8 \
    --dataset_num_proc 8 \
    --num_generations 6 \
    --temperature 0.8 \
    --top_p 0.95 \
    --top_k 50 \
    --system './rlhf/prompt.txt'\
    --deepspeed zero2 \
    --log_completions true \
    --report_to 'wandb' \
    --run_name "$VERSION_NAME" \
    --enable_channel_loss 'True' \
    --output_dir $MODEL_SAVE_PATH
```

## F. MLLM Metrics on Entire Dataset

The dataset is partitioned at the patient level to prevent data leakage, utilizing stratified sampling based on the ES:NES ratio. In addition to a standard fixed train-test split, our framework also supports 5-fold cross-validation. To facilitate future research, the dataset, along with the inference results of 11 MLLMs across the entire dataset, will be publicly released.

Detailed metrics for each task are provided in the following tables: Table 13 for task 1; Table 14 for task 2; Table 15 for task 3; Table 16 for task 4; Table 17 for task 5; Table 18 for task 6.

## G. MLLM Metrics on Test Dataset

Detailed results for Tasks 1 through 6 are presented in Tables 19, 20,  21, 22, 23,  24, respectively.

## H. MLLMs Baselines Implementation Details

A key challenge in processing seizure videos with current MLLMs is balancing input context limits with the need for high temporal resolution. Seizure semiology often involves rapid, high-frequency changes, such as eye blinking or subtle facial twitching, which require high resolution and could be easily missed by standard sparse sampling strategies. To mitigate this trade-off, we design task-specific sampling protocols.

**Uniform Sliding Window Segmentation (Tasks 1, 2, 5, 6).** For detection and detailed description tasks, we employ a sliding window approach to maintain sufficient temporal granularity (2 FPS). Videos are segmented into 30-second windows (extended to 60 seconds for Qwen3-VL and Qwen3-Omni models due to enhanced long-context capabilities), with a 5-second overlap between adjacent segments to avoid truncating continuous semiological features at segment boundaries. Segment-level inference results are aggregated according to task demands: Task 1 uses logical ANY-of-YES pooling; Tasks 2 and 6 employ an LLM specifically prompted to merge segment-wise descriptions into unified narratives; Task 5 aligns sequential outputs across overlapping windows to preserve temporal order.

Our 2FPS sampling strategy was designed by jointly considering MLLMs context length limits with the visual clarity required for medical symptom detection. In Qwen2.5-VL, total video tokens follow: Total Tokens = Frames/2 × Width/28 × Height/28, with a cap of 24,576 tokens. At 720P this allows  40 frames; at 480P  118 frames. Since seizure symptoms involve subtle facial and limb changes demanding adequate spatial resolution, we set resolution between 480P and 720P with a maximum of 60 frames (120 for Qwen3). Compared to standard benchmarks that use sparse sampling (Video-MME: 16–128 frames, MVBench: 8–16, MotionBench:  0.2 FPS), seizure symptom evolution requires denser temporal coverage, so we adopted 2 FPS with 30-second clips.

**Event-Centric Video Clipping (Tasks 3 & 4).** To facilitate precise localization, we reduce the input space by extracting event-centered video clips. In Task 3 (Lateralization), we specifically extract video segments containing lateralizing features (e.g., head turning, arm movement). In Task 4 (Temporal Localization), each input is clipped to a 60-second window centered around the ground-truth feature onset. All models process clips at 1 FPS (60 frames), except Qwen3-VL and Qwen3-Omni, which operate at 2 FPS (120 frames) to leverage their improved temporal resolution.

**Full Video Uniform Sampling (Task 7).** Since clinical diagnosis relies on global seizure context, we avoid segmentation and instead sample frames uniformly across the entire video. Most models receive 60 evenly spaced frames, while Qwen3-VL and Qwen3-Omni are given 120 frames to leverage their extended visual context capabilities.

All models are run with deterministic generation (temperature = 0) to ensure reproducibility across all evaluations.

## I. LLM-as-Judge Sensitivity in Evaluation

Task 2 and parts of Task 6 report evaluation rely on LLM-based extraction/judging, we tested the evaluator sensitivity as in Table 25.

## J. Impact of Video Sampling on VLM Performance

We compared clinician and MLLM performance under identical sampled-frame constraints (2 FPS). As shown in Table 26, clinicians consistently outperformed the best MLLM across all Task 1 evaluated features (e.g., tonic 0.87 vs. 0.69, figure-4 0.78 vs. 0.42, face twitching 0.83 vs. 0.44), confirming that the performance gap reflects genuine model limitations, not solely preprocessing-induced information loss. Eye blinking yields lower scores even for clinicians (0.52), indicating frame subsampling does remove some information for transient features, but the remaining clinician–MLLM gap still points to fundamental model weaknesses. For Task 3, clinicians achieved F1 = 0.89 vs. 0.38 (MLLM) on head turing direction and 0.54 vs. 0.25 on body-region onset in Table 28; For Task 7, the MLLM (F1 = 0.96) outperformed the clinician (F1 = 0.91) in Table 27, showing that the large gap is not attributable to preprocessing alone. Overall, preprocessing introduces some information loss for subtle perceptual events, but does not fully explain model underperformance — the impact is task-dependent.

Second, we conducted an ablation study on sampling rate: increasing from 2 FPS to 4 FPS and 10 FPS yields average Task 1 F1 improvements of 0.06 and 0.08, respectively in Table 29.

To accommodate the context-length restrictions of current VLMs while preserving enough precision required for seizure semilogy detection tasks, the maximum number of frames we can upload at one time is 60 frames. We thus conducted an experiment for Task 4 on full videos by uniformly sampling 60 frames. As shown in Table 30, full-video evaluation degrades temporal localization performance, with average MAE increasing by 4.91s and the clinically critical feature clonic showing an MAE increase of 11.39s. Critically, even under the current 60s event-centric setting, the best model (Qwen2.5-VL-32B) achieves an onset MAE of only 8.19 seconds — already well beyond the clinically acceptable precision required for ictal localization. This result demonstrates that temporal grounding is a fundamental limitation of current MLLMs. Full-video evaluation amplifies this deficiency further due to compounding reasons: (1) lower frame rates reduce temporal granularity. Nearly 0.5 FPS(average video duration is 120.35s) makes precise onset detection fundamentally harder; (2) longer video duration expands the maximum possible error.

## K. Qualitative Error Analysis

As shown in Figure 8, the confusion matrices reveal a consistent positive prediction bias: across nearly all 18 features, false positives (FP) substantially outnumber false negatives (FN), indicating the model systematically over-reports the presence of semiological signs. This is most pronounced for visually ambiguous or low-prevalence features — limb automatisms (FP=288, acc=0.33), blank stare (FP=242), face pulling (FP=272), and arms move simultaneously (FP=259) — where the model effectively defaults to "yes," likely conflating incidental limb or facial motion with clinically defined ictal phenomena.

The only feature showing well-calibrated discrimination is figure-4 (acc=0.76, FP=71), a posturally distinctive sign with clear visual boundaries. Occur during sleep also performs reasonably (acc=0.74), benefiting from unambiguous environmental cues.

For epileptic-seizure markers such as tonic and clonic, moderate FP rates suggest the model recognizes general movement rigidity or rhythmicity but lacks the clinical precision to distinguish ictal from non-ictal motor activity. This aligns with the paper's finding that MLLMs exhibit systematic perceptual weaknesses in fine-grained seizure semiology, particularly when subtle temporal dynamics and domain-specific thresholds are required.

## L. Seizure Videos Access

**Option 1: Secure access on UCLA infrastructure.** Any new use of the video dataset by additional investigators would require prior IRB review and ethical approval. Until such approval is obtained, the data cannot be shared with additional users. If approved, authorized users may analyze the videos within a secure UCLA-hosted environment, where the raw video files remain on institutional servers and are not distributed externally. This approach supports research use while minimizing risks related to copying, transfer, or redistribution of sensitive video data.

**Option 2: Institutional agreement with UCLA.** Alternatively, access may be considered through a formal data use agreement with UCLA, but only after appropriate IRB

and institutional approvals are in place. At present, the data cannot be shared with additional investigators or institutions without further ethical review and approval. Any approved access would require institutional review of the proposed use, agreement to restrictions on redistribution, re-identification, and unauthorized sharing, and compliance with all applicable IRB, privacy, and data-security requirements.

For access to and use of the seizure video data, please contact Rajarshi Mazumder at Rmazumder@mednet.ucla.edu.

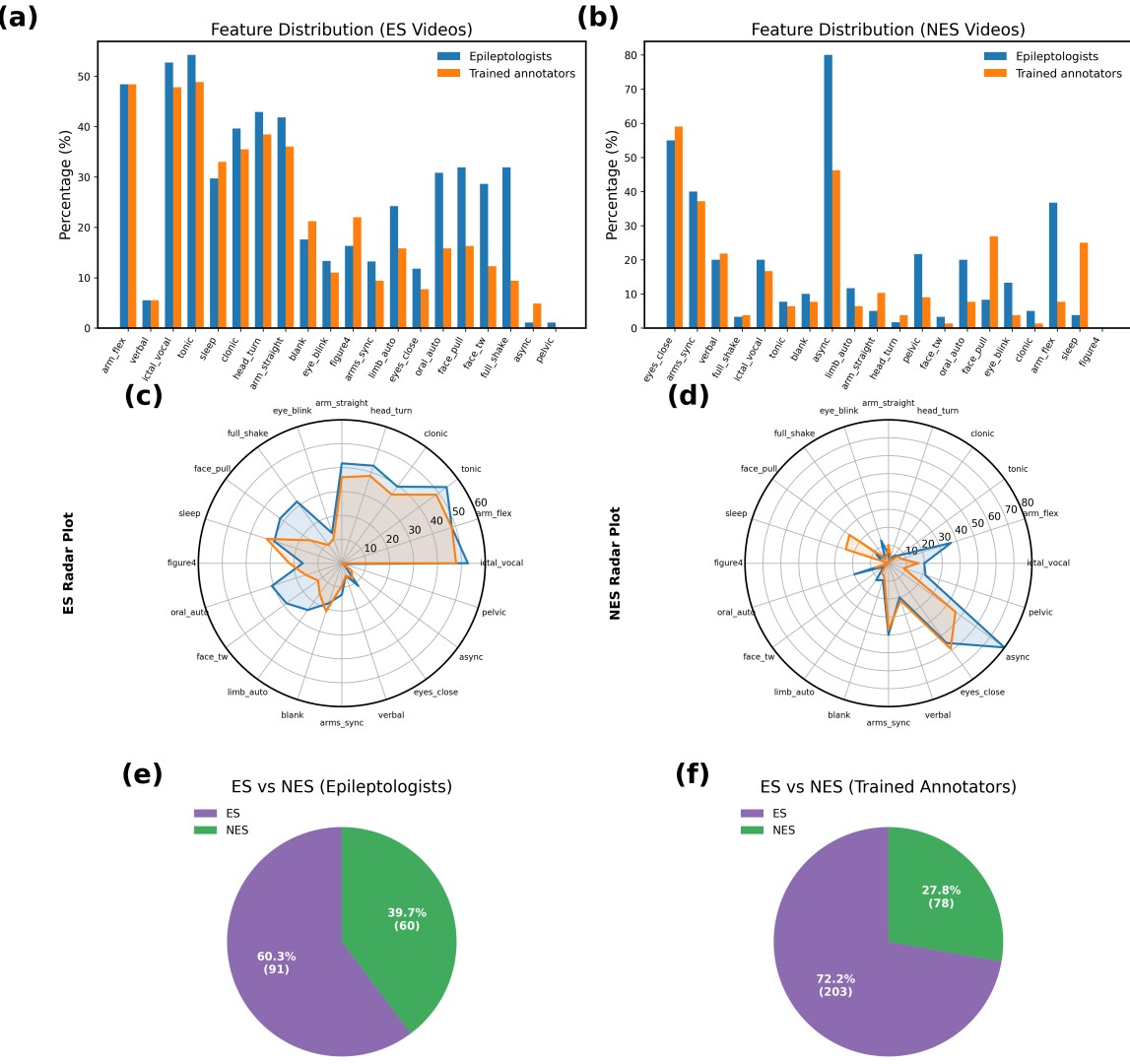

*Figure 7.* **Comparison of epileptologist and trained-annotator seizure semiology distributions across epileptic (ES) and nonepileptic (NES) events.** **(a)** Sorted bar plot of ES semiology feature frequencies annotated by epileptologists and trained annotators. Features are ordered by relative difference between groups, highlighting minimal divergence for high-frequency ES behaviors such as tonic, clonic, and ictal vocalization. **(b)** Analogous NES feature comparison showing similarly small discrepancies. **(c)** Radar plot (polar) illustrating the full 20-dimensional ES semiology profiles for experts and annotators. The strong overlap reflects high multivariate alignment, consistent with the quantitative similarity metrics ($r = 0.893$, cosine similarity = 0.966, EMD = 0.0086). **(d)** Radar plot for NES features, again showing substantial overlap between groups despite the greater behavioral heterogeneity of NES events (NES: $r = 0.782$, cosine = 0.883, EMD = 0.0075). **(e)** Pie chart summarizing the proportion of ES vs. NES videos labeled by epileptologists. **(f)** Corresponding ES/NES distribution for trained annotators. The nearly identical global composition in (e)–(f) confirms that agreement analyses are not confounded by differences in event prevalence. Together, panels (a)–(f) demonstrate that trained annotators reproduce expert-level seizure semiology distributions with high fidelity across both epileptic and nonepileptic events.

Table 7. Semiological features: importance, citations, and MLLM prompts/questions for Tasks 1 & 2 (Part 2).

| Feature | Importance / Citation | MLLM Prompt / Questions / Context |
|---|---|---|
| **Arm extension** | **Importance:** Contralateral to seizure onset in "figure-4" asymmetric tonic postures.
**PubMed citation:** (Kotagal et al., 2000) | **MLLM Prompt:** Does the patient have sustained arm extension? Answer with 'yes' or 'no'.
**Questions:** (1) Which arm is extended? (2) Is the extension sustained?
**Context:** N/A |
| **Figure-4 sign** | **Importance:** Asymmetric tonic posture; the flexed arm is ipsilateral, the extended arm contralateral to seizure onset.
**PubMed citation:** (Kotagal et al., 2000) | **MLLM Prompt:** Does the patient's posture resemble a "figure-4" pattern, with one arm flexed and the other extended? Answer with 'yes' or 'no'.
**Questions:** (1) Which arm is flexed? (2) Which arm is extended? (3) Is the posture sustained?
**Context:** N/A |
| **Head turning** | **Importance:** Forced head/eye deviation is strongly lateralizing (contralateral to seizure focus).
**PubMed citation:** (Kernan et al., 1993) | **MLLM Prompt:** Does the patient forcibly or stiffly rotate their head to one side in the video? Answer with 'yes' or 'no'.
**Questions:** (1) Does the head turn forcefully relative to shoulders to one side for at least a few seconds? (2) Does the head turn to the patients left or to the patients right?
**Context:** It must be distinguished from a normal head movement |
| **Blank stare** | **Importance:** Suggests impaired awareness, commonly seen in focal seizures (temporal lobe).
**PubMed citation:** (Tatum IV, 2012) | **MLLM Prompt:** Does the patient exhibit a blank stare? Answer with 'yes' or 'no'.
**Questions:** (1) Are the eyes open? (2) Do the iris of the eyes remain in a fixed position?
**Context:** Patient not having tonic posture or having convulsions |
| **Oral automatisms** | **Importance:** Lip-smacking, chewing point toward temporal lobe seizures with impaired awareness.
**PubMed citation:** (Yang et al., 2022) | **MLLM Prompt:** Does the patient exhibit repetitive, stereotyped mouth or tongue movements such as chewing, lip-smacking, or swallowing? Answer with 'yes' or 'no'.
**Questions:** (1) Does the patient have repetitive, stereotyped mouth or lip movements?
**Context:** Does not occur during full body tonic,tonic-clonic or clonic movements |
| **Limb automatisms** | **Importance:** Fumbling, cycling, or patting behaviors typical in focal seizures with impaired awareness.
**PubMed citation:** (Blumenfeld, 2012) | **MLLM Prompt:** Does the patient exhibit repetitive, stereotyped limb movements such as fumbling, picking, rubbing or patting? Answer with 'yes' or 'no'.
**Questions:** (1) Does the patient have repetitive stereotyped movements with their hands or legs? (2) For each involved limb, specify which side of the patient is involved?
**Context:** Does not occur during full body tonic,tonic-clonic or clonic movements |
| **Pelvic thrusting** | **Importance:** More frequent in NES than epilepsy; helps avoid misdiagnosis.
**PubMed citation:** (Duncan et al., 2022) | **MLLM Prompt:** Does the patient display repetitive, rhythmic, anteroposterior (forward-and-backward) movements of the hips? Answer with 'yes' or 'no'.
**Questions:** (1) Are there repetitive anterior posterior movements of the hips?
**Context:** Does not occur during full body tonic,tonic-clonic or clonic movements |

*Table 8.* Semiological features: importance, citations, and MLLM prompts/questions for Tasks 1 & 2 (Part 3).

| Feature | Importance / Citation | MLLM Prompt / Questions / Context |
|---|---|---|
| **Asynchronous movement** | **Importance:** Asynchronous, variable movements favor PNES; epileptic seizures are usually rhythmic and evolving.
**PubMed citation:** (Chaudhry et al., 2019; Vinton et al., 2004) | **MLLM Prompt:** Do you observe the patient's limbs shake with variable frequency or amplitude with respect to one another? Answer with 'yes' or 'no'.
**Questions:** (1) Which limbs of the patients are shaking? (2) Are the limbs shaking at variable frequency, amplitude or both? (3) Do the movements appear non-stereotyped? (4) Are the limb movements asynchronous with respect to one another?
**Context:** Does not occur during full body tonic,tonic-clonic or clonic movements |
| **Arms move simultaneously** | **Importance:** Stereotyped, simultaneous movements point toward epilepsy; asynchronous favors PNES.
**PubMed citation:** (Muthusamy et al., 2022) | **MLLM Prompt:** Do the patient's arms start moving approximately at the same time? Answer with 'yes' or 'no'.
**Questions:** (1) Did both arms begin to move approximately at the same time?
**Context:** N/A |
| **Full body shaking** | **Importance:** Asynchronous, variable movements favor PNES; epileptic seizures are usually rhythmic and evolving.
**PubMed citation:** (Chaudhry et al., 2019; Vinton et al., 2004) | **MLLM Prompt:** Does the patient experience shaking of the entire body including arms, legs, torso? Answer with 'yes' or 'no'.
**Questions:** (1) Does the patient exhibit full body shaking?
**Context:** Does not occur during full body tonic,tonic-clonic or clonic movements |
| **Close eyes** | **Importance:** Forced eye closure is strongly lateralizing (contralateral to seizure focus).
**PubMed citation:** (Kernan et al., 1993) | **MLLM Prompt:** Do the patient's eyes remain consistently closed or mostly closed throughout the video? Answer with 'yes' or 'no'.
**Questions:** (1) Does the closure appear forceful, as if being squeezed shut?
**Context:** N/A |
| **Eye blinking** | **Importance:** Repetitive eyelid myoclonia is a key marker in idiopathic generalized epilepsies.
**PubMed citation:** (Ahmed et al., 2025) | **MLLM Prompt:** Does the patient show rapid blinking of the eyes during the video? Answer with 'yes' or 'no'.
**Questions:** (1) Is the blinking happening at a frequency of 1Hz or more? (2) Does the eye blinking happen for at least a few seconds?
**Context:** N/A |
| **Verbal responsiveness** | **Importance:** Preserved responsiveness suggests non-epileptic events; loss of responsiveness is typical in focal to bilateral seizures.
**PubMed citation:** (Gedzelman & LaRoche, 2014) | **MLLM Prompt:** If the patient is addressed verbally by a different person, did they respond verbally in a coherent manner? Answer "yes" or "no". If the patient is not addressed verbally by a different person, then the answer should be 'NA'.
**Questions:** (1) Did anyone address the patient verbally? (2) If yes, did the patient provide a coherent answer?
**Context:** N/A |
| **Ictal vocalization** | **Importance:** Ictal cry or moan often precedes generalized tonic-clonic seizures; can help differentiate from PNES.
**PubMed citation:** (Elzawahry et al., 2010) | **MLLM Prompt:** Does the patient make any groaning, moaning, gutteral sounds or do they utter stereotyped repetitive phrases? Answer "yes" or "no".
**Questions:** (1) Is the patient making groaning, moaning or gutteral sounds? (2) Are the noises stereotyped and repetitive?
**Context:** Does not occur as normal speech |

*Table 9.* Prompts for Task 3, 4, 5, 6 and 7.

| | |
|---|---|
| **Task 3 Spatial & Anatomical Analysis** | |
| Head turning lateralization | Does the patient's head turn to the patient's left or to the patient's right? Answer with "left" or "right" only. Do not include any extra text. Return exactly one word: left or right. |
| Arm movement lateralization | Which arm of the patient is moving in the video? Answer with "left" or "right" only. Do not include any extra text. Return exactly one word: left or right. |
| Seizure onset localization | Localize which body part shows the earliest visible seizure sign. Answer only with one of the following options: head, eyes, mouth, face, left arm, left leg, right arm, right leg, arms, legs, full body. |
| **Task 4 Temporal Boundary Detection** | |
| Seizure start/end | This video shows the start/end of a seizure event. Output the exact timestamp (MM:SS) when you first observe any seizure sign. Return only the JSON format: {"timestamp": "MM:SS"} |
| Feature start | This video shows when a patient {description} Output the exact timestamp (MM:SS) when this symptom first appears. |
| **Task 5 Semiological Sequence Analysis** | |
| Semiological sequence | Output the sequence of any observed seizure symptoms of the patient in the video in chronological order. The symptoms are limited to head turning, blank stare, close eyes, eye blinking, face pulling, face twitching, tonic, clonic, arm straightening, arm flexion, figure-4, oral automatisms, limb automatisms, asynchronous movement, pelvic thrusting, full body shaking, arms move simultaneously. If a symptom is not present in the video, it should not be included in the output. Example output: head turning, arm straightening, arm flexion, tonic, clonic. Output only the seizure symptoms. Do not include any other text. |
| **Task 6 Holistic Narrative Report Generation** | |
| Clinical report generation | You are an expert clinician. Write a concise semiology description for this seizure video, focusing ONLY on observable patient signs. HARD RESTRICTIONS: (i) Do NOT mention staff, restraints, bed/blanket/pillow, room devices, cameras, EEG leads/overlays, or timestamps. (ii) Avoid vague words like "agitation", "restlessness", "discomfort", or "adjusting position". WHAT TO COVER (include an item ONLY if it is clearly visible in this video). (iii) Write 1–3 short sentences in English only, specific and minimal. (iv) Examples (no labels): "Blank stare, then rightward head version with right arm extension; later bilateral tonic–clonic." Output ONLY the paragraph (no lists, no headers, no JSON). |
| **Task 7 Clinical diagnosis and Reasoning** | |
| Seizure video + report | Based on the patient's seizure video and seizure semiology report, determine whether the patient has epileptic seizures (ES) or non-epileptic events (NES). Answer with 'ES' or 'NES' and do not include any other text. |
| Seizure video only | Describe the patient's seizure symptoms in the video and diagnose whether it is an epileptic seizure (ES) or a non-epileptic event (NES). Provide a description and answer with 'ES' or 'NES'. Respond with exactly one JSON object in the format {"description": "...","answer": "..."} and do not include any extra text outside of the JSON. |

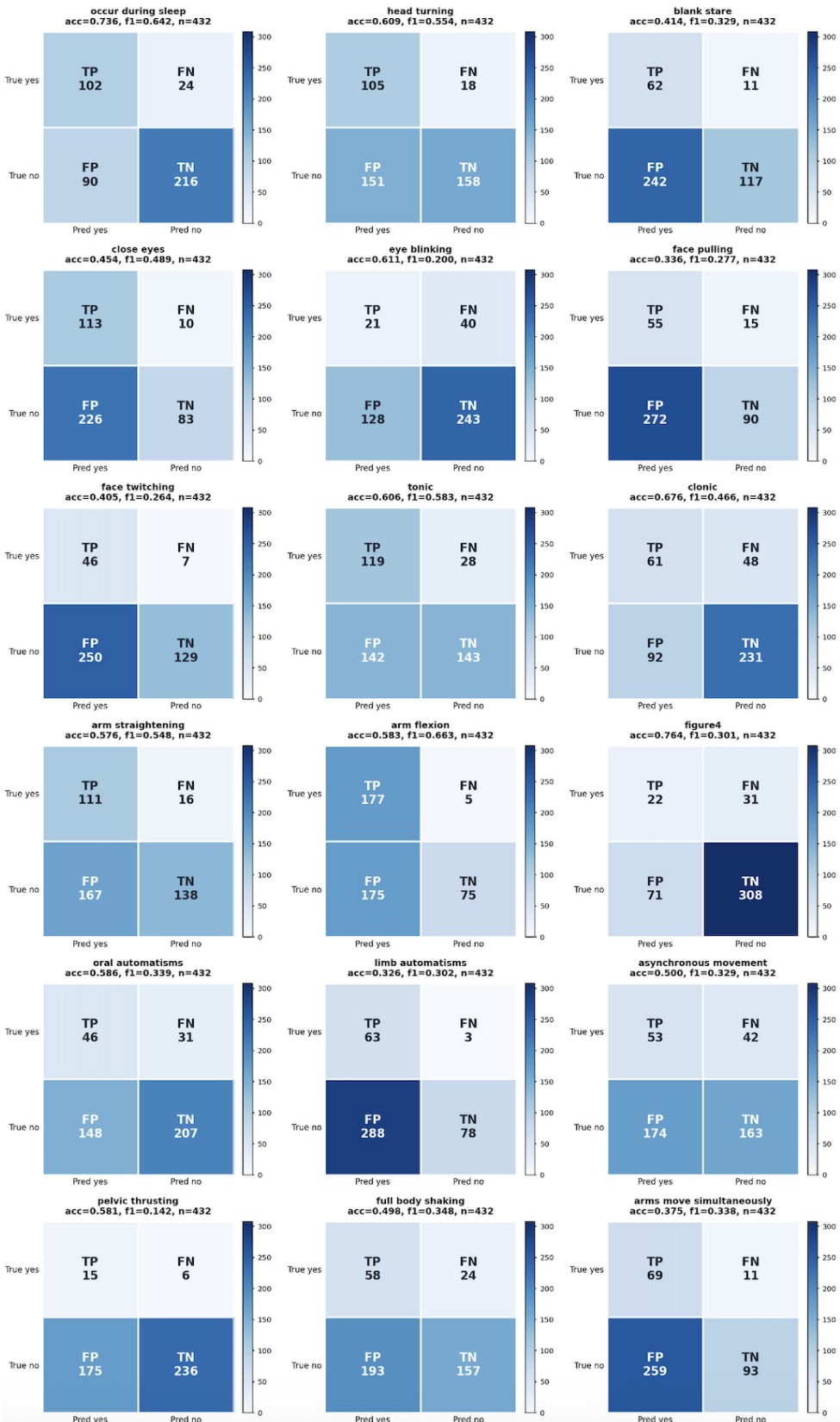

*Figure 8.* **Confusion matrix of Qwen2.5-VL-72B predictions on Task 1.**

*Table 10.* Comparison between report metrics and expert ratings.

| metric | pearson | spearman | kendall | pairwise_accuracy |
|---|---|---|---|---|
| bleu_corpus | 0.07 | 0.10 | 0.08 | 0.54 |
| rouge1_f1 | 0.09 | 0.10 | 0.07 | 0.52 |
| rougeL_f1 | 0.10 | 0.11 | 0.08 | 0.52 |
| berts_f1 | 0.07 | 0.12 | 0.08 | 0.52 |
| seizure rqi | **0.57** | **0.52** | **0.43** | **0.74** |

*Table 11.* Automatic report metrics and epileptologist ratings for 60 MLLM-generated reports on Task 6.

| model | video_id | bleu_corpus | rouge1_f1 | rougeL_f1 | berts_f1 | RQI | expert |
|---|---|---|---|---|---|---|---|
| InternVL3_5-38B | 1 | 3.52 | 0.32 | 0.24 | 0.60 | 47.57 | 2.00 |
| InternVL3_5-38B | 2 | 0.77 | 0.25 | 0.17 | 0.58 | 30.54 | 1.00 |
| InternVL3_5-38B | 3 | 0.57 | 0.09 | 0.05 | 0.54 | 51.92 | 2.50 |
| InternVL3_5-38B | 4 | 3.30 | 0.25 | 0.15 | 0.60 | 25.00 | 1.50 |
| InternVL3_5-38B | 5 | 0.86 | 0.18 | 0.13 | 0.55 | 32.00 | 2.00 |
| InternVL3_5-38B | 6 | 1.81 | 0.31 | 0.16 | 0.59 | 30.54 | 1.50 |
| InternVL3_5-38B | 7 | 4.56 | 0.34 | 0.18 | 0.61 | 34.81 | 1.50 |
| InternVL3_5-38B | 8 | 3.54 | 0.24 | 0.24 | 0.57 | 45.00 | 2.50 |
| InternVL3_5-38B | 9 | 1.48 | 0.17 | 0.11 | 0.58 | 38.58 | 2.00 |
| InternVL3_5-38B | 10 | 0.87 | 0.14 | 0.13 | 0.53 | 18.83 | 1.00 |
| InternVL3_5-38B | 11 | 1.01 | 0.21 | 0.11 | 0.52 | 30.54 | 1.00 |
| InternVL3_5-38B | 12 | 0.73 | 0.05 | 0.05 | 0.46 | 25.00 | 1.00 |
| InternVL3_5-38B | 13 | 0.70 | 0.12 | 0.09 | 0.55 | 42.97 | 2.00 |
| InternVL3_5-38B | 14 | 0.98 | 0.15 | 0.11 | 0.51 | 33.12 | 1.50 |
| InternVL3_5-38B | 15 | 1.53 | 0.21 | 0.18 | 0.64 | 48.73 | 1.50 |
| InternVL3_5-38B | 16 | 1.56 | 0.15 | 0.13 | 0.57 | 31.59 | 1.00 |
| InternVL3_5-38B | 17 | 1.32 | 0.13 | 0.11 | 0.50 | 19.73 | 3.00 |
| InternVL3_5-38B | 18 | 0.73 | 0.02 | 0.02 | 0.46 | 29.33 | 1.50 |
| InternVL3_5-38B | 19 | 1.08 | 0.21 | 0.12 | 0.52 | 49.58 | 1.00 |
| InternVL3_5-38B | 20 | 2.69 | 0.21 | 0.13 | 0.54 | 22.40 | 2.00 |
| Qwen2.5-VL-72B | 1 | 3.07 | 0.37 | 0.19 | 0.61 | 38.92 | 2.00 |
| Qwen2.5-VL-72B | 2 | 1.82 | 0.28 | 0.20 | 0.57 | 33.85 | 1.50 |
| Qwen2.5-VL-72B | 3 | 2.08 | 0.16 | 0.13 | 0.61 | 44.52 | 2.00 |
| Qwen2.5-VL-72B | 4 | 2.24 | 0.26 | 0.16 | 0.55 | 28.18 | 1.50 |
| Qwen2.5-VL-72B | 5 | 1.08 | 0.14 | 0.07 | 0.52 | 47.50 | 2.50 |
| Qwen2.5-VL-72B | 6 | 1.99 | 0.24 | 0.12 | 0.59 | 41.74 | 2.00 |
| Qwen2.5-VL-72B | 7 | 2.38 | 0.31 | 0.12 | 0.62 | 55.28 | 2.50 |
| Qwen2.5-VL-72B | 8 | 1.24 | 0.10 | 0.07 | 0.54 | 45.37 | 2.50 |
| Qwen2.5-VL-72B | 9 | 1.48 | 0.15 | 0.15 | 0.58 | 58.94 | 3.00 |
| Qwen2.5-VL-72B | 10 | 0.66 | 0.17 | 0.09 | 0.49 | 23.23 | 1.00 |
| Qwen2.5-VL-72B | 11 | 0.26 | 0.13 | 0.08 | 0.49 | 60.02 | 3.00 |
| Qwen2.5-VL-72B | 12 | 1.83 | 0.10 | 0.10 | 0.50 | 43.94 | 1.00 |
| Qwen2.5-VL-72B | 13 | 0.89 | 0.16 | 0.08 | 0.56 | 33.35 | 1.50 |
| Qwen2.5-VL-72B | 14 | 2.21 | 0.20 | 0.12 | 0.55 | 39.30 | 2.00 |
| Qwen2.5-VL-72B | 15 | 1.94 | 0.16 | 0.05 | 0.60 | 39.87 | 1.00 |
| Qwen2.5-VL-72B | 16 | 0.65 | 0.11 | 0.06 | 0.56 | 48.23 | 3.00 |
| Qwen2.5-VL-72B | 17 | 3.22 | 0.18 | 0.11 | 0.52 | 22.73 | 2.50 |
| Qwen2.5-VL-72B | 18 | 0.46 | 0.04 | 0.04 | 0.42 | 45.90 | 2.50 |
| Qwen2.5-VL-72B | 19 | 0.96 | 0.19 | 0.11 | 0.47 | 43.81 | 1.50 |
| Qwen2.5-VL-72B | 20 | 1.41 | 0.17 | 0.07 | 0.55 | 28.69 | 2.00 |
| Qwen2.5-Omni-7B | 1 | 1.74 | 0.30 | 0.19 | 0.63 | 69.94 | 3.00 |
| Qwen2.5-Omni-7B | 2 | 0.44 | 0.17 | 0.10 | 0.58 | 22.97 | 1.00 |
| Qwen2.5-Omni-7B | 3 | 2.16 | 0.11 | 0.11 | 0.68 | 34.62 | 1.50 |
| Qwen2.5-Omni-7B | 4 | 1.23 | 0.08 | 0.08 | 0.57 | 21.75 | 1.00 |
| Qwen2.5-Omni-7B | 5 | 0.25 | 0.08 | 0.08 | 0.57 | 48.75 | 2.50 |
| Qwen2.5-Omni-7B | 6 | 0.78 | 0.16 | 0.11 | 0.59 | 54.98 | 2.50 |
| Qwen2.5-Omni-7B | 7 | 1.30 | 0.24 | 0.15 | 0.63 | 51.89 | 2.50 |
| Qwen2.5-Omni-7B | 8 | 1.89 | 0.09 | 0.09 | 0.55 | 30.20 | 1.50 |
| Qwen2.5-Omni-7B | 9 | 1.66 | 0.13 | 0.09 | 0.56 | 41.27 | 2.00 |
| Qwen2.5-Omni-7B | 10 | 1.28 | 0.07 | 0.07 | 0.52 | 23.54 | 1.00 |
| Qwen2.5-Omni-7B | 11 | 0.21 | 0.09 | 0.09 | 0.51 | 22.40 | 1.00 |
| Qwen2.5-Omni-7B | 12 | 1.96 | 0.10 | 0.05 | 0.50 | 45.46 | 1.00 |
| Qwen2.5-Omni-7B | 13 | 1.62 | 0.14 | 0.07 | 0.62 | 40.82 | 1.00 |
| Qwen2.5-Omni-7B | 14 | 0.21 | 0.05 | 0.02 | 0.55 | 27.31 | 1.50 |
| Qwen2.5-Omni-7B | 15 | 2.18 | 0.29 | 0.23 | 0.66 | 42.47 | 1.50 |
| Qwen2.5-Omni-7B | 16 | 0.50 | 0.06 | 0.06 | 0.54 | 31.92 | 1.00 |
| Qwen2.5-Omni-7B | 17 | 2.27 | 0.15 | 0.15 | 0.55 | 23.08 | 2.00 |
| Qwen2.5-Omni-7B | 18 | 1.48 | 0.05 | 0.05 | 0.51 | 38.37 | 2.00 |
| Qwen2.5-Omni-7B | 19 | 2.14 | 0.24 | 0.15 | 0.59 | 31.67 | 1.50 |
| Qwen2.5-Omni-7B | 20 | 1.14 | 0.13 | 0.10 | 0.55 | 28.69 | 1.00 |

*Table 12.* **Sample distribution across tasks in the finetuning training dataset.** The finetuning set covers seven clinical reasoning tasks, including recognition, justification, spatial/laterality reasoning, temporal boundary localization, sequence analysis, report generation, and diagnosis. Tasks 1–2 contain the largest number of samples, reflecting the dense feature-level recognition and explanation annotations used for model adaptation. For Task 3, the 527 samples include 98 head turning samples and 83 arm movement samples.

| Task | Samples |
|---|---|
| Tasks 1–2 (Recognition + Justification) | 3242 |
| Task 3 (Spatial/Laterality) | 527 |
| Task 4 (Temporal Boundary) | 1926 |
| Task 5 (Sequence Analysis) | 1743 |
| Task 6 (Report Generation) | 1787 |
| Task 7 (Diagnosis) | 700 |

*Table 13.* Task 1 (Seizure Semiology Recognition) Metrics on Whole Dataset (438 videos)

| Model | Occur During Sleep | | | | Head Turning | | | | Blank Stare | | | | Close Eyes | | | |
|---|---|---|---|---|---|---|---|---|---|---|---|---|---|---|---|---|
| | P | R | F1 | A | P | R | F1 | A | P | R | F1 | A | P | R | F1 | A |
| InternVL3_5-8B | 0.41 | 0.83 | 0.55 | 0.60 | 0.80 | 0.07 | 0.12 | 0.73 | 0.17 | 1.00 | 0.29 | 0.18 | 0.29 | 1.00 | 0.44 | 0.29 |
| Qwen2.5-VL-7B | 0.40 | 0.90 | 0.55 | 0.57 | 0.34 | 0.98 | 0.51 | 0.46 | 0.18 | 1.00 | 0.30 | 0.21 | 0.29 | 1.00 | 0.45 | 0.31 |
| Qwen3-VL-8B | 0.54 | 0.67 | 0.60 | 0.74 | 1.00 | 0.03 | 0.06 | 0.73 | 0.79 | 0.30 | 0.37 | 0.35 | 0.83 | 0.49 | 0.51 | 0.00 |
| InternVL3_5-38B | 0.51 | 0.88 | 0.65 | 0.72 | 0.00 | 0.00 | 0.00 | 0.18 | 0.99 | 0.30 | 0.23 | 0.28 | 1.00 | 0.44 | 0.28 | 0.08 |
| Qwen2.5-VL-32B | 0.61 | 0.51 | 0.55 | 0.76 | 0.60 | 0.25 | 0.35 | 0.74 | 0.20 | 0.88 | 0.32 | 0.37 | 0.35 | 0.88 | 0.50 | 0.50 |
| Qwen3-VL-32B | 0.60 | 0.74 | 0.66 | 0.78 | 0.54 | 0.46 | 0.50 | 0.74 | 0.19 | 0.86 | 0.31 | 0.36 | 0.35 | 0.82 | 0.49 | 0.51 |
| Qwen2.5-VL-72B | 0.53 | 0.81 | 0.64 | 0.74 | 0.41 | 0.85 | 0.55 | 0.61 | 0.20 | 0.85 | 0.33 | 0.41 | 0.33 | 0.92 | 0.49 | 0.45 |
| audio-flamingo-3 | - | - | - | - | - | - | - | - | - | - | - | - | - | - | - | - |
| Qwen2.5-Omni-7B | 0.41 | 0.93 | 0.57 | 0.59 | 0.53 | 0.67 | 0.59 | 0.73 | 0.19 | 1.00 | 0.32 | 0.22 | 0.30 | 1.00 | 0.46 | 0.30 |
| Qwen3-Omni-30B | 0.50 | 0.82 | 0.62 | 0.71 | 0.81 | 0.43 | 0.56 | 0.81 | 0.21 | 0.73 | 0.32 | 0.41 | 0.31 | 0.81 | 0.44 | 0.62 |
| Lingshu-32B | 0.54 | 0.75 | 0.63 | 0.74 | 0.41 | 0.72 | 0.52 | 0.63 | 0.18 | 0.97 | 0.31 | 0.26 | 0.31 | 0.98 | 0.47 | 0.36 |

| Model | Eye Blinking | | | | Face Pulling | | | | Face Twitching | | | | Tonic | | | |
|---|---|---|---|---|---|---|---|---|---|---|---|---|---|---|---|---|
| | P | R | F1 | A | P | R | F1 | A | P | R | F1 | A | P | R | F1 | A |
| InternVL3_5-8B | 0.00 | 0.00 | 0.00 | 0.86 | 0.30 | 0.09 | 0.13 | 0.82 | 0.12 | 1.00 | 0.22 | 0.12 | 0.51 | 0.48 | 0.50 | 0.67 |
| Qwen2.5-VL-7B | 0.18 | 0.90 | 0.29 | 0.39 | 0.19 | 0.71 | 0.30 | 0.45 | 0.13 | 0.98 | 0.24 | 0.22 | 0.31 | 0.07 | 0.12 | 0.63 |
| Qwen3-VL-8B | 0.00 | 0.00 | 0.00 | 0.86 | 0.00 | 0.00 | 0.00 | 0.84 | 0.13 | 1.00 | 0.23 | 0.18 | 0.58 | 0.52 | 0.54 | 0.71 |
| InternVL3_5-38B | 0.05 | 0.06 | 0.79 | 0.19 | 0.60 | 0.29 | 0.52 | 0.12 | 1.00 | 0.22 | 0.12 | 0.61 | 0.47 | 0.53 | 0.72 | 0.38 |
| Qwen2.5-VL-32B | 0.23 | 0.36 | 0.28 | 0.74 | 0.18 | 0.14 | 0.16 | 0.75 | 0.14 | 0.96 | 0.24 | 0.27 | 0.73 | 0.22 | 0.34 | 0.71 |
| Qwen3-VL-32B | 0.29 | 0.08 | 0.13 | 0.84 | 0.20 | 0.26 | 0.22 | 0.71 | 0.13 | 0.98 | 0.22 | 0.18 | 0.55 | 0.52 | 0.53 | 0.70 |
| Qwen2.5-VL-72B | 0.14 | 0.34 | 0.20 | 0.61 | 0.17 | 0.79 | 0.28 | 0.34 | 0.16 | 0.87 | 0.26 | 0.41 | 0.46 | 0.81 | 0.58 | 0.61 |
| audio-flamingo-3 | - | - | - | - | - | - | - | - | - | - | - | - | - | - | - | - |
| Qwen2.5-Omni-7B | 0.21 | 0.34 | 0.26 | 0.73 | 0.28 | 0.37 | 0.32 | 0.73 | 0.14 | 1.00 | 0.24 | 0.16 | 0.50 | 0.02 | 0.04 | 0.65 |
| Qwen3-Omni-30B | 0.05 | 0.06 | 0.05 | 0.79 | 0.21 | 0.47 | 0.29 | 0.59 | 0.16 | 1.00 | 0.28 | 0.24 | 0.61 | 0.64 | 0.63 | 0.71 |
| Lingshu-32B | 0.15 | 0.49 | 0.23 | 0.53 | 0.18 | 0.60 | 0.28 | 0.49 | 0.13 | 0.98 | 0.23 | 0.21 | 0.58 | 0.38 | 0.46 | 0.69 |

| Model | Clonic | | | | Arm Straightening | | | | Arm Flexion | | | | Figure 4 | | | |
|---|---|---|---|---|---|---|---|---|---|---|---|---|---|---|---|---|
| | P | R | F1 | A | P | R | F1 | A | P | R | F1 | A | P | R | F1 | A |
| InternVL3_5-8B | 0.45 | 0.17 | 0.25 | 0.74 | 0.43 | 0.76 | 0.55 | 0.63 | 0.48 | 0.98 | 0.65 | 0.55 | 0.34 | 0.23 | 0.27 | 0.85 |
| Qwen2.5-VL-7B | 0.39 | 0.39 | 0.39 | 0.69 | 0.35 | 0.98 | 0.52 | 0.46 | 0.46 | 0.99 | 0.63 | 0.51 | 0.32 | 0.23 | 0.26 | 0.84 |
| Qwen3-VL-8B | 0.42 | 0.16 | 0.23 | 0.74 | 0.48 | 0.72 | 0.58 | 0.69 | 0.60 | 0.77 | 0.67 | 0.69 | 0.75 | 0.06 | 0.11 | 0.88 |
| InternVL3_5-38B | 0.38 | 0.38 | 0.38 | 0.69 | 0.41 | 0.71 | 0.52 | 0.62 | 0.60 | 0.90 | 0.72 | 0.71 | 0.27 | 0.57 | 0.36 | 0.75 |
| Qwen2.5-VL-32B | 0.46 | 0.33 | 0.38 | 0.73 | 0.45 | 0.83 | 0.59 | 0.66 | 0.57 | 0.88 | 0.69 | 0.67 | 0.50 | 0.25 | 0.33 | 0.88 |
| Qwen3-VL-32B | 0.56 | 0.41 | 0.48 | 0.77 | 0.40 | 0.87 | 0.54 | 0.58 | 0.56 | 0.85 | 0.68 | 0.66 | 0.50 | 0.36 | 0.42 | 0.88 |
| Qwen2.5-VL-72B | 0.40 | 0.56 | 0.47 | 0.68 | 0.40 | 0.87 | 0.55 | 0.58 | 0.50 | 0.97 | 0.66 | 0.58 | 0.24 | 0.42 | 0.30 | 0.76 |
| audio-flamingo-3 | - | - | - | - | - | - | - | - | - | - | - | - | - | - | - | - |
| Qwen2.5-Omni-7B | 0.30 | 0.03 | 0.06 | 0.73 | 0.33 | 0.97 | 0.49 | 0.42 | 0.45 | 0.99 | 0.62 | 0.49 | 0.34 | 0.48 | 0.40 | 0.81 |
| Qwen3-Omni-30B | 0.31 | 0.97 | 0.47 | 0.51 | 0.38 | 0.96 | 0.54 | 0.57 | 0.53 | 0.97 | 0.69 | 0.62 | 0.25 | 0.58 | 0.35 | 0.76 |
| Lingshu-32B | 0.52 | 0.28 | 0.37 | 0.75 | 0.39 | 0.89 | 0.54 | 0.56 | 0.52 | 0.96 | 0.67 | 0.61 | 0.21 | 0.72 | 0.33 | 0.64 |

| Model | Oral Automatisms | | | | Limb Automatisms | | | | Asynchronous Movement | | | | Pelvic Thrusting | | | |
|---|---|---|---|---|---|---|---|---|---|---|---|---|---|---|---|---|
| | P | R | F1 | A | P | R | F1 | A | P | R | F1 | A | P | R | F1 | A |
| InternVL3_5-8B | 0.19 | 0.87 | 0.32 | 0.33 | 0.17 | 0.88 | 0.29 | 0.33 | 0.33 | 0.02 | 0.04 | 0.78 | 0.05 | 0.10 | 0.07 | 0.88 |
| Qwen2.5-VL-7B | 0.24 | 0.66 | 0.36 | 0.57 | 0.19 | 0.89 | 0.31 | 0.38 | 0.28 | 0.56 | 0.38 | 0.59 | 0.08 | 0.33 | 0.13 | 0.79 |
| Qwen3-VL-8B | 0.54 | 0.09 | 0.16 | 0.83 | 0.19 | 0.61 | 0.29 | 0.56 | 0.24 | 0.11 | 0.15 | 0.74 | 0.14 | 0.24 | 0.17 | 0.89 |
| InternVL3_5-38B | 0.23 | 0.45 | 0.31 | 0.63 | 0.19 | 0.91 | 0.31 | 0.38 | 0.26 | 0.39 | 0.31 | 0.62 | 0.06 | 0.38 | 0.10 | 0.67 |
| Qwen2.5-VL-32B | 0.23 | 0.49 | 0.32 | 0.62 | 0.18 | 0.76 | 0.29 | 0.43 | 0.29 | 0.25 | 0.27 | 0.70 | 0.05 | 0.24 | 0.09 | 0.75 |
| Qwen3-VL-32B | 0.24 | 0.53 | 0.33 | 0.62 | 0.19 | 0.88 | 0.31 | 0.40 | 0.26 | 0.43 | 0.33 | 0.61 | 0.07 | 0.24 | 0.11 | 0.82 |
| Qwen2.5-VL-72B | 0.24 | 0.60 | 0.34 | 0.59 | 0.18 | 0.95 | 0.30 | 0.33 | 0.23 | 0.56 | 0.33 | 0.50 | 0.08 | 0.71 | 0.14 | 0.58 |
| audio-flamingo-3 | - | - | - | - | - | - | - | - | - | - | - | - | - | - | - | - |
| Qwen2.5-Omni-7B | 0.25 | 0.73 | 0.37 | 0.53 | 0.21 | 0.53 | 0.30 | 0.59 | 0.21 | 0.68 | 0.33 | 0.47 | 0.05 | 0.32 | 0.09 | 0.67 |
| Qwen3-Omni-30B | 0.19 | 0.46 | 0.27 | 0.64 | 0.17 | 0.56 | 0.26 | 0.49 | 0.24 | 0.68 | 0.36 | 0.64 | 0.04 | 0.80 | 0.07 | 0.42 |
| Lingshu-32B | 0.23 | 0.58 | 0.33 | 0.57 | 0.20 | 0.77 | 0.32 | 0.50 | 0.24 | 0.25 | 0.24 | 0.66 | 0.06 | 0.38 | 0.11 | 0.70 |

| Model | Full Body Shaking | | | | Arms Move Simultaneously | | | | Verbal Responsiveness | | | | Ictal Vocalization | | | |
|---|---|---|---|---|---|---|---|---|---|---|---|---|---|---|---|---|
| | P | R | F1 | A | P | R | F1 | A | P | R | F1 | A | P | R | F1 | A |
| InternVL3_5-8B | 0.15 | 0.12 | 0.14 | 0.71 | 0.23 | 0.54 | 0.32 | 0.58 | - | - | - | - | - | - | - | - |
| Qwen2.5-VL-7B | 0.25 | 0.89 | 0.39 | 0.47 | 0.19 | 0.99 | 0.32 | 0.22 | - | - | - | - | - | - | - | - |
| Qwen3-VL-8B | 0.26 | 0.34 | 0.29 | 0.69 | 0.22 | 0.45 | 0.30 | 0.61 | - | - | - | - | - | - | - | - |
| InternVL3_5-38B | 0.20 | 0.48 | 0.28 | 0.53 | 0.19 | 0.99 | 0.31 | 0.20 | - | - | - | - | - | - | - | - |
| Qwen2.5-VL-32B | 0.25 | 0.30 | 0.27 | 0.69 | 0.24 | 0.56 | 0.34 | 0.59 | - | - | - | - | - | - | - | - |
| Qwen3-VL-32B | 0.27 | 0.65 | 0.38 | 0.61 | 0.21 | 0.82 | 0.34 | 0.41 | - | - | - | - | - | - | - | - |
| Qwen2.5-VL-72B | 0.23 | 0.71 | 0.35 | 0.50 | 0.21 | 0.86 | 0.34 | 0.38 | - | - | - | - | - | - | - | - |
| audio-flamingo-3 | - | - | - | - | - | - | - | - | 0.38 | 0.22 | 0.27 | 0.31 | 0.55 | 0.48 | 0.51 | 0.65 |
| Qwen2.5-Omni-7B | 0.19 | 0.94 | 0.32 | 0.35 | 0.17 | 1.00 | 0.28 | 0.17 | 0.69 | 0.70 | 0.69 | 0.75 | 0.54 | 0.94 | 0.69 | 0.66 |
| Qwen3-Omni-30B | 0.25 | 0.85 | 0.39 | 0.59 | 0.10 | 1.00 | 0.19 | 0.14 | 0.80 | 0.61 | 0.62 | 0.77 | 0.62 | 0.75 | 0.68 | 0.75 |
| Lingshu-32B | 0.26 | 0.55 | 0.35 | 0.62 | 0.19 | 0.94 | 0.32 | 0.25 | - | - | - | - | - | - | - | - |

*Table 14.* Task 2 (Feature Justification) Metrics (R:ROUGE, B:BLEU, BS:BERTScore, S:Semiology Interpretation Score) on Whole Dataset (438 videos)

| 2*Model | Occur During Sleep | | | | Head Turning | | | | Blank Stare | | | | Close Eyes | | | |
|---|---|---|---|---|---|---|---|---|---|---|---|---|---|---|---|---|
| | R | B | BS | S | R | B | BS | S | R | B | BS | S | R | B | BS | S |
| InternVL3.5-8B | 0.32 | 4.85 | 0.66 | 0.53 | 0.37 | 4.91 | 0.66 | 0.70 | 0.25 | 1.83 | 0.63 | 0.14 | 0.38 | 9.10 | 0.68 | 0.23 |
| Qwen2.5-VL-7B | 0.27 | 3.87 | 0.64 | 0.51 | 0.39 | 4.84 | 0.67 | 0.35 | 0.29 | 3.91 | 0.64 | 0.12 | 0.34 | 3.95 | 0.66 | 0.25 |
| Qwen3-VL-8B | 0.38 | 9.13 | 0.69 | 0.69 | 0.38 | 7.36 | 0.67 | 0.71 | 0.33 | 6.31 | 0.65 | 0.28 | 0.38 | 10.20 | 0.68 | 0.46 |
| InternVL3.5-38B | 0.31 | 4.92 | 0.65 | 0.64 | 0.34 | 4.16 | 0.65 | 0.69 | 0.25 | 1.73 | 0.62 | 0.18 | 0.37 | 5.18 | 0.67 | 0.22 |
| Qwen2.5-VL-32B | 0.27 | 4.29 | 0.64 | 0.65 | 0.37 | 6.83 | 0.68 | 0.70 | 0.27 | 3.37 | 0.62 | 0.29 | 0.36 | 9.36 | 0.66 | 0.45 |
| Qwen3-VL-32B | 0.34 | 7.33 | 0.66 | 0.71 | 0.37 | 5.88 | 0.66 | 0.69 | 0.35 | 7.01 | 0.66 | 0.30 | 0.37 | 8.64 | 0.67 | 0.45 |
| Qwen2.5-VL-72B | 0.28 | 3.63 | 0.65 | 0.67 | 0.38 | 5.86 | 0.68 | 0.53 | 0.30 | 3.97 | 0.65 | 0.31 | 0.38 | 10.13 | 0.66 | 0.40 |
| Audio-flamingo-3 | - | - | - | - | - | - | - | - | - | - | - | - | - | - | - | - |
| Qwen2.5-Omni-7B | 0.29 | 3.00 | 0.66 | 0.53 | 0.35 | 3.46 | 0.67 | 0.65 | 0.29 | 3.99 | 0.64 | 0.19 | 0.39 | 10.65 | 0.69 | 0.24 |
| Qwen3-Omni-30B | 0.35 | 5.13 | 0.67 | 0.67 | 0.40 | 8.57 | 0.68 | 0.75 | 0.34 | 6.81 | 0.66 | 0.31 | 0.44 | 13.85 | 0.71 | 0.62 |
| Lingshu-32B | 0.28 | 3.00 | 0.65 | 0.65 | 0.38 | 6.80 | 0.67 | 0.56 | 0.30 | 5.20 | 0.64 | 0.21 | 0.37 | 9.35 | 0.67 | 0.30 |

| 2*Model | Eye Blinking | | | | Face Pulling | | | | Face Twitching | | | | Tonic | | | |
|---|---|---|---|---|---|---|---|---|---|---|---|---|---|---|---|---|
| | R | B | BS | S | R | B | BS | S | R | B | BS | S | R | B | BS | S |
| InternVL3.5-8B | 0.39 | 6.55 | 0.68 | 0.86 | 0.25 | 1.33 | 0.60 | 0.81 | 0.33 | 5.01 | 0.64 | 0.09 | 0.33 | 4.18 | 0.64 | 0.58 |
| Qwen2.5-VL-7B | 0.36 | 5.10 | 0.67 | 0.36 | 0.29 | 2.16 | 0.60 | 0.38 | 0.29 | 1.55 | 0.63 | 0.19 | 0.30 | 3.64 | 0.63 | 0.56 |
| Qwen3-VL-8B | 0.40 | 5.90 | 0.70 | 0.86 | 0.29 | 1.88 | 0.60 | 0.84 | 0.29 | 2.79 | 0.64 | 0.14 | 0.36 | 4.88 | 0.65 | 0.61 |
| InternVL3.5-38B | 0.33 | 2.80 | 0.64 | 0.79 | 0.28 | 1.48 | 0.61 | 0.45 | 0.31 | 3.15 | 0.64 | 0.09 | 0.33 | 3.84 | 0.65 | 0.61 |
| Qwen2.5-VL-32B | 0.38 | 5.04 | 0.68 | 0.73 | 0.31 | 3.25 | 0.63 | 0.73 | 0.30 | 2.35 | 0.63 | 0.23 | 0.32 | 3.30 | 0.63 | 0.62 |
| Qwen3-VL-32B | 0.46 | 8.58 | 0.72 | 0.84 | 0.28 | 2.11 | 0.60 | 0.69 | 0.31 | 3.22 | 0.64 | 0.15 | 0.34 | 4.79 | 0.65 | 0.61 |
| Qwen2.5-VL-72B | 0.42 | 9.60 | 0.69 | 0.60 | 0.31 | 2.76 | 0.62 | 0.25 | 0.30 | 3.36 | 0.65 | 0.37 | 0.35 | 5.12 | 0.67 | 0.51 |
| Audio-flamingo-3 | - | - | - | - | - | - | - | - | - | - | - | - | - | - | - | - |
| Qwen2.5-Omni-7B | 0.45 | 11.86 | 0.71 | 0.72 | 0.32 | 1.92 | 0.65 | 0.69 | 0.31 | 1.86 | 0.66 | 0.12 | 0.29 | 4.36 | 0.65 | 0.61 |
| Qwen3-Omni-30B | 0.44 | 10.95 | 0.71 | 0.79 | 0.35 | 5.66 | 0.64 | 0.54 | 0.35 | 4.62 | 0.67 | 0.21 | 0.34 | 5.64 | 0.66 | 0.60 |
| Lingshu-32B | 0.42 | 7.77 | 0.70 | 0.51 | 0.30 | 3.41 | 0.64 | 0.43 | 0.28 | 2.43 | 0.65 | 0.16 | 0.33 | 4.32 | 0.64 | 0.59 |

| 2*Model | Clonic | | | | Arm Straightening | | | | Arm Flexion | | | | Figure 4 | | | |
|---|---|---|---|---|---|---|---|---|---|---|---|---|---|---|---|---|
| | R | B | BS | S | R | B | BS | S | R | B | BS | S | R | B | BS | S |
| InternVL3.5-8B | 0.35 | 4.27 | 0.65 | 0.69 | 0.35 | 4.43 | 0.65 | 0.57 | 0.31 | 2.23 | 0.64 | 0.44 | 0.41 | 5.99 | 0.66 | 0.76 |
| Qwen2.5-VL-7B | 0.33 | 3.85 | 0.65 | 0.64 | 0.33 | 3.51 | 0.63 | 0.40 | 0.30 | 3.49 | 0.63 | 0.42 | 0.41 | 6.07 | 0.67 | 0.65 |
| Qwen3-VL-8B | 0.36 | 4.11 | 0.67 | 0.69 | 0.37 | 6.06 | 0.66 | 0.64 | 0.30 | 3.96 | 0.62 | 0.57 | 0.43 | 6.13 | 0.68 | 0.78 |
| InternVL3.5-38B | 0.35 | 3.93 | 0.65 | 0.64 | 0.29 | 2.69 | 0.62 | 0.58 | 0.32 | 3.82 | 0.64 | 0.58 | 0.36 | 4.04 | 0.65 | 0.68 |
| Qwen2.5-VL-32B | 0.34 | 3.11 | 0.64 | 0.69 | 0.39 | 6.98 | 0.68 | 0.60 | 0.32 | 4.93 | 0.65 | 0.56 | 0.40 | 4.99 | 0.64 | 0.79 |
| Qwen3-VL-32B | 0.37 | 4.75 | 0.68 | 0.72 | 0.33 | 4.53 | 0.63 | 0.53 | 0.30 | 3.57 | 0.61 | 0.55 | 0.43 | 5.96 | 0.68 | 0.79 |
| Qwen2.5-VL-72B | 0.36 | 5.25 | 0.67 | 0.62 | 0.35 | 5.70 | 0.67 | 0.54 | 0.31 | 5.02 | 0.64 | 0.50 | 0.40 | 5.54 | 0.66 | 0.71 |
| Audio-flamingo-3 | - | - | - | - | - | - | - | - | - | - | - | - | - | - | - | - |
| Qwen2.5-Omni-7B | 0.32 | 2.45 | 0.65 | 0.69 | 0.33 | 3.03 | 0.65 | 0.37 | 0.30 | 3.35 | 0.64 | 0.43 | 0.39 | 4.74 | 0.66 | 0.73 |
| Qwen3-Omni-30B | 0.39 | 5.39 | 0.67 | 0.41 | 0.35 | 6.00 | 0.65 | 0.52 | 0.33 | 4.49 | 0.64 | 0.55 | 0.42 | 5.66 | 0.67 | 0.66 |
| Lingshu-32B | 0.33 | 4.03 | 0.65 | 0.71 | 0.35 | 4.68 | 0.66 | 0.50 | 0.30 | 3.08 | 0.64 | 0.51 | 0.42 | 6.03 | 0.67 | 0.57 |

| 2*Model | Oral Automatisms | | | | Limb Automatisms | | | | Asynchronous Movement | | | | Pelvic Thrusting | | | |
|---|---|---|---|---|---|---|---|---|---|---|---|---|---|---|---|---|
| | R | B | BS | S | R | B | BS | S | R | B | BS | S | R | B | BS | S |
| InternVL3.5-8B | 0.37 | 11.51 | 0.66 | 0.32 | 0.32 | 3.48 | 0.65 | 0.28 | 0.30 | 2.03 | 0.63 | 0.61 | 0.33 | 4.07 | 0.65 | 0.84 |
| Qwen2.5-VL-7B | 0.34 | 7.67 | 0.66 | 0.56 | 0.32 | 2.19 | 0.62 | 0.33 | 0.28 | 1.94 | 0.61 | 0.43 | 0.32 | 4.37 | 0.64 | 0.76 |
| Qwen3-VL-8B | 0.38 | 11.06 | 0.68 | 0.82 | 0.35 | 4.48 | 0.64 | 0.52 | 0.32 | 2.24 | 0.64 | 0.58 | 0.35 | 5.39 | 0.66 | 0.86 |
| InternVL3.5-38B | 0.36 | 9.63 | 0.67 | 0.62 | 0.32 | 3.41 | 0.65 | 0.33 | 0.24 | 1.77 | 0.57 | 0.44 | 0.35 | 5.40 | 0.66 | 0.65 |
| Qwen2.5-VL-32B | 0.36 | 8.99 | 0.67 | 0.61 | 0.30 | 3.22 | 0.65 | 0.38 | 0.30 | 2.04 | 0.62 | 0.56 | 0.30 | 3.50 | 0.65 | 0.73 |
| Qwen3-VL-32B | 0.33 | 8.72 | 0.65 | 0.60 | 0.29 | 2.79 | 0.60 | 0.36 | 0.32 | 2.73 | 0.64 | 0.52 | 0.33 | 4.55 | 0.64 | 0.79 |
| Qwen2.5-VL-72B | 0.37 | 9.20 | 0.68 | 0.57 | 0.29 | 2.50 | 0.66 | 0.28 | 0.30 | 2.60 | 0.62 | 0.43 | 0.34 | 5.76 | 0.67 | 0.56 |
| Audio-flamingo-3 | - | - | - | - | - | - | - | - | - | - | - | - | - | - | - | - |
| Qwen2.5-Omni-7B | 0.38 | 12.54 | 0.68 | 0.52 | 0.36 | 5.84 | 0.68 | 0.56 | 0.29 | 0.72 | 0.64 | 0.27 | 0.33 | 4.84 | 0.67 | 0.67 |
| Qwen3-Omni-30B | 0.35 | 7.44 | 0.67 | 0.62 | 0.32 | 3.69 | 0.62 | 0.47 | 0.32 | 2.57 | 0.64 | 0.55 | 0.34 | 5.05 | 0.64 | 0.41 |
| Lingshu-32B | 0.37 | 10.30 | 0.68 | 0.56 | 0.33 | 3.39 | 0.67 | 0.45 | 0.30 | 2.28 | 0.65 | 0.53 | 0.32 | 5.56 | 0.67 | 0.67 |

| 2*Model | Full Body Shaking | | | | Arms Move Simultaneously | | | | Verbal Responsiveness | | | | Ictal Vocalization | | | |
|---|---|---|---|---|---|---|---|---|---|---|---|---|---|---|---|---|
| | R | B | BS | S | R | B | BS | S | R | B | BS | S | R | B | BS | S |
| InternVL3.5-8B | 0.33 | 6.99 | 0.63 | 0.67 | 0.36 | 5.27 | 0.63 | 0.55 | - | - | - | - | - | - | - | - |
| Qwen2.5-VL-7B | 0.33 | 8.46 | 0.64 | 0.45 | 0.30 | 2.44 | 0.62 | 0.17 | - | - | - | - | - | - | - | - |
| Qwen3-VL-8B | 0.36 | 8.76 | 0.66 | 0.67 | 0.36 | 6.03 | 0.65 | 0.57 | - | - | - | - | - | - | - | - |
| InternVL3.5-38B | 0.36 | 9.26 | 0.66 | 0.51 | 0.22 | 0.80 | 0.54 | 0.10 | - | - | - | - | - | - | - | - |
| Qwen2.5-VL-32B | 0.33 | 6.73 | 0.64 | 0.66 | 0.37 | 6.83 | 0.62 | 0.56 | - | - | - | - | - | - | - | - |
| Qwen3-VL-32B | 0.35 | 9.34 | 0.65 | 0.58 | 0.33 | 3.96 | 0.64 | 0.37 | - | - | - | - | - | - | - | - |
| Qwen2.5-VL-72B | 0.36 | 10.77 | 0.67 | 0.48 | 0.33 | 4.72 | 0.64 | 0.33 | - | - | - | - | - | - | - | - |
| Audio-flamingo-3 | - | - | - | - | - | - | - | - | 0.17 | 0.66 | 0.62 | 0.51 | 0.26 | 6.87 | 0.65 | 0.55 |
| Qwen2.5-Omni-7B | 0.38 | 11.33 | 0.68 | 0.33 | 0.38 | 4.46 | 0.64 | 0.13 | 0.24 | 1.44 | 0.63 | 0.51 | 0.30 | 6.73 | 0.66 | 0.57 |
| Qwen3-Omni-30B | 0.38 | 9.76 | 0.67 | 0.58 | 0.36 | 3.46 | 0.65 | 0.11 | 0.29 | 3.72 | 0.62 | 0.62 | 0.29 | 4.87 | 0.64 | 0.64 |
| Lingshu-32B | 0.35 | 8.99 | 0.66 | 0.59 | 0.34 | 1.99 | 0.65 | 0.21 | - | - | - | - | - | - | - | - |

*Table 15.* Task 3 (Spatial and anatomical analysis) Metrics on Whole Dataset (438 videos)

| Model | Head turning | | | | Arm movement | | | | Onset body part | | | |
|---|---|---|---|---|---|---|---|---|---|---|---|---|
| | P | R | F1 | A | P | R | F1 | A | P | R | F1 | A |
| InternVL3.5-8B | 0.54 | 0.99 | 0.69 | 0.53 | 0.66 | 0.83 | 0.74 | 0.59 | 0.10 | 0.11 | 0.05 | 0.16 |
| Qwen2.5-VL-7B | 0.54 | 0.94 | 0.68 | 0.53 | 0.48 | 0.21 | 0.29 | 0.30 | 0.09 | 0.11 | 0.09 | 0.22 |
| Qwen3-VL-8B | 0.54 | 0.19 | 0.28 | 0.48 | 0.20 | 0.03 | 0.05 | 0.25 | 0.13 | 0.08 | 0.05 | 0.14 |
| InternVL3.5-38B | 0.54 | **1.00** | **0.70** | 0.54 | 0.67 | **0.89** | **0.76** | 0.62 | 0.10 | **0.12** | 0.07 | 0.15 |
| Qwen2.5-VL-32B | **0.62** | 0.49 | 0.55 | **0.56** | 0.55 | 0.34 | 0.42 | 0.35 | **0.15** | 0.09 | 0.08 | **0.33** |
| Qwen3-VL-32B | 0.52 | 0.74 | 0.61 | 0.49 | 0.74 | 0.59 | 0.66 | 0.57 | 0.14 | 0.11 | 0.08 | 0.15 |
| Qwen2.5-VL-72B | 0.50 | 0.10 | 0.17 | 0.46 | **0.96** | 0.62 | 0.75 | **0.72** | 0.12 | **0.12** | 0.09 | 0.29 |
| Qwen2.5-Omni-7B | 0.56 | 0.46 | 0.51 | 0.52 | 0.36 | 0.11 | 0.16 | 0.25 | 0.05 | 0.06 | 0.02 | 0.06 |
| Qwen3-Omni-30B | 0.50 | 0.01 | 0.03 | 0.46 | 0.31 | 0.07 | 0.11 | 0.25 | 0.10 | 0.10 | **0.10** | 0.24 |
| Lingshu-32B | 0.54 | **1.00** | **0.70** | 0.54 | 0.73 | 0.70 | 0.71 | 0.61 | 0.08 | 0.08 | 0.07 | 0.24 |

*Table 16.* Task 4 (Temporal boundary detection) Metrics on Whole Dataset (438 videos)

| Model | Arm Flexion | Arm Straightening | Arms Move Simultaneously | Asynchronous Movement |
|---|---|---|---|---|
| Intern3.5-VL-8B | 13.93 | 12.28 | 16.04 | 21.55 |
| Qwen2.5-VL-7B | 14.55 | 14.46 | 13.04 | 24.86 |
| Qwen3-VL-8B | 12.19 | 9.08 | 16.74 | 19.91 |
| Intern3.5-VL-38B | 24.99 | 19.89 | 19.59 | 22.88 |
| Qwen2.5-VL-32B | 11.68 | 10.19 | 8.93 | 17.67 |
| Qwen3-VL-32B | 13.88 | 13.48 | 11.83 | 20.45 |
| Qwen2.5-VL-72B | 11.09 | 9.20 | 7.64 | 15.84 |
| Qwen2.5-Omni-7B | 21.51 | 20.12 | 21.14 | 28.18 |
| Qwen3-Omni-30B | 20.69 | 18.36 | 27.36 | 22.97 |
| Lingshu-32B | 59.99 | 60.91 | 46.68 | 39.77 |

| | Blank Stare | Clonic | Close Eyes | Eye Blinking |
|---|---|---|---|---|
| Intern3.5-VL-8B | 28.21 | 19.32 | 26.32 | 22.08 |
| Qwen2.5-VL-7B | 28.12 | 21.13 | 27.01 | 25.38 |
| Qwen3-VL-8B | 27.86 | 22.14 | 27.09 | 23.44 |
| Intern3.5-VL-38B | 25.88 | 26.10 | 25.90 | 25.35 |
| Qwen2.5-VL-32B | 11.40 | 18.27 | 17.69 | 13.48 |
| Qwen3-VL-32B | 22.72 | 22.43 | 24.16 | 21.10 |
| Qwen2.5-VL-72B | 19.11 | 18.63 | 19.61 | 15.59 |
| Qwen2.5-Omni-7B | 29.61 | 29.75 | 28.40 | 29.84 |
| Qwen3-Omni-30B | 28.08 | 18.48 | 27.09 | 18.56 |
| Lingshu-32B | 40.04 | 32.04 | 39.86 | 38.33 |

| | Face Pulling | Face Twitching | Figure4 | Full Body Shaking |
|---|---|---|---|---|
| Intern3.5-VL-8B | 20.41 | 20.02 | 10.96 | 20.40 |
| Qwen2.5-VL-7B | 19.24 | 22.94 | 17.86 | 24.96 |
| Qwen3-VL-8B | 13.94 | 29.19 | 12.08 | 15.72 |
| Intern3.5-VL-38B | 21.73 | 22.50 | 20.76 | 19.75 |
| Qwen2.5-VL-32B | 10.16 | 13.15 | 12.63 | 10.16 |
| Qwen3-VL-32B | 14.83 | 28.41 | 15.56 | 16.00 |
| Qwen2.5-VL-72B | 11.20 | 18.96 | 12.43 | 13.00 |
| Qwen2.5-Omni-7B | 27.29 | 29.68 | 26.24 | 27.05 |
| Qwen3-Omni-30B | 22.21 | 22.72 | 13.69 | 20.69 |
| Lingshu-32B | 50.20 | 40.40 | 45.25 | 42.63 |

| | Head Turning | Limb Automatisms | Tonic | Oral Automatisms |
|---|---|---|---|---|
| Intern3.5-VL-8B | 20.02 | 18.51 | 14.34 | 20.76 |
| Qwen2.5-VL-7B | 21.34 | 26.15 | 21.90 | 27.51 |
| Qwen3-VL-8B | 15.72 | 19.53 | 13.68 | 21.42 |
| Intern3.5-VL-38B | 20.06 | 22.03 | 24.03 | 22.96 |
| Qwen2.5-VL-32B | 9.56 | 12.71 | 12.64 | 13.34 |
| Qwen3-VL-32B | 12.99 | 17.93 | 14.34 | 24.09 |
| Qwen2.5-VL-72B | 11.71 | 14.10 | 11.48 | 19.93 |
| Qwen2.5-Omni-7B | 27.95 | 29.58 | 29.65 | 29.36 |
| Qwen3-Omni-30B | 21.07 | 22.73 | 24.72 | 27.91 |
| Lingshu-32B | 57.21 | 35.69 | 33.57 | 42.51 |

| | Pelvic Thrusting | Seizure Start | Seizure End | |
|---|---|---|---|---|
| Intern3.5-VL-8B | 15.44 | 21.43 | 25.45 | |
| Qwen2.5-VL-7B | 23.19 | 21.08 | 15.41 | |
| Qwen3-VL-8B | 18.81 | 19.88 | 28.10 | |
| Intern3.5-VL-38B | 20.50 | 16.88 | 17.00 | |
| Qwen2.5-VL-32B | 9.10 | 10.14 | 11.89 | |
| Qwen3-VL-32B | 16.00 | 14.60 | 18.83 | |
| Qwen2.5-VL-72B | 12.38 | 12.32 | 18.63 | |
| Qwen2.5-Omni-7B | 29.38 | 27.48 | 19.53 | |
| Qwen3-Omni-30B | 20.00 | 32.83 | 52.47 | |
| Lingshu-32B | 43.12 | 43.76 | 63.31 | |

*Table 17.* Task 5 (Semiological Sequence Analysis) Metrics on Whole Dataset (438 videos)

| Model | Edit distance | Temporal f1 | LCS ratio |
|---|---|---|---|
| Intern3.5-VL-8B | 867.79 | **0.61** | 0.04 |
| Qwen2.5-VL-7B | 39.83 | 0.54 | 0.10 |
| Qwen3-VL-8B | 19.44 | 0.52 | 0.14 |
| Intern3.5-VL-38B | 13.91 | 0.51 | 0.16 |
| Qwen2.5-VL-32B | 543.53 | 0.55 | 0.05 |
| Qwen3-VL-32B | 17.51 | 0.51 | 0.15 |
| Qwen2.5-VL-72B | 17.36 | 0.50 | 0.16 |
| Qwen2.5-Omni-7B | 43.30 | 0.54 | 0.09 |
| Qwen3-Omni-30B | **4.21** | 0.13 | **0.24** |
| Lingshu-32B | 36.84 | 0.52 | 0.10 |

*Table 18.* Task 6 (Holistic Narrative Report Generation) Metrics on Whole Dataset (438 videos)

| Model | BLEU | ROUGE-1 F1 | ROUGE-L F1 | BERTScore F1 | RQI |
|---|---|---|---|---|---|
| InternVL3.5-8B | 0.54 | **0.20** | **0.13** | **0.56** | 40.76 |
| Qwen2.5-VL-7B | 1.26 | 0.19 | 0.12 | **0.56** | 38.63 |
| Qwen3-VL-8B | 0.39 | 0.16 | 0.11 | **0.56** | 41.12 |
| InternVL3.5-38B | 1.18 | **0.20** | **0.13** | 0.55 | 30.60 |
| Qwen2.5-VL-32B | 0.76 | 0.18 | 0.11 | 0.54 | 36.88 |
| Qwen3-VL-32B | 0.76 | 0.18 | 0.11 | 0.55 | 39.14 |
| Qwen2.5-VL-72B | 0.73 | 0.17 | 0.11 | 0.54 | 39.19 |
| Qwen2.5-Omni-7B | 0.35 | 0.14 | 0.10 | **0.56** | 39.99 |
| Qwen3-Omni-30B | **1.28** | 0.19 | 0.12 | **0.56** | **42.16** |
| Lingshu-32B | 0.82 | 0.19 | 0.12 | 0.55 | 39.48 |

*Table 19.* Task 1 (Seizure Semiology Recognition) Metrics on Test Dataset (82 videos)

| 2*Model | Occur During Sleep | | | | Head Turning | | | | Blank Stare | | | | Close Eyes | | | |
|---|---|---|---|---|---|---|---|---|---|---|---|---|---|---|---|---|
| | P | R | F1 | A | P | R | F1 | A | P | R | F1 | A | P | R | F1 | A |
| InternVL3_5-8B | 0.45 | 0.79 | 0.57 | 0.59 | 1.00 | 0.11 | 0.19 | 0.69 | 0.26 | 1.00 | 0.41 | 0.26 | 0.16 | 1.00 | 0.28 | 0.16 |
| Qwen2.5-VL-7B | 0.50 | 0.89 | 0.64 | 0.65 | 0.40 | 1.00 | 0.57 | 0.48 | 0.28 | 1.00 | 0.44 | 0.33 | 0.17 | 1.00 | 0.29 | 0.20 |
| Qwen3-VL-8B | 0.55 | 0.75 | 0.64 | 0.72 | 0.00 | 0.00 | 0.00 | 0.67 | 0.33 | 0.90 | 0.48 | 0.52 | 0.30 | 0.93 | 0.45 | 0.63 |
| InternVL3_5-38B | 0.55 | 0.86 | 0.67 | 0.70 | 0.00 | 0.00 | 0.00 | 0.65 | 0.26 | 0.95 | 0.40 | 0.27 | 0.16 | 1.00 | 0.28 | 0.16 |
| Qwen2.5-VL-32B | 0.58 | 0.50 | 0.54 | 0.70 | 0.60 | 0.21 | 0.32 | 0.68 | 0.28 | 0.81 | 0.42 | 0.42 | 0.22 | 0.85 | 0.35 | 0.49 |
| Qwen3-VL-32B | 0.55 | 0.82 | 0.66 | 0.72 | 0.56 | 0.50 | 0.53 | 0.71 | 0.29 | 0.86 | 0.43 | 0.44 | 0.27 | 1.00 | 0.43 | 0.57 |
| Qwen2.5-VL-72B | 0.57 | 0.82 | 0.68 | 0.73 | 0.45 | 0.79 | 0.57 | 0.59 | 0.34 | 0.86 | 0.49 | 0.53 | 0.23 | 0.92 | 0.36 | 0.48 |
| audio-flamingo-3 | - | - | - | - | - | - | - | - | - | - | - | - | - | - | - | - |
| Qwen2.5-Omni-7B | 0.57 | 0.93 | 0.70 | 0.64 | 0.46 | 0.54 | 0.50 | 0.55 | 0.37 | 1.00 | 0.54 | 0.38 | 0.16 | 1.00 | 0.27 | 0.16 |
| Qwen3-Omni-30B | 0.77 | 0.89 | 0.83 | 0.80 | 0.78 | 0.64 | 0.70 | 0.83 | 0.46 | 0.73 | 0.56 | 0.51 | 0.13 | 0.67 | 0.22 | 0.60 |
| Lingshu-32B | 0.57 | 0.75 | 0.65 | 0.72 | 0.46 | 0.82 | 0.59 | 0.60 | 0.27 | 0.95 | 0.42 | 0.32 | 0.19 | 1.00 | 0.31 | 0.30 |
| seizure_omni_sft | 0.50 | 0.93 | 0.65 | 0.65 | 0.42 | 0.96 | 0.59 | 0.53 | 0.30 | 0.90 | 0.45 | 0.42 | 0.18 | 1.00 | 0.30 | 0.25 |
| seizure_omni_grpo | 0.52 | 0.89 | 0.66 | 0.68 | 0.53 | 0.68 | 0.59 | 0.68 | 0.27 | 0.95 | 0.42 | 0.32 | 0.17 | 1.00 | 0.29 | 0.18 |

| 2*Model | Eye Blinking | | | | Face Pulling | | | | Face Twitching | | | | Tonic | | | |
|---|---|---|---|---|---|---|---|---|---|---|---|---|---|---|---|---|
| | P | R | F1 | A | P | R | F1 | A | P | R | F1 | A | P | R | F1 | A |
| InternVL3_5-8B | 0.00 | 0.00 | 0.00 | 0.74 | 0.50 | 0.12 | 0.19 | 0.68 | 0.20 | 1.00 | 0.33 | 0.20 | 0.59 | 0.51 | 0.55 | 0.53 |
| Qwen2.5-VL-7B | 0.29 | 0.86 | 0.43 | 0.42 | 0.27 | 0.62 | 0.37 | 0.33 | 0.19 | 0.94 | 0.32 | 0.22 | 0.25 | 0.04 | 0.08 | 0.40 |
| Qwen3-VL-8B | 0.00 | 0.00 | 0.00 | 0.76 | 0.00 | 0.00 | 0.00 | 0.70 | 0.20 | 1.00 | 0.33 | 0.24 | 0.62 | 0.47 | 0.53 | 0.57 |
| InternVL3_5-38B | 0.29 | 0.10 | 0.14 | 0.70 | 0.28 | 0.62 | 0.39 | 0.37 | 0.20 | 1.00 | 0.33 | 0.20 | 0.79 | 0.49 | 0.60 | 0.64 |
| Qwen2.5-VL-32B | 0.16 | 0.14 | 0.15 | 0.58 | 0.15 | 0.08 | 0.10 | 0.57 | 0.20 | 0.94 | 0.33 | 0.26 | 0.78 | 0.16 | 0.26 | 0.51 |
| Qwen3-VL-32B | 0.25 | 0.05 | 0.08 | 0.73 | 0.25 | 0.19 | 0.22 | 0.58 | 0.20 | 1.00 | 0.33 | 0.24 | 0.59 | 0.51 | 0.55 | 0.56 |
| Qwen2.5-VL-72B | 0.33 | 0.29 | 0.31 | 0.67 | 0.29 | 0.62 | 0.40 | 0.40 | 0.24 | 0.81 | 0.37 | 0.46 | 0.57 | 0.71 | 0.63 | 0.54 |
| audio-flamingo-3 | - | - | - | - | - | - | - | - | - | - | - | - | - | - | - | - |
| Qwen2.5-Omni-7B | 0.24 | 0.21 | 0.22 | 0.52 | 0.41 | 0.29 | 0.34 | 0.53 | 0.28 | 1.00 | 0.43 | 0.28 | 1.00 | 0.05 | 0.09 | 0.33 |
| Qwen3-Omni-30B | 0.00 | 0.00 | 0.00 | 0.54 | 0.18 | 0.23 | 0.20 | 0.31 | 0.29 | 1.00 | 0.44 | 0.29 | 0.71 | 0.48 | 0.57 | 0.49 |
| Lingshu-32B | 0.15 | 0.29 | 0.20 | 0.41 | 0.31 | 0.58 | 0.41 | 0.46 | 0.21 | 1.00 | 0.34 | 0.23 | 0.65 | 0.38 | 0.48 | 0.54 |
| seizure_omni_sft | 0.27 | 0.90 | 0.41 | 0.33 | 0.28 | 0.50 | 0.36 | 0.43 | 0.20 | 1.00 | 0.34 | 0.22 | 0.61 | 0.80 | 0.69 | 0.60 |
| seizure_omni_grpo | 0.26 | 0.43 | 0.32 | 0.54 | 0.28 | 0.31 | 0.29 | 0.52 | 0.20 | 1.00 | 0.33 | 0.20 | 0.90 | 0.20 | 0.33 | 0.55 |

| 2*Model | Clonic | | | | Arm Straightening | | | | Arm Flexion | | | | Figure 4 | | | |
|---|---|---|---|---|---|---|---|---|---|---|---|---|---|---|---|---|
| | P | R | F1 | A | P | R | F1 | A | P | R | F1 | A | P | R | F1 | A |
| InternVL3_5-8B | 0.43 | 0.23 | 0.30 | 0.65 | 0.64 | 0.82 | 0.72 | 0.64 | 0.65 | 0.96 | 0.78 | 0.64 | 0.31 | 0.24 | 0.27 | 0.73 |
| Qwen2.5-VL-7B | 0.39 | 0.50 | 0.44 | 0.59 | 0.56 | 1.00 | 0.72 | 0.57 | 0.65 | 1.00 | 0.79 | 0.65 | 0.23 | 0.29 | 0.26 | 0.64 |
| Qwen3-VL-8B | 0.43 | 0.12 | 0.18 | 0.69 | 0.72 | 0.73 | 0.73 | 0.71 | 0.77 | 0.68 | 0.72 | 0.67 | 0.50 | 0.06 | 0.11 | 0.80 |
| InternVL3_5-38B | 0.41 | 0.46 | 0.44 | 0.62 | 0.63 | 0.73 | 0.68 | 0.62 | 0.75 | 0.92 | 0.83 | 0.75 | 0.28 | 0.41 | 0.33 | 0.65 |
| Qwen2.5-VL-32B | 0.40 | 0.38 | 0.39 | 0.62 | 0.64 | 0.91 | 0.75 | 0.67 | 0.73 | 0.96 | 0.83 | 0.74 | 0.50 | 0.29 | 0.37 | 0.79 |
| Qwen3-VL-32B | 0.63 | 0.46 | 0.53 | 0.76 | 0.61 | 0.93 | 0.74 | 0.65 | 0.70 | 0.74 | 0.72 | 0.64 | 0.44 | 0.41 | 0.42 | 0.78 |
| Qwen2.5-VL-72B | 0.40 | 0.54 | 0.46 | 0.59 | 0.59 | 0.91 | 0.72 | 0.60 | 0.69 | 1.00 | 0.82 | 0.70 | 0.20 | 0.29 | 0.24 | 0.60 |
| audio-flamingo-3 | - | - | - | - | - | - | - | - | - | - | - | - | - | - | - | - |
| Qwen2.5-Omni-7B | 0.00 | 0.00 | 0.00 | 0.60 | 0.62 | 1.00 | 0.77 | 0.62 | 0.67 | 1.00 | 0.80 | 0.67 | 0.29 | 0.43 | 0.34 | 0.60 |
| Qwen3-Omni-30B | 0.30 | 1.00 | 0.47 | 0.34 | 0.69 | 1.00 | 0.81 | 0.71 | 0.75 | 0.96 | 0.84 | 0.74 | 0.28 | 0.56 | 0.37 | 0.51 |
| Lingshu-32B | 0.56 | 0.38 | 0.45 | 0.70 | 0.61 | 0.96 | 0.74 | 0.63 | 0.73 | 1.00 | 0.84 | 0.75 | 0.24 | 0.65 | 0.35 | 0.51 |
| seizure_omni_sft | 0.43 | 0.92 | 0.59 | 0.58 | 0.66 | 0.89 | 0.75 | 0.68 | 0.66 | 0.98 | 0.79 | 0.65 | 0.26 | 0.82 | 0.39 | 0.47 |
| seizure_omni_grpo | 0.70 | 0.27 | 0.39 | 0.73 | 0.56 | 1.00 | 0.71 | 0.56 | 0.65 | 1.00 | 0.79 | 0.65 | 0.26 | 0.65 | 0.37 | 0.55 |

| 2*Model | Oral Automatisms | | | | Limb Automatisms | | | | Asynchronous Movement | | | | Pelvic Thrusting | | | |
|---|---|---|---|---|---|---|---|---|---|---|---|---|---|---|---|---|
| | P | R | F1 | A | P | R | F1 | A | P | R | F1 | A | P | R | F1 | A |
| InternVL3_5-8B | 0.14 | 1.00 | 0.25 | 0.27 | 0.16 | 0.86 | 0.27 | 0.20 | 0.00 | 0.00 | 0.00 | 0.77 | 0.14 | 1.00 | 0.25 | 0.93 |
| Qwen2.5-VL-7B | 0.19 | 0.80 | 0.31 | 0.56 | 0.16 | 0.79 | 0.27 | 0.26 | 0.35 | 0.89 | 0.50 | 0.58 | 0.09 | 1.00 | 0.17 | 0.88 |
| Qwen3-VL-8B | 0.00 | 0.00 | 0.00 | 0.88 | 0.17 | 0.43 | 0.24 | 0.57 | 0.38 | 0.16 | 0.22 | 0.76 | 0.17 | 1.00 | 0.29 | 0.94 |
| InternVL3_5-38B | 0.18 | 0.60 | 0.27 | 0.60 | 0.18 | 0.86 | 0.30 | 0.30 | 0.32 | 0.53 | 0.40 | 0.63 | 0.07 | 1.00 | 0.12 | 0.83 |
| Qwen2.5-VL-32B | 0.18 | 0.70 | 0.29 | 0.58 | 0.18 | 0.86 | 0.30 | 0.31 | 0.50 | 0.53 | 0.51 | 0.77 | 0.07 | 1.00 | 0.13 | 0.84 |
| Qwen3-VL-32B | 0.14 | 0.50 | 0.22 | 0.59 | 0.15 | 0.64 | 0.24 | 0.33 | 0.35 | 0.79 | 0.48 | 0.63 | 0.08 | 1.00 | 0.15 | 0.87 |
| Qwen2.5-VL-72B | 0.19 | 0.70 | 0.30 | 0.59 | 0.16 | 0.86 | 0.28 | 0.22 | 0.31 | 0.79 | 0.44 | 0.53 | 0.04 | 1.00 | 0.08 | 0.70 |
| audio-flamingo-3 | - | - | - | - | - | - | - | - | - | - | - | - | - | - | - | - |
| Qwen2.5-Omni-7B | 0.22 | 1.00 | 0.36 | 0.38 | 0.17 | 0.31 | 0.22 | 0.52 | 0.07 | 0.75 | 0.13 | 0.31 | 0.00 | 0.00 | 0.00 | 0.74 |
| Qwen3-Omni-30B | 0.15 | 0.67 | 0.25 | 0.66 | 0.17 | 0.40 | 0.24 | 0.29 | 0.20 | 1.00 | 0.33 | 0.54 | 0.00 | 0.00 | 0.00 | 0.23 |
| Lingshu-32B | 0.18 | 0.80 | 0.30 | 0.53 | 0.19 | 0.79 | 0.30 | 0.37 | 0.26 | 0.42 | 0.32 | 0.58 | 0.06 | 1.00 | 0.12 | 0.81 |
| seizure_omni_sft | 0.14 | 1.00 | 0.24 | 0.23 | 0.16 | 0.21 | 0.18 | 0.67 | 0.32 | 0.95 | 0.47 | 0.51 | 0.05 | 1.00 | 0.10 | 0.78 |
| seizure_omni_grpo | 0.16 | 1.00 | 0.28 | 0.37 | 0.13 | 0.43 | 0.20 | 0.41 | 0.29 | 0.95 | 0.44 | 0.45 | 0.04 | 1.00 | 0.07 | 0.70 |

| 2*Model | Full Body Shaking | | | | Arms Move Simultaneously | | | | Verbal Responsiveness | | | | Ictal Vocalization | | | |
|---|---|---|---|---|---|---|---|---|---|---|---|---|---|---|---|---|
| | P | R | F1 | A | P | R | F1 | A | P | R | F1 | A | P | R | F1 | A |
| InternVL3_5-8B | 0.20 | 0.09 | 0.12 | 0.64 | 0.38 | 0.75 | 0.50 | 0.63 | - | - | - | - | - | - | - | - |
| Qwen2.5-VL-7B | 0.34 | 0.96 | 0.50 | 0.46 | 0.25 | 1.00 | 0.40 | 0.25 | - | - | - | - | - | - | - | - |
| Qwen3-VL-8B | 0.39 | 0.39 | 0.39 | 0.67 | 0.29 | 0.45 | 0.35 | 0.62 | - | - | - | - | - | - | - | - |
| InternVL3_5-38B | 0.37 | 0.70 | 0.48 | 0.58 | 0.25 | 0.95 | 0.40 | 0.30 | - | - | - | - | - | - | - | - |
| Qwen2.5-VL-32B | 0.44 | 0.35 | 0.39 | 0.69 | 0.38 | 0.80 | 0.52 | 0.63 | - | - | - | - | - | - | - | - |
| Qwen3-VL-32B | 0.37 | 0.61 | 0.46 | 0.62 | 0.27 | 0.75 | 0.40 | 0.48 | - | - | - | - | - | - | - | - |
| Qwen2.5-VL-72B | 0.35 | 0.83 | 0.49 | 0.51 | 0.28 | 0.90 | 0.42 | 0.40 | - | - | - | - | - | - | - | - |
| audio-flamingo-3 | - | - | - | - | - | - | - | - | 0.33 | 0.28 | 0.27 | 0.33 | 0.59 | 0.56 | 0.57 | 0.58 |
| Qwen2.5-Omni-7B | 0.15 | 1.00 | 0.26 | 0.22 | 0.17 | 1.00 | 0.29 | 0.17 | 0.87 | 0.60 | 0.59 | 0.75 | 0.66 | 0.94 | 0.78 | 0.69 |
| Qwen3-Omni-30B | 0.20 | 1.00 | 0.33 | 0.54 | 0.09 | 1.00 | 0.16 | 0.11 | 0.83 | 0.62 | 0.60 | 0.70 | 0.61 | 0.55 | 0.58 | 0.54 |
| Lingshu-32B | 0.35 | 0.57 | 0.43 | 0.58 | 0.24 | 0.90 | 0.38 | 0.28 | - | - | - | - | - | - | - | - |
| seizure_omni_sft | 0.33 | 1.00 | 0.49 | 0.42 | 0.25 | 1.00 | 0.40 | 0.25 | 0.90 | 0.57 | 0.57 | 0.80 | 0.62 | 0.79 | 0.69 | 0.64 |
| seizure_omni_grpo | 0.33 | 1.00 | 0.50 | 0.44 | 0.24 | 1.00 | 0.39 | 0.24 | 0.60 | 0.61 | 0.60 | 0.70 | 0.58 | 0.90 | 0.71 | 0.62 |

*Table 20.* Task 2 (Feature Justification) Metrics (R:ROUGE, B:BLEU, BS:BERTScore, S:Semiology Interpretation Score) on Test Dataset (82 videos)

| 2*Model | Occur During Sleep | | | | Head Turning | | | | Blank Stare | | | | Close Eyes | | | |
|---|---|---|---|---|---|---|---|---|---|---|---|---|---|---|---|---|
| | R | B | BS | S | R | B | BS | S | R | B | BS | S | R | B | BS | S |
| InternVL3.5-8B | 0.31 | 4.65 | 0.66 | 0.54 | 0.35 | 4.88 | 0.66 | 0.64 | 0.24 | 1.33 | 0.61 | 0.19 | 0.38 | 9.73 | 0.68 | 0.14 |
| Qwen2.5-VL-7B | 0.27 | 4.07 | 0.64 | 0.60 | 0.39 | 4.87 | 0.67 | 0.36 | 0.30 | 5.07 | 0.65 | 0.15 | 0.33 | 4.19 | 0.66 | 0.17 |
| Qwen3-VL-8B | 0.41 | 11.09 | 0.70 | 0.71 | 0.34 | 5.82 | 0.65 | 0.64 | 0.33 | 7.65 | 0.66 | 0.39 | 0.39 | 10.81 | 0.68 | 0.62 |
| InternVL3.5-38B | 0.32 | 5.69 | 0.65 | 0.63 | 0.33 | 3.73 | 0.63 | 0.62 | 0.25 | 1.49 | 0.61 | 0.22 | 0.38 | 5.19 | 0.68 | 0.14 |
| Qwen2.5-VL-32B | 0.28 | 4.63 | 0.63 | 0.60 | 0.35 | 5.88 | 0.64 | 0.63 | 0.26 | 3.06 | 0.62 | 0.34 | 0.38 | 10.60 | 0.68 | 0.48 |
| Qwen3-VL-32B | 0.36 | 8.42 | 0.67 | 0.67 | 0.35 | 5.14 | 0.65 | 0.64 | 0.34 | 7.72 | 0.66 | 0.39 | 0.38 | 8.60 | 0.67 | 0.55 |
| Qwen2.5-VL-72B | 0.28 | 3.31 | 0.64 | 0.68 | 0.37 | 5.67 | 0.66 | 0.48 | 0.28 | 3.95 | 0.65 | 0.39 | 0.39 | 10.70 | 0.67 | 0.46 |
| Audio-flamingo-3 | - | - | - | - | - | - | - | - | - | - | - | - | - | - | - | - |
| Qwen2.5-Omni-7B | 0.29 | 2.88 | 0.66 | 0.59 | 0.36 | 2.97 | 0.64 | 0.48 | 0.31 | 4.33 | 0.64 | 0.35 | 0.39 | 9.72 | 0.69 | 0.12 |
| Qwen3-Omni-30B | 0.34 | 5.98 | 0.68 | 0.80 | 0.38 | 6.59 | 0.66 | 0.77 | 0.35 | 8.07 | 0.66 | 0.41 | 0.42 | 10.39 | 0.70 | 0.60 |
| Lingshu-32B | 0.28 | 3.21 | 0.65 | 0.62 | 0.37 | 6.73 | 0.66 | 0.53 | 0.31 | 6.78 | 0.65 | 0.25 | 0.38 | 9.64 | 0.68 | 0.27 |
| seizure_omni_sft | 0.42 | 14.29 | 0.70 | 0.62 | 0.39 | 3.40 | 0.68 | 0.42 | 0.38 | 8.94 | 0.68 | 0.33 | 0.38 | 7.80 | 0.66 | 0.22 |
| seizure_omni_grpo | 0.31 | 4.86 | 0.66 | 0.67 | 0.37 | 3.32 | 0.65 | 0.59 | 0.30 | 4.92 | 0.65 | 0.26 | 0.40 | 11.70 | 0.68 | 0.16 |

| 2*Model | Eye Blinking | | | | Face Pulling | | | | Face Twitching | | | | Tonic | | | |
|---|---|---|---|---|---|---|---|---|---|---|---|---|---|---|---|---|
| | R | B | BS | S | R | B | BS | S | R | B | BS | S | R | B | BS | S |
| InternVL3.5-8B | 0.37 | 5.70 | 0.69 | 0.74 | 0.24 | 1.03 | 0.56 | 0.65 | 0.31 | 4.71 | 0.63 | 0.13 | 0.31 | 3.08 | 0.64 | 0.40 |
| Qwen2.5-VL-7B | 0.34 | 4.98 | 0.65 | 0.34 | 0.29 | 1.51 | 0.58 | 0.22 | 0.31 | 1.57 | 0.63 | 0.15 | 0.29 | 3.26 | 0.61 | 0.26 |
| Qwen3-VL-8B | 0.38 | 4.41 | 0.68 | 0.76 | 0.27 | 1.19 | 0.58 | 0.70 | 0.28 | 1.93 | 0.63 | 0.19 | 0.34 | 3.01 | 0.64 | 0.42 |
| InternVL3.5-38B | 0.34 | 2.93 | 0.64 | 0.69 | 0.27 | 0.96 | 0.57 | 0.25 | 0.29 | 3.05 | 0.62 | 0.12 | 0.30 | 2.83 | 0.63 | 0.43 |
| Qwen2.5-VL-32B | 0.33 | 3.96 | 0.66 | 0.58 | 0.27 | 1.63 | 0.58 | 0.54 | 0.27 | 1.13 | 0.61 | 0.17 | 0.29 | 3.01 | 0.63 | 0.30 |
| Qwen3-VL-32B | 0.43 | 7.50 | 0.70 | 0.73 | 0.26 | 1.19 | 0.58 | 0.56 | 0.29 | 2.95 | 0.62 | 0.18 | 0.32 | 4.15 | 0.64 | 0.41 |
| Qwen2.5-VL-72B | 0.40 | 9.47 | 0.70 | 0.66 | 0.29 | 1.78 | 0.59 | 0.29 | 0.29 | 4.13 | 0.64 | 0.40 | 0.33 | 3.59 | 0.63 | 0.37 |
| Audio-flamingo-3 | - | - | - | - | - | - | - | - | - | - | - | - | - | - | - | - |
| Qwen2.5-Omni-7B | 0.41 | 9.84 | 0.70 | 0.50 | 0.29 | 1.13 | 0.59 | 0.46 | 0.31 | 1.62 | 0.63 | 0.17 | 0.26 | 2.20 | 0.59 | 0.27 |
| Qwen3-Omni-30B | 0.41 | 9.01 | 0.69 | 0.54 | 0.30 | 3.78 | 0.59 | 0.27 | 0.31 | 2.67 | 0.65 | 0.23 | 0.33 | 4.43 | 0.64 | 0.34 |
| Lingshu-32B | 0.38 | 5.04 | 0.68 | 0.37 | 0.28 | 1.04 | 0.58 | 0.35 | 0.27 | 2.07 | 0.63 | 0.17 | 0.31 | 3.10 | 0.63 | 0.33 |
| seizure_omni_sft | 0.42 | 10.52 | 0.70 | 0.22 | 0.42 | 8.55 | 0.67 | 0.36 | 0.36 | 7.60 | 0.68 | 0.17 | 0.38 | 8.36 | 0.68 | 0.45 |
| seizure_omni_grpo | 0.41 | 10.54 | 0.70 | 0.46 | 0.33 | 2.78 | 0.61 | 0.48 | 0.32 | 3.86 | 0.64 | 0.14 | 0.31 | 4.90 | 0.63 | 0.35 |

| 2*Model | Clonic | | | | Arm Straightening | | | | Arm Flexion | | | | Figure 4 | | | |
|---|---|---|---|---|---|---|---|---|---|---|---|---|---|---|---|---|
| | R | B | BS | S | R | B | BS | S | R | B | BS | S | R | B | BS | S |
| InternVL3.5-8B | 0.35 | 3.66 | 0.66 | 0.56 | 0.33 | 3.20 | 0.64 | 0.54 | 0.31 | 1.97 | 0.63 | 0.51 | 0.40 | 5.48 | 0.67 | 0.61 |
| Qwen2.5-VL-7B | 0.33 | 3.16 | 0.65 | 0.49 | 0.32 | 2.70 | 0.63 | 0.46 | 0.30 | 3.17 | 0.62 | 0.53 | 0.41 | 5.70 | 0.67 | 0.46 |
| Qwen3-VL-8B | 0.36 | 4.34 | 0.66 | 0.61 | 0.34 | 3.88 | 0.64 | 0.65 | 0.29 | 2.73 | 0.61 | 0.60 | 0.42 | 6.11 | 0.67 | 0.67 |
| InternVL3.5-38B | 0.36 | 4.30 | 0.67 | 0.53 | 0.29 | 3.39 | 0.61 | 0.55 | 0.32 | 3.69 | 0.63 | 0.65 | 0.36 | 4.13 | 0.64 | 0.57 |
| Qwen2.5-VL-32B | 0.35 | 3.42 | 0.66 | 0.54 | 0.34 | 3.19 | 0.66 | 0.55 | 0.31 | 3.77 | 0.62 | 0.65 | 0.39 | 4.85 | 0.67 | 0.68 |
| Qwen3-VL-32B | 0.38 | 4.62 | 0.68 | 0.67 | 0.31 | 3.56 | 0.61 | 0.58 | 0.28 | 3.83 | 0.61 | 0.57 | 0.41 | 5.77 | 0.67 | 0.69 |
| Qwen2.5-VL-72B | 0.36 | 5.00 | 0.67 | 0.52 | 0.33 | 3.35 | 0.65 | 0.53 | 0.32 | 5.50 | 0.63 | 0.61 | 0.39 | 4.54 | 0.67 | 0.56 |
| Audio-flamingo-3 | - | - | - | - | - | - | - | - | - | - | - | - | - | - | - | - |
| Qwen2.5-Omni-7B | 0.32 | 2.31 | 0.65 | 0.52 | 0.31 | 0.94 | 0.62 | 0.48 | 0.32 | 3.20 | 0.64 | 0.62 | 0.38 | 4.73 | 0.65 | 0.47 |
| Qwen3-Omni-30B | 0.40 | 5.45 | 0.67 | 0.23 | 0.30 | 1.77 | 0.61 | 0.62 | 0.34 | 5.86 | 0.65 | 0.69 | 0.42 | 4.70 | 0.67 | 0.36 |
| Lingshu-32B | 0.33 | 3.82 | 0.65 | 0.63 | 0.33 | 3.34 | 0.64 | 0.51 | 0.31 | 2.88 | 0.62 | 0.64 | 0.41 | 6.59 | 0.68 | 0.43 |
| seizure_omni_sft | 0.43 | 10.10 | 0.71 | 0.47 | 0.39 | 6.45 | 0.68 | 0.59 | 0.36 | 6.60 | 0.66 | 0.58 | 0.42 | 9.85 | 0.69 | 0.34 |
| seizure_omni_grpo | 0.34 | 4.84 | 0.66 | 0.65 | 0.32 | 1.83 | 0.63 | 0.46 | 0.32 | 3.96 | 0.63 | 0.57 | 0.39 | 5.67 | 0.67 | 0.48 |

| 2*Model | Oral Automatisms | | | | Limb Automatisms | | | | Asynchronous Movement | | | | Pelvic Thrusting | | | |
|---|---|---|---|---|---|---|---|---|---|---|---|---|---|---|---|---|
| | R | B | BS | S | R | B | BS | S | R | B | BS | S | R | B | BS | S |
| InternVL3.5-8B | 0.37 | 10.69 | 0.65 | 0.27 | 0.33 | 3.31 | 0.62 | 0.15 | 0.30 | 1.53 | 0.63 | 0.63 | 0.32 | 3.54 | 0.64 | 0.86 |
| Qwen2.5-VL-7B | 0.32 | 6.51 | 0.64 | 0.56 | 0.33 | 2.25 | 0.62 | 0.21 | 0.27 | 1.45 | 0.61 | 0.41 | 0.32 | 3.39 | 0.63 | 0.81 |
| Qwen3-VL-8B | 0.38 | 11.22 | 0.68 | 0.87 | 0.34 | 4.02 | 0.63 | 0.54 | 0.31 | 1.74 | 0.63 | 0.61 | 0.33 | 4.32 | 0.65 | 0.90 |
| InternVL3.5-38B | 0.36 | 9.83 | 0.66 | 0.60 | 0.34 | 2.99 | 0.63 | 0.25 | 0.22 | 1.03 | 0.52 | 0.43 | 0.34 | 4.43 | 0.63 | 0.79 |
| Qwen2.5-VL-32B | 0.35 | 8.83 | 0.66 | 0.58 | 0.30 | 3.16 | 0.62 | 0.26 | 0.30 | 2.04 | 0.62 | 0.64 | 0.30 | 2.89 | 0.63 | 0.79 |
| Qwen3-VL-32B | 0.35 | 9.51 | 0.65 | 0.59 | 0.30 | 2.72 | 0.61 | 0.31 | 0.30 | 2.16 | 0.63 | 0.53 | 0.33 | 3.63 | 0.64 | 0.83 |
| Qwen2.5-VL-72B | 0.38 | 10.93 | 0.68 | 0.59 | 0.30 | 2.61 | 0.62 | 0.18 | 0.28 | 2.40 | 0.61 | 0.47 | 0.34 | 3.59 | 0.65 | 0.67 |
| Audio-flamingo-3 | - | - | - | - | - | - | - | - | - | - | - | - | - | - | - | - |
| Qwen2.5-Omni-7B | 0.35 | 10.41 | 0.66 | 0.38 | 0.35 | 6.02 | 0.64 | 0.50 | 0.28 | 0.63 | 0.61 | 0.14 | 0.33 | 3.19 | 0.63 | 0.74 |
| Qwen3-Omni-30B | 0.38 | 7.44 | 0.68 | 0.63 | 0.31 | 1.85 | 0.62 | 0.25 | 0.30 | 1.86 | 0.63 | 0.47 | 0.32 | 2.77 | 0.63 | 0.23 |
| Lingshu-32B | 0.38 | 11.61 | 0.66 | 0.53 | 0.32 | 3.70 | 0.63 | 0.32 | 0.30 | 1.81 | 0.63 | 0.49 | 0.32 | 4.37 | 0.63 | 0.77 |
| seizure_omni_sft | 0.43 | 10.35 | 0.70 | 0.23 | 0.45 | 11.51 | 0.71 | 0.64 | 0.33 | 3.25 | 0.68 | 0.45 | 0.37 | 6.74 | 0.69 | 0.74 |
| seizure_omni_grpo | 0.38 | 7.99 | 0.66 | 0.37 | 0.36 | 4.57 | 0.65 | 0.37 | 0.30 | 0.97 | 0.63 | 0.35 | 0.34 | 5.16 | 0.65 | 0.63 |

| 2*Model | Full Body Shaking | | | | Arms Move Simultaneously | | | | Verbal Responsiveness | | | | Ictal Vocalization | | | |
|---|---|---|---|---|---|---|---|---|---|---|---|---|---|---|---|---|
| | R | B | BS | S | R | B | BS | S | R | B | BS | S | R | B | BS | S |
| InternVL3.5-8B | 0.34 | 5.58 | 0.64 | 0.62 | 0.36 | 5.39 | 0.65 | 0.58 | - | - | - | - | - | - | - | - |
| Qwen2.5-VL-7B | 0.34 | 7.24 | 0.64 | 0.43 | 0.30 | 2.63 | 0.61 | 0.20 | - | - | - | - | - | - | - | - |
| Qwen3-VL-8B | 0.37 | 8.04 | 0.65 | 0.65 | 0.39 | 5.65 | 0.66 | 0.58 | - | - | - | - | - | - | - | - |
| InternVL3.5-38B | 0.37 | 9.16 | 0.65 | 0.57 | 0.23 | 0.42 | 0.54 | 0.12 | - | - | - | - | - | - | - | - |
| Qwen2.5-VL-32B | 0.32 | 5.45 | 0.64 | 0.67 | 0.39 | 7.23 | 0.67 | 0.58 | - | - | - | - | - | - | - | - |
| Qwen3-VL-32B | 0.36 | 8.69 | 0.65 | 0.60 | 0.35 | 4.24 | 0.65 | 0.43 | - | - | - | - | - | - | - | - |
| Qwen2.5-VL-72B | 0.36 | 9.24 | 0.66 | 0.49 | 0.36 | 5.17 | 0.66 | 0.35 | - | - | - | - | - | - | - | - |
| Audio-flamingo-3 | - | - | - | - | - | - | - | - | 0.16 | 0.23 | 0.57 | 0.65 | 0.26 | 6.83 | 0.63 | 0.50 |
| Qwen2.5-Omni-7B | 0.37 | 8.95 | 0.66 | 0.22 | 0.40 | 5.08 | 0.68 | 0.17 | 0.24 | 2.16 | 0.61 | 0.53 | 0.30 | 5.27 | 0.65 | 0.60 |
| Qwen3-Omni-30B | 0.38 | 7.75 | 0.66 | 0.54 | 0.39 | 3.46 | 0.67 | 0.11 | 0.28 | 2.08 | 0.62 | 0.68 | 0.24 | 2.22 | 0.61 | 0.48 |
| Lingshu-32B | 0.34 | 8.53 | 0.66 | 0.58 | 0.35 | 2.13 | 0.66 | 0.23 | - | - | - | - | - | - | - | - |
| seizure_omni_sft | 0.43 | 11.63 | 0.69 | 0.40 | 0.42 | 6.57 | 0.69 | 0.20 | 0.34 | 3.36 | 0.64 | 0.35 | 0.37 | 10.20 | 0.68 | 0.58 |
| seizure_omni_grpo | 0.38 | 9.89 | 0.67 | 0.41 | 0.40 | 5.41 | 0.67 | 0.20 | 0.27 | 2.21 | 0.61 | 0.39 | 0.32 | 5.92 | 0.65 | 0.57 |

*Table 21.* Task 3 (Spatial and anatomical analysis) Metrics on Test Dataset (82 videos)

| Model | Head turning | | | | Arm movement | | | | Onset body part | | | |
|---|---|---|---|---|---|---|---|---|---|---|---|---|
| | P | R | F1 | A | P | R | F1 | A | P | R | F1 | A |
| InternVL3.5-8B | 0.22 | **1.00** | 0.36 | 0.22 | 0.88 | 0.88 | 0.88 | 0.78 | 0.09 | 0.12 | 0.08 | 0.14 |
| Qwen2.5-VL-7B | 0.23 | **1.00** | 0.37 | 0.25 | 0.60 | 0.12 | 0.21 | 0.15 | 0.19 | **0.28** | 0.21 | 0.28 |
| Qwen3-VL-8B | 0.14 | 0.14 | 0.14 | 0.62 | 0.50 | 0.04 | 0.08 | 0.11 | 0.08 | 0.09 | 0.06 | 0.16 |
| InternVL3.5-38B | 0.22 | **1.00** | 0.36 | 0.22 | 0.88 | **0.92** | **0.90** | **0.81** | 0.02 | 0.08 | 0.02 | 0.07 |
| Qwen2.5-VL-32B | **0.25** | 0.43 | 0.32 | 0.59 | 0.78 | 0.29 | 0.42 | 0.30 | **0.34** | **0.28** | **0.25** | 0.37 |
| Qwen3-VL-32B | 0.24 | 0.86 | **0.38** | 0.38 | 0.87 | 0.54 | 0.67 | 0.52 | 0.18 | 0.26 | 0.15 | 0.20 |
| Qwen2.5-VL-72B | 0.17 | 0.14 | 0.15 | 0.66 | **1.00** | 0.71 | 0.83 | 0.74 | 0.23 | 0.21 | 0.18 | 0.36 |
| Qwen2.5-Omni-7B | 0.15 | 0.29 | 0.20 | 0.50 | 0.50 | 0.08 | 0.14 | 0.11 | 0.10 | 0.14 | 0.05 | 0.08 |
| Qwen3-Omni-30B | 0.00 | 0.00 | 0.00 | 0.75 | 0.50 | 0.08 | 0.14 | 0.11 | 0.15 | 0.18 | 0.14 | 0.22 |
| Lingshu-32B | 0.22 | **1.00** | 0.36 | 0.22 | 0.94 | 0.67 | 0.78 | 0.67 | 0.15 | 0.19 | 0.17 | 0.40 |
| seizure_omni_sft | 0.00 | 0.00 | 0.00 | **0.78** | 0.00 | 0.00 | 0.00 | 0.11 | 0.08 | 0.14 | 0.10 | **0.43** |
| seizure_omni_grpo | 0.00 | 0.00 | 0.00 | **0.78** | 0.00 | 0.00 | 0.00 | 0.11 | 0.18 | 0.14 | 0.11 | **0.43** |

*Table 22.* Task 4 (Temporal boundary detection) Metrics on Test Dataset (82 videos)

| Model | Arm Flexion | Arm Straightening | Arms Move Simultaneously | Asynchronous Movement |
|---|---|---|---|---|
| Intern3.5-VL-8B | 13.63 | 13.80 | 15.21 | 17.24 |
| Qwen2.5-VL-7B | 13.10 | 12.48 | 8.63 | 15.11 |
| Qwen3-VL-8B | 12.13 | 7.55 | 9.53 | 10.16 |
| Intern3.5-VL-38B | 24.54 | 20.43 | 20.32 | 21.53 |
| Qwen2.5-VL-32B | 12.67 | 10.77 | 9.00 | 7.89 |
| Qwen3-VL-32B | 12.10 | 11.93 | 5.74 | 9.35 |
| Qwen2.5-VL-72B | 10.40 | 9.25 | 7.47 | 7.63 |
| Qwen2.5-Omni-7B | 19.08 | 16.09 | 13.00 | 24.63 |
| Qwen3-Omni-30B | 18.21 | 15.36 | 24.11 | 18.58 |
| Lingshu-32B | 61.96 | 58.98 | 61.21 | 53.41 |
| seizure_omni_sft | 17.42 | 14.52 | 12.42 | 19.79 |
| seizure_omni_grpo | 14.46 | 12.48 | 7.89 | 15.79 |

| | Blank Stare | Clonic | Close Eyes | Eye Blinking |
|---|---|---|---|---|
| Intern3.5-VL-8B | 28.90 | 22.04 | 24.69 | 20.52 |
| Qwen2.5-VL-7B | 28.33 | 23.42 | 26.85 | 22.38 |
| Qwen3-VL-8B | 30.00 | 23.65 | 26.38 | 24.19 |
| Intern3.5-VL-38B | 26.74 | 27.30 | 28.08 | 25.85 |
| Qwen2.5-VL-32B | 11.67 | 22.96 | 18.08 | 14.00 |
| Qwen3-VL-32B | 23.48 | 23.54 | 25.62 | 17.05 |
| Qwen2.5-VL-72B | 16.19 | 21.77 | 20.46 | 13.67 |
| Qwen2.5-Omni-7B | 30.00 | 30.00 | 27.15 | 30.52 |
| Qwen3-Omni-30B | 30.81 | 16.54 | 28.69 | 12.52 |
| Lingshu-32B | 33.86 | 28.16 | 34.85 | 42.95 |
| seizure_omni_sft | 27.81 | 29.62 | 27.15 | 30.52 |
| seizure_omni_grpo | 25.52 | 28.31 | 25.92 | 30.52 |

| | Face Pulling | Face Twitching | Figure4 | Full Body Shaking |
|---|---|---|---|---|
| Intern3.5-VL-8B | 19.16 | 20.19 | 8.69 | 16.13 |
| Qwen2.5-VL-7B | 18.58 | 19.75 | 16.00 | 15.70 |
| Qwen3-VL-8B | 8.65 | 28.44 | 10.38 | 6.22 |
| Intern3.5-VL-38B | 21.76 | 23.57 | 22.38 | 20.05 |
| Qwen2.5-VL-32B | 10.73 | 11.62 | 11.94 | 9.48 |
| Qwen3-VL-32B | 12.38 | 28.56 | 16.19 | 6.23 |
| Qwen2.5-VL-72B | 10.04 | 17.81 | 13.19 | 8.35 |
| Qwen2.5-Omni-7B | 26.32 | 30.31 | 25.62 | 20.52 |
| Qwen3-Omni-30B | 23.73 | 23.19 | 8.12 | 18.78 |
| Lingshu-32B | 52.19 | 44.25 | 50.19 | 44.30 |
| seizure_omni_sft | 13.58 | 30.31 | 25.75 | 14.43 |
| seizure_omni_grpo | 8.85 | 30.31 | 17.31 | 12.78 |

| | Head Turning | Limb Automatisms | Tonic | Oral Automatisms |
|---|---|---|---|---|
| Intern3.5-VL-8B | 23.04 | 17.31 | 12.07 | 16.40 |
| Qwen2.5-VL-7B | 24.04 | 27.92 | 22.84 | 27.00 |
| Qwen3-VL-8B | 15.75 | 18.38 | 10.96 | 18.80 |
| Intern3.5-VL-38B | 19.92 | 22.69 | 23.72 | 21.62 |
| Qwen2.5-VL-32B | 9.75 | 11.15 | 14.49 | 13.20 |
| Qwen3-VL-32B | 15.65 | 13.85 | 12.00 | 23.00 |
| Qwen2.5-VL-72B | 11.86 | 12.31 | 11.44 | 18.20 |
| Qwen2.5-Omni-7B | 25.57 | 30.54 | 29.76 | 30.00 |
| Qwen3-Omni-30B | 21.29 | 25.69 | 23.13 | 37.70 |
| Lingshu-32B | 57.07 | 44.77 | 37.67 | 47.30 |
| seizure_omni_sft | 23.68 | 30.54 | 18.40 | 30.00 |
| seizure_omni_grpo | 19.86 | 24.31 | 13.51 | 27.40 |

| | Pelvic Thrusting | Seizure Start | Seizure End |
|---|---|---|---|
| Intern3.5-VL-8B | 5.00 | 19.14 | 24.33 |
| Qwen2.5-VL-7B | 5.00 | 19.57 | 14.69 |
| Qwen3-VL-8B | 2.00 | 16.17 | 29.06 |
| Intern3.5-VL-38B | 20.00 | 18.76 | 19.14 |
| Qwen2.5-VL-32B | 1.00 | 8.19 | 12.72 |
| Qwen3-VL-32B | 1.00 | 12.95 | 17.11 |
| Qwen2.5-VL-72B | 9.00 | 10.27 | 17.93 |
| Qwen2.5-Omni-7B | 30.00 | 25.93 | 19.43 |
| Qwen3-Omni-30B | 0.00 | 36.21 | 45.37 |
| Lingshu-32B | 78.00 | 50.48 | 64.47 |
| seizure_omni_sft | 30.00 | 19.05 | 22.33 |
| seizure_omni_grpo | 30.00 | 15.66 | 19.45 |

*Table 23.* Task 5 (Semiological Sequence Analysis) Metrics on Test Dataset (82 videos)

| Model | Edit distance | Temporal f1 | LCS ratio |
|---|---|---|---|
| Intern3.5-VL-8B | 874.66 | **0.57** | 0.02 |
| Qwen2.5-VL-7B | 34.72 | 0.51 | 0.12 |
| Qwen3-VL-8B | 18.86 | 0.45 | 0.19 |
| Intern3.5-VL-38B | 12.06 | 0.43 | 0.22 |
| Qwen2.5-VL-32B | 445.09 | 0.48 | 0.07 |
| Qwen3-VL-32B | 14.96 | 0.44 | 0.19 |
| Qwen2.5-VL-72B | 14.47 | 0.42 | 0.22 |
| Qwen2.5-Omni-7B | 36.33 | 0.45 | 0.12 |
| Qwen3-Omni-30B | **5.15** | 0.18 | **0.43** |
| Lingshu-32B | 29.92 | 0.46 | 0.15 |
| seizure_omni_sft | 7.47 | 0.32 | 0.18 |
| seizure_omni_grpo | 12.24 | 0.32 | 0.18 |

*Table 24.* Task 6 (Holistic Narrative Report Generation) Metrics on Test Dataset (82 videos)

| Model | BLEU | ROUGE-1 F1 | ROUGE-L F1 | BERTScore F1 | RQI |
|---|---|---|---|---|---|
| InternVL3_5-8B | 0.57 | 0.19 | 0.13 | 0.56 | 38.43 |
| Qwen2.5-VL-7B | **1.32** | 0.19 | 0.12 | 0.56 | 36.94 |
| Qwen3-VL-8B | 0.18 | 0.16 | 0.11 | 0.56 | 38.20 |
| InternVL3_5-38B | 1.53 | 0.19 | 0.13 | 0.55 | 36.14 |
| Qwen2.5-VL-32B | 0.97 | 0.18 | 0.12 | 0.54 | 37.48 |
| Qwen3-VL-32B | 0.99 | 0.18 | 0.12 | 0.55 | 39.13 |
| Qwen2.5-VL-72B | 0.77 | 0.17 | 0.11 | 0.55 | 37.37 |
| Qwen2.5-Omni-7B | 0.14 | 0.15 | 0.10 | 0.56 | 35.91 |
| Qwen3-Omni-30B | 1.04 | 0.18 | 0.12 | 0.56 | 37.52 |
| Lingshu-32B | 1.10 | 0.19 | 0.12 | 0.55 | **39.80** |
| seizure_omni_sft | 0.62 | 0.18 | 0.13 | 0.56 | 31.69 |
| seizure_omni_grpo | 0.95 | **0.21** | **0.15** | **0.58** | 36.44 |

*Table 25.* LLM sensitivity comparisons for Task 2 and Task 6. This table compares three LLMs' performance for the structured extraction and evaluation pipeline. Qwen-Plus, Kimi-K2.5, and GPT-5.4 produce nearly identical Task 2 results and similar Task 6 report-quality scores, indicating metrics are not sensitive to the specific LLM used.

| Judge | Task 2 Score | Task 6 RQI |
|---|---|---|
| **Qwen3-Plus** | **0.62** | **39.80** |
| Kimi-K2.5 | 0.61 | 38.70 |
| GPT-5.4 | 0.61 | 42.26 |

*Table 26.* **Matched clinician-versus-MLLM comparison for selected Task 1 semiology features under the same 2 FPS input constraint.** To control for preprocessing and temporal sampling, both clinicians and the best performing MLLM were evaluated using the same downsampled 2 FPS input. F1 score is reported for six representative Task 1 features spanning prominent motor signs and more subtle visual cues. Clinicians outperform the MLLM on all six features, including tonic, clonic, arm flexion, and face twitching. This result indicates that the remaining performance gap cannot be explained solely by the 2 FPS preprocessing constraint, but instead reflects a genuine limitation of the MLLM in fine-grained semiology recognition. Notably, eye blinking shows relatively low F1 for both clinicians and the MLLM, suggesting that this feature is intrinsically difficult to recognize under strong temporal downsampling because it evolves very rapidly.

| Feature | Clinician F1 at 2 FPS | Best MLLM F1 at 2 FPS |
|---|---|---|
| Tonic | 0.87 | 0.69 |
| Clonic | 0.84 | 0.59 |
| Arm flexion | 0.98 | 0.83 |
| Fig 4 | 0.78 | 0.42 |
| Eye blinking | 0.52 | 0.43 |
| Face twitching | 0.83 | 0.44 |

*Table 27.* **Clinician-versus-MLLM comparison for Task 7 diagnosis under the same 2 FPS input constraint.** To control for preprocessing and temporal sampling, both clinicians and the best performing MLLM were evaluated using the same downsampled 2 FPS input. F1 score is reported for the diagnostic classification task (Task 7). In this matched setting, the best-performing MLLM exceeds the clinician baseline, achieving 0.96 versus 0.91. This contrast with the lower-level semiology tasks suggests that diagnostic classification is less bottlenecked by the 2 FPS sampling pipeline and may depend more on higher-level aggregation of multiple cues than on fine-grained frame-by-frame perception.

| Task | Clinician F1 at 2 FPS | Best MLLM F1 at 2 FPS |
|---|---|---|
| 7 | 0.91 | 0.96 |

*Table 28.* **Clinician-versus-MLLM comparison for two Task 3 subtasks under the same 2 FPS input constraint.** To control for preprocessing and temporal sampling, both clinicians and the best performing MLLM were evaluated using the same downsampled 2 FPS input. F1 score is reported for two Task 3 subtasks that require localization and directional interpretation of semiological signs. Clinicians outperform the best MLLM on both subtasks, with a particularly large gap for head turning. These results indicate that current MLLMs remain substantially weaker than clinicians on spatially grounded semiology reasoning, especially for directional judgments such as left-right head turning, even when both are evaluated under the same benchmarked input constraint.

| Task 3 Subtype | Clinician F1 at 2 FPS | Best MLLM F1 at 2 FPS |
|---|---|---|
| Head turning | 0.89 | 0.38 |
| Body region onset | 0.54 | 0.25 |

*Table 29.* Frames sample ablation experiment for Task 1 features. This table measures how recognition performance changes when the input sampling rate increases from the 2 FPS benchmark setting to 4 FPS and 10 FPS; the reported values are feature-level F1 scores, and the gains relative to the 2 FPS baseline are shown in parentheses. Denser sampling helps the most for rapid or subtle phenomena such as clonic motion and eye blinking, but the magnitude of the gains is moderate.

| Feature | 2 FPS F1 | 4 FPS F1 | 10 FPS F1 |
|---|---|---|---|
| Tonic | 0.55 | 0.61 (+0.06) | 0.62 (+0.07) |
| Clonic | 0.53 | 0.64 (+0.11) | 0.66 (+0.13) |
| Eye blinking | 0.08 | 0.15 (+0.07) | 0.19 (+0.11) |
| Face twitching | 0.33 | 0.33 (+0.00) | 0.34 (+0.01) |

*Table 30.* Task 4 full-video timing localization on the held-out test set for Qwen3-VL-8B. Switching from centered clips to the more realistic full-video, MAEs for the three features all increase, with the largest degradation observed for clonic onset, supporting that full-video temporal grounding is substantially harder.

| Feature | Centered Clip MAE | Full Video MAE | Delta MAE |
|---|---|---|---|
| tonic | 10.96 | 12.07 | 1.11 |
| clonic | 23.65 | 35.04 | 11.39 |
| eye blinking | 24.19 | 26.43 | 2.24 |

*Table 31.* **Feature wise Cohen's kappa for annotation reliability on the independently labeled 75 video subset.** A total of 151 videos were annotated by epileptologists. Of these, 76 videos were used in an initial training phase, during which clinicians annotated jointly with the student annotators and explained the definition of each semiological feature. The remaining 75 videos were then labeled independently by both groups and used to compute feature-wise Cohen's kappa. Agreement is strong for most features, with perfect agreement for several clinically important labels (e.g., tonic, clonic, verbal responsiveness, and sleep occurrence). Lower agreement is observed for more subtle or rapidly evolving signs, such as blank stare, close eyes, and rapid eye blinking, indicating that these features are intrinsically more difficult to annotate consistently even after training.

| Feature | Kappa Score |
|---|---|
| Head turning | 0.84 |
| Tonic | 1.00 |
| Rapid eye blinking | 0.64 |
| Occur during sleep | 1.00 |
| Arm flexion | 0.89 |
| Arm straightening | 0.81 |
| Limb automatisms | 0.86 |
| Full body shaking | 0.85 |
| Asynchronous movement | 0.76 |
| Ictal vocalization | 0.76 |
| Verbal responsiveness | 1.00 |
| Clonic | 1.00 |
| Fig 4 | 1.00 |
| Pelvic thrusting | 0.89 |
| Face pulling | 0.94 |
| Face twitching | 0.81 |
| Arms move simultaneously | 0.75 |
| Blank stare | 0.53 |
| Close eyes | 0.60 |
| Oral automatisms | 0.86 |

*Table 32.* **Five-fold cross-validation F1 scores for ES-versus-NES classification using the two-stage pipeline.** Four standard classifiers were trained on the same structured semiology representation produced by the first stage of the pipeline, and F1 score is reported for each fold together with the mean and variance across folds. Random Forest achieves the highest mean F1, with SVM performing comparably, while KNN and Logistic Regression are lower. Overall, the relatively small gap between the top-performing classifiers suggests that the structured semiology representation is the primary source of predictive gain, whereas the choice of downstream classifier plays a secondary role.

| Model | Fold 1 | Fold 2 | Fold 3 | Fold 4 | Fold 5 | Mean | Variance |
|---|---|---|---|---|---|---|---|
| Random Forest | 0.96 | 0.97 | 0.947 | 0.93 | 0.94 | 0.9494 | 0.000214 |
| KNN | 0.887 | 0.89 | 0.856 | 0.849 | 0.86 | 0.8684 | 0.000286 |
| SVM | 0.951 | 0.958 | 0.939 | 0.922 | 0.93 | 0.9400 | 0.000192 |
| Logistic Regression | 0.90 | 0.914 | 0.885 | 0.877 | 0.89 | 0.8932 | 0.000166 |

*Table 33.* **Random Forest feature importances for ES-versus-NES classification.** Importance scores are derived from the trained Random Forest model using the standard mean decrease in impurity criterion, i.e., the total reduction in node impurity attributable to each feature, averaged over all trees in the ensemble and normalized so that the scores sum to one. Higher values therefore indicate features that contribute more strongly to the model's decision-making for distinguishing epileptic seizures (ES) from nonepileptic seizures (NES). Head turning and tonic features emerge as the most influential predictors, followed by rapid eye blinking and occurrence during sleep. Overall, the importance distribution suggests that the classifier relies on a small set of highly discriminative semiological cues, while many other clinically relevant features provide complementary but smaller contributions.

| Feature | Importance Score |
| --- | --- |
| Head turning | 0.194 |
| Tonic | 0.130 |
| Rapid eye blinking | 0.100 |
| Occur during sleep | 0.080 |
| Arm flexion | 0.070 |
| Arm straightening | 0.060 |
| Limb automatisms | 0.053 |
| Full body shaking | 0.050 |
| Asynchronous movement | 0.039 |
| Ictal vocalization | 0.037 |
| Verbal responsiveness | 0.037 |
| Clonic | 0.030 |
| Fig 4 | 0.023 |
| Pelvic thrusting | 0.020 |
| Face pulling | 0.020 |
| Face twitching | 0.020 |
| Arms move simultaneously | 0.020 |
| Blank stare | 0.010 |
| Close eyes | 0.004 |
| Oral automatisms | 0.003 |

*Table 34.* **Representative examples of ground-truth semiology reports and Qwen 2.5 VLM 72B generated reports.** Each row shows one seizure video together with the clinician written semiology report and the corresponding free-text report generated by Qwen 2.5 VLM 72B. The selected examples illustrate cases in which the model captures the overall seizure structure and many of the major semiological elements such as staring, head turning/version, asymmetric posturing, automatisms, and progression to bilateral tonic-clonic activity while still differing from the ground truth in finer details such as laterality, specific limb configuration, or the presence of subtle facial signs. Overall, these examples show that the MLLM often produces high-level summaries, but residual errors remain in fine-grained and spatially specific semiology description. This qualitative comparison complements the quantitative Task 6 evaluation by illustrating both the strengths and the limitations of report generation at the case level.

| File name | Ground truth report | Qwen 2.5 VLM 72B report |
| --- | --- | --- |
| S0014 | The patient arouses from sleep and is seen with left facial pulling and left arm jerking, followed by progression to bilateral tonic-clonic activity. | The patient has a blank stare, head version, asymmetric arm posturing, and then bilateral tonic-clonic activity with unresponsiveness. |
| R0001 | The patient exhibits staring, lip smacking, giggling, head version, fencer posture, and then bilateral tonic-clonic movements. | The patient shows a blank stare, lip smacking/automatisms, head version, asymmetric arm posturing, and then bilateral tonic-clonic movements. |
| Y0005 | The patient is initially drinking water, then develops an ictal cry, left head turn, figure-of-4 posture, and bilateral tonic-clonic movements. | The patient has a blank stare, left head turn, left arm flexion/asymmetric posturing, followed by bilateral clonic/tonic-clonic activity. |
| S0009 | The patient awakens from sleep, vocalizes, develops right facial pulling, rightward head version, tonic posturing, and generalized tonic-clonic movements. | The patient shows staring, head turning, asymmetric tonic posturing, and progression to bilateral convulsive activity. |

