# OpenReview forum: "Seizure-Semiology-Suite($S^3$): A Clinically Multimodal Dataset, Benchmark, and Models for Seizure Semiology Understanding"
_ICML.cc/2026/Conference — ICML 2026 spotlight_

### Official Review · Reviewer_uKma · 2026-03-11

**Soundness:** 2
**Presentation:** 1
**Significance:** 3
**Originality:** 4
**Overall Recommendation:** 4
**Confidence:** 3

**Summary:**

This paper introduces a benchmark for seizure semiology understanding from videos. The authors construct expert-informed annotations and organize the problem into seven clinically relevant tasks with corresponding evaluation metrics. To better assess clinical usefulness, the paper further proposes the Seizure Report Quality Index (Seizure-RQI). In addition, the authors evaluate 11 open-weight multimodal large language models as baselines and compare them with seizure-specialized fine-tuned models.

**Compliance With Llm Reviewing Policy:**

Affirmed.

**Final Justification:**

While the architectural novelty is limited, I recognize the originality of this work as the first effective automated assessment tool for seizure semiology reports in this domain. The authors have adequately addressed my concerns regarding RQI formulation, baseline performance, and sampling constraints. I raise my score to weak accept.

**Key Questions For Authors:**

- Please provide a formal and reproducible definition of the Report Quality Index, including how S, C, L, and T are computed.
- Please analyze why SeizureMLLM underperforms the baseline on some tasks despite task-specific fine-tuning.

**Limitations:**

yes

**Strengths And Weaknesses:**

### Strenghts
- The paper addresses an important and clinically meaningful problem, namely seizure semiology understanding from videos.
- The dataset and annotations appear to be constructed with expert knowledge, which increases the clinical relevance of the benchmark.
- The benchmark is decomposed into seven tasks, enabling a more fine-grained evaluation of model capabilities.
- The proposed Seizure-RQI is a meaningful attempt to evaluate report quality from a clinical perspective.
- The experimental evaluation is extensive, including 11 open-weight MLLMs as baselines as well as seizure-specialized fine-tuned models.

### Weaknesses
- The formulation of the Report Quality Index in Section 3.2.6 is not sufficiently formal. It is difficult to understand how S, C, L, and T are exactly defined and computed, which makes the metric hard to interpret and reproduce.
- According to Table 1, SeizureMLLM does not consistently outperform the baseline, especially on arm movement and Tasks 4 and 5.
- In Section 3.3.2, the implementation details indicate that Tasks 1, 2, 5, and 6 use videos sampled at 2 FPS. This appears too sparse to adequately capture temporally continuous seizure dynamics.
- The main contribution of the paper is largely limited to the dataset and benchmark, while the technical contribution on the modeling side appears limited.

---

> ### Author Rebuttal · Authors · 2026-03-31
>
> All tables referred in this response can be found at this anonymous link: https://anonymous.4open.science/r/ICML-FF68/
>
> Thanks for your feedback. We are happy to provide more details beyond the following.
>
> [Q1,W1]:We appreciate this feedback and will expand the Seizure-RQI description in the **main text** of the revision. Currently,the **complete formulation** is provided in Appendix Section 4 (Lines 748–837) (Equations 2–7), while the four components of Seizure-RQI are briefly described in Section 3.2.6 (Lines 256–266) of the main text. Additionally, the full evaluation code will be released publicly via Hugging Face, enabling exact reproductions.
>
> Structural Completeness (S):  Assesses whether the report contains expected 3 sections: onset, propagation, and postictal state.  Symptom Coverage (C): Ratio of correctly extracted features from the generated report over all extracted features from the ground-truth report.  Key Localizing Features (L): Extracts specific laterality values (left, right, or none) for four features from both generated and ground-truth reports. L is the proportion of matched features.  Temporal Fidelity (T): Temporal-relation F1-Score computed between the chronologically ordered feature list extracted from the MLLM reports and the ground-truth feature list.
>
> [Q2,W2]:Thanks for your comment. Please note that we select Qwen2.5-Omni-7B as the base model due to its capability to jointly process visual and auditory modalities. Compared to this baseline, excluding Task 3, the fine-tuned models achieve consistent improvements across all remaining six tasks, with an average relative improvement of 12% (SFT) and 15% (GRPO), as shown in Table 10.
>
> Task 3 catastrophic forgetting due to limited laterality data samples, with details in rebuttal for reviewer rHK6[Q2].
> For Task 4, the main text reports only seizure start/end MAE, but the full evaluation covers 19 events (17 features in Appendix Table 20). Across all 19 events, 18 show reduced MAE after finetuning. Average MAE decreases from 25.50s (baseline) to 23.02s (SFT) and 20.02s (GRPO), representing 9.7% and 21.5% relative improvement. We will incorporate the complete 19-event Task 4 results into the main text tables in the final version.
> For Task 5, we use LCS ratio as the primary metric because it accounts for output length, unlike Edit Distance and Temporal F1 which can be inflated by repetitive outputs. LCS ratio improves from 0.12 to 0.18 (50% gain) after finetuning. The Temporal F1 decrease (0.45→0.32) is an artifact of reduced repetition — the baseline generates substantial meaningless repetitions, and once eliminated, fewer pairwise comparisons lower the F1 numerically despite improved sequence quality.
>
> [W3] Thank you for this comment. First, we compared clinician performance under identical sampled-frame constraints (2 FPS). Clinicians consistently outperformed the best MLLM across Task1,3,7, confirming the model limitations not solely preprocessing-induced information loss.  More details in rebuttal for rHK6 [Q5].
> Second, we conducted an ablation study on sampling rate: increasing from 2 FPS to 4 FPS and 10 FPS yields average Task 1 F1 improvements of 0.06 and 0.08, respectively in Table 11.
> Third, our 2FPS sampling strategy balances MLLM context length limits with the visual clarity, with explanation in the 2nd paragraph of rebuttal for reviewer HwXc [Q3, W3].
>
> [W4] The finetuned models achieve consistent improvements across six tasks excluding Task 3. Our two-stage neuro-symbolic framework achieves F1 = 0.96 on ES vs. NES classification, dramatically outperforming direct MLLM classification (F1 = 0.70), offering a pathway to scalable, resource-efficient screening through mobile-captured video. This has direct clinical significance: over 65million people worldwide have epilepsy, especially those in regions where trained epileptologists are scarce. Seizure-RQI dramatically outperforms standard NLP metrics as a clinically aligned evaluation method, representing the first effective automated assessment tool for seizure semiology reports in this field. Our systematic analysis of MLLM failure modes across the seven-task hierarchy directly informs future model development. Unlike prior discriminative deep learning approaches that produce opaque categorical outputs within limited symptom categories, our MLLM-based pipeline generates interpretable natural-language symptom descriptions, making diagnosis transparent and clinically auditable. Beyond clinical application, seizure semiology — characterized by rapid, concurrent, spatiotemporally evolving involuntary movements — represents a uniquely challenging visual perception task, and seizure diagnosis requires domain-specific clinical reasoning far beyond general-purpose video understanding. S³ enables cross-modality pathological research through temporal alignment with EEG and MRI, accelerating progress toward expert-level multimodal medical intelligence.

---

> > ### Author Rebuttal · Reviewer_uKma · 2026-04-04
> >
> > While the architectural novelty is limited, I recognize the originality of this work as the first effective automated assessment tool for seizure semiology reports in this domain. I will adjust my score accordingly.

---

> > > ### Author Response · Authors · 2026-04-06
> > >
> > > We again thank the reviewer for your original constructive comments that have improved the presentation of our paper. We greatly appreciate the reviewer’s kind recognition of the originality of our work “as the first effective automated assessment tool for seizure semiology reports in this domain.”
> > >
> > > Beyond the first application of MLLMs to quantifying seizure semiology, our paper also is one of the first to quantify both the power and limitations of MLLMs in quantifying pathological motion and motor artifacts—categories critically underrepresented in their training data. The detailed task and prompt designs introduced in our paper can also help in leveraging behavioral video analysis in diagnosing neurological conditions such as Alzheimer's and Parkinson's disease.
> > >
> > > We are genuinely grateful that the reviewer has agreed to increase their overall recommendation score. We note that the OpenReview system currently has not changed it, which may require updating the original review separately. We would greatly appreciate a timely adjustment of the score before the April 7th AOE deadline.

---

### Official Review · Reviewer_rHK6 · 2026-03-13

**Soundness:** 3
**Presentation:** 3
**Significance:** 3
**Originality:** 4
**Overall Recommendation:** 5
**Confidence:** 3

**Summary:**

This paper introduces Seizure-Semiology-Suite (S³), a dataset and benchmark for evaluating Multimodal Large Language Models (MLLMs) on seizure semiology understanding from clinical video. The dataset contains 438 seizure videos from 116 patients, annotated with over 35,000 dense labels spanning 20 ILAE-defined semiological features. The authors propose a seven-task hierarchical benchmark progressing from low-level feature recognition (Task 1) through justification (Task 2), spatial/anatomical analysis (Task 3), temporal boundary detection (Task 4), semiological sequencing (Task 5), narrative report generation (Task 6), to clinical diagnosis (Task 7). A new evaluation metric, Seizure-RQI, is introduced for report quality assessment. Extensive baselines across 11 open-weight MLLMs are reported, revealing systematic weaknesses in laterality reasoning, temporal localization, and clinical reporting. The authors further show that seizure-specific finetuning (SFT and GRPO) and a two-stage neuro-symbolic classification framework improve performance, with the latter achieving an F1 of 0.96 on ES vs. NES classification.

**Compliance With Llm Reviewing Policy:**

Affirmed.

**Final Justification:**

I maintain my score because I believe it's a good reflection of my review, as evidenced by the discussions.

**Key Questions For Authors:**

Q1. Can you provide instance-level inter-rater agreement metrics (e.g., Cohen's kappa per feature) between epileptologists and trained annotators on the shared 151-video subset?

Q2. The finetuned models (seizure_omni_sft, seizure_omni_grpo) show catastrophic degradation on Task 3 (F1 drops to 0.00 on head turning and arm movement lateralization). Can you explain this failure mode?

Q3. For the two-stage classification achieving F1 = 0.96: (a) How many patients are in the test fold? (b) Can you provide 5-fold cross-validated results with standard deviations? (c) What are the Random Forest feature importances

Q4. The Seizure-RQI relies on Qwen3-Plus for structured extraction from generated reports. What is the extraction accuracy of this step? Have you measured how often the LLM extractor introduces errors, and how these propagate into the final RQI score?

Q5. Can you provide a human clinician baseline on at least Tasks 1, 3, and 7, under the same input constraints (e.g., viewing the same sampled frames rather than full video)? This would help disentangle whether poor model performance reflects fundamental model limitations versus information loss from the preprocessing pipeline

**Limitations:**

The authors discuss several limitations in their Impact Statement, including the risk of misrecognition, incorrect temporal localization, and overreliance on model outputs. They appropriately frame the system as decision-support. However, the following limitations are insufficiently discussed:

- The single-institution origin and its implications for generalizability.
- The potential for the benchmark itself to become a misleading proxy if models overfit to the specific annotation style, camera setup, or patient demographics of this EMU.
- The reliance on an external LLM for both annotation cleanup (Section 3.1, stage v) and RQI computation introduces circular dependencies that could inflate or deflate metrics in systematic ways.

**Strengths And Weaknesses:**

Strengths

1.The paper provides clinical data that adresses the fact that automated seizure semiology analysis from video is labor-intensive, subjective, and difficult to scale. The dataset construction is rigorous, with a five-stage annotation workflow involving epileptologist supervision, iterative verification, and multiple quality control measures. The choice of 20 ILAE-defined features as the annotation taxonomy is well-motivated and clinically standardized.

2. The seven-task hierarchy is one of the paper's strongest contributions IMO. It decomposes the complex clinical reasoning process into evaluable subtasks that mirror the cognitive steps a clinician would follow. From perceptual recognition through temporal reasoning to holistic diagnosis, this decomposition enables fine-grained diagnosis of model failure modes rather than a single end-to-end accuracy number, which is far more informative for guiding future research.

3. The Report Quality Index is a compelling contribution as the authors demonstarte that standard NLP metrics (BLEU, ROUGE-L, BERTScore) show negligible correlation with expert judgment (Pearson r ≤ 0.10), while Seizure-RQI achieves r = 0.57 and pairwise accuracy of 0.74 (Figure 5, Table 9). The metric's design, with clinically motivated components (structural completeness, symptom coverage, localizing features, temporal fidelity) and penalty terms for hallucinations, off-topic content, verbosity, and hazardous statements, reflects engagement with domain experts about what matters in seizure reporting.

4. The evaluation of 11 models spanning multiple architecture families, scales, and modality configurations is extensive. The analysis yields several actionable insights: scaling alone does not overcome architectural bottlenecks for this domain (similar F1 across 7B–72B Qwen variants); multimodal fusion provides tangible benefits on audio-relevant features; medical pretraining helps reporting (Lingshu-32B achieves highest RQI) but can hurt language reasoning (its F1 drops with report augmentation). The finding that laterality reasoning remains uniformly poor even with explicit prompt tuning (Task 3, mean F1 < 0.2) is an important negative result for the MLLM community.


Weaknesses

1. Instance-Level Annotation Agreement Is Missing:
The annotation quality validation (Section 3.1, Appendix 1.2) relies entirely on distribution-level agreement metrics (Pearson correlation, cosine similarity, EMD between aggregate feature frequency vectors). While these show that annotators reproduce expert-level feature distributions, they do not demonstrate instance-level agreement. Two annotators could produce identical marginal distributions while disagreeing substantially on individual videos. Standard inter-rater reliability metrics like Cohen's kappa or Fleiss' kappa per feature, or at minimum a confusion matrix between expert and annotator labels on a shared subset are useful here to establish annotation quality at the level relevant for supervised learning.

2. Potential Overfitting and Insufficient Validation:
The paper mentions patient-level splits but does not report the number of patients in the test set, nor does it provide cross-validation results in the main text. With an imbalanced dataset, a 20-dimensional binary feature vector, and a small test set, a Random Forest can easily overfit to distributional artifacts. The paper should report: (a) cross-validated results with variance, (b) the feature importance from the Random Forest to verify clinical plausibility, and  (c) a comparison with simpler classifiers (logistic regression, SVM) to confirm that the improvement comes from the structured feature extraction rather than the classifier choice.

3. Limited Error Analysis and Qualitative Examples. Despite the extensive quantitative evaluation, the paper lacks qualitative error analysis. No failure case visualizations, no confusion matrices for Task 1 features, no examples of generated reports versus ground truth

---

> ### Author Rebuttal · Authors · 2026-03-31
>
> All tables referred in this response can be found at this anonymous link: https://anonymous.4open.science/r/ICML-FF68/
>
> We appreciate the reviewer rHK6 for very thoughtful and technically deep questions.
>
> [Q1,W1]:We compute feature-wise Cohen’s kappa between epileptologists and trained annotators on the independently labeled subset as illustrated in Table 3. These results(average 0.8395 ) prove that the trained annotators reproduce epileptologist labels with high fidelity at the per-feature level, supporting the validity of our annotation pipeline.
>
> **[Q2]**: This failure of VLMs on left-right spatial reasoning is consistent with broader findings in the literature and was included as a demonstration of the challenges MLLMs would face in seizure semiology understanding; Kamath et al. (2023) found that all 18 evaluated VLMs perform poorly on spatial reasoning due to insufficient spatial relationship data in pretraining corpora like LAION-2B. Specific to our case, Task 3  has only 527 training samples (~1/8 of Tasks 1–2) in Table 9, with head turning lateralization at 98 and arm movement lateralization at 83 samples — insufficient for fine-grained spatial reasoning. During SFT, the model collapses all predictions to a single direction ("left"), yielding F1 = 0.00 (positive label is "right"). This is catastrophic forgetting: the base model achieves F1 = 0.20 on head turning before finetuning, but narrow finetuning on limited laterality data erases this capability. To address Task 3 degradation, feasible solutions are augmenting training data via horizontal flipping with label swapping to double the laterality samples, and adopting task-balanced multi-task training to prevent catastrophic forgetting.
>
> [Q3,W2]:  (a) The reported result was evaluated on a test fold of 24 patients, 82 videos.
> (b) As requested we performed 5-fold cross-validation for the Random Forest classifier used in the paper (see Table 4). The fold-wise F1 scores were 0.960, 0.970, 0.947, 0.930, 0.940, corresponding to a mean F1 of 0.9494 and standard deviation of 0.0146. We also evaluated several other classifiers: Random Forest 0.9494 ± 0.0146, SVM 0.9400 ± 0.0139, Logistic Regression 0.8932 ± 0.0129, and KNN 0.8684 ± 0.0169. The similar performance suggests that the primary benefit comes from the extracted structured semiology features, with classifier choice having a smaller effect.
> (c) We additionally show Random Forest feature importances in Table 5. The most important features were head turning (0.194), tonic (0.130), rapid eye blinking (0.100). These are clinically meaningful semiological cues and further support the interpretability of the two-stage framework. We will include these results in the revised manuscript.
>
> [Q4]:  We address this in three different ways.  First, LLMs are being increasingly used to evaluate text tasks, and we added experiments involving two other LLMs to compare RQI scores, as shown in Table1. All LLMs yield minimal variation in Task 2 scores (0.61–0.62) and small differences in Task 6 RQI (38.70–42.26).
> Second,Seizure-RQI validation against human experts(Fig.5 in Main text).  Ranking consistency  was computed using the pairwise accuracy. High correlations/accuracy –hence, agreement between clinicians and the LLM computed RQI scores – indicates that not only the RQI score by its design is a better metric, but that the LLM did an accurate job in computing the scores.
> Third, we will add experiments in the revision to directly quantify extraction accuracy against human annotated ground truth for RQI.
>
> **[Q5]**: We compared clinician and MLLM performance under identical sampled-frame constraints (2 FPS). As shown in Table 6, clinicians consistently outperformed the best MLLM across all Task 1 evaluated features (e.g., tonic 0.87 vs. 0.69, figure-4 0.78 vs. 0.42, face twitching 0.83 vs. 0.44), confirming that the performance gap reflects genuine model limitations, not solely preprocessing-induced information loss. Eye blinking yields lower scores even for clinicians (0.52), indicating frame subsampling does remove some information for transient features, but the remaining clinician–MLLM gap still points to fundamental model weaknesses.
> For Task 3, clinicians achieved F1 = 0.89 vs. 0.38 (MLLM) on head turing direction and 0.54 vs. 0.25 on body-region onset in Table 8; For Task 7, the MLLM (F1 = 0.96) outperformed the clinician (F1 = 0.91) in Table 7,  showing that the large gap is not attributable to preprocessing alone.
> Overall, preprocessing introduces some information loss for subtle perceptual events, but does not fully explain model underperformance — the impact is task-dependent. We will include these clinician baselines in the revised manuscript.
>
> [W3]: We add sample reports  in Table 12, and Task1 confusion matrices will be added in revision.

---

> > ### Author Rebuttal · Reviewer_rHK6 · 2026-04-04
> >
> > Thanks for the authors’ response, it address most of my concerns. I will keep my score unchanged.

---

> > > ### Author Response · Authors · 2026-04-06
> > >
> > > Thank you for your constructive feedback and the time you invested in reviewing our work. We are pleased to hear that our rebuttal has fully resolved your concerns. Your feedback has been invaluable in helping us improve the manuscript, and we will incorporate all the suggested revisions.

---

### Official Review · Reviewer_HwXc · 2026-03-13

**Soundness:** 3
**Presentation:** 3
**Significance:** 3
**Originality:** 3
**Overall Recommendation:** 4
**Confidence:** 3

**Summary:**

The paper introduces Seizure-Semiology-Suite (S³), a clinically grounded dataset and benchmark for seizure semiology understanding from video. On top of this dataset, the paper proposes a seven-task benchmark spanning recognition, justification, spatial/laterality reasoning, temporal boundary detection, symptom progression analysis, report generation, and diagnosis. It also introduces Seizure-RQI, a clinically grounded metric for evaluating generated semiology reports.

**Compliance With Llm Reviewing Policy:**

Affirmed.

**Final Justification:**

I would prefer to keep my rating unchanged.

**Key Questions For Authors:**

1. Are the main tables in the paper reported on the held-out test split or another evaluation subset?
2. How sensitive are Task 2 and Task 6 results to the external LLM-based extraction/judging pipeline?
3. For Task 4, clips are centered around ground-truth feature onset during evaluation/processing. How much does performance drop in a more realistic full-video temporal localization setting?
4. Do the authors have any external-validation evidence, or at least a concrete plan, for generalization beyond this single-center adult EMU dataset?

**Limitations:**

yes

**Strengths And Weaknesses:**

**Strengths**

1. Clinically important problem and strong task design. The paper targets seizure semiology understanding from video, which is clearly relevant to epilepsy diagnosis and is not well covered by existing general video benchmarks. The 7-task hierarchy is well designed and moves from perception to reporting and diagnosis.
2. Clear 7-task benchmark covering recognition, explanation, localization, sequencing, reporting, and diagnosis.
3. Strong experimental coverage: 11 open models, plus fine-tuning and a two-stage classifier.
4. Broad and honest empirical analysis. The authors benchmark 11 open-weight models and clearly identify failure modes rather than overstating performance. Fine-tuning and the two-stage classifier also add practical value.

**Weaknesses:**
1. The best models still perform poorly on laterality/spatial reasoning, temporal localization, and symptom sequencing, which limits near-term clinical usefulness.
2. Some evaluation depends on external LLM judging/extraction. In particular, Task 2 and parts of report evaluation rely on LLM-based extraction/judging, which may introduce evaluator sensitivity.
3. Some settings are easier than full real-world deployment. For example, Task 4 uses event-centric clips centered around ground-truth onset, so real full-video localization may be harder.

---

> ### Author Rebuttal · Authors · 2026-03-31
>
> All tables referred in this response can be found at this anonymous link:https://anonymous.4open.science/r/ICML-FF68/
>
> Thanks for your feedback. We are happy to provide more details beyond the following.
>
> [Q1]: Yes, tables in the main text report results on the held-out test dataset(82 videos). Full-dataset results (432 videos) are provided in Appendix Tables 11–16 for completeness.
>
> [Q2,W2]:This is a valid concern. To reduce the degrees of freedom for hallucination or drift, we used a structured extraction design: The LLM extractor uses carefully designed structured prompts (Appendix Tables 5–7) that constrain outputs to predefined clinical categories.
> We added an ablation experiment in the revision to quantify extraction consistency across different LLM backends. As shown in Table1, replacing the LLM judge with Kimi-K2.5 or GPT-5.4 yields minimal variation in Task 2 scores (0.61–0.62) and small differences in Task 6 RQI (38.70–42.26), demonstrating that our evaluation pipeline is robust.
> Finally, we validate Seizure-RQI against human experts’ scores(Figure5 in Main text). It demonstrates that our evaluation pipeline, including the LLM extraction step, produces scores that meaningfully correlate with clinical judgment.
>
> **[Q3, W3]**:Thank you for raising this point.  We show in the next paragraph that to accommodate the context-length restrictions of current VLMs while preserving enough precision required for seizure semilogy detection tasks, the maximum number of frames we can upload at one time is 60 frames.  We thus conducted an experiment for Task 4 on full videos by uniformly sampling 60 frames. As shown in Table 2, full-video evaluation degrades temporal localization performance, with average MAE increasing by 4.91s and the clinically critical feature clonic showing an MAE increase of 11.39s. Critically, even under the current 60s event-centric setting, the best model (Qwen2.5-VL-32B) achieves an onset MAE of only 8.19 seconds — already well beyond the clinically acceptable precision required for ictal localization. This result demonstrates that temporal grounding is a fundamental limitation of current MLLMs. Full-video evaluation amplifies this deficiency further due to compounding reasons: (1) lower frame rates reduce temporal granularity. Nearly 0.5 FPS(average video duration is 120.35s) makes precise onset detection fundamentally harder; (2) longer video duration expands the maximum possible error.
>
> Returning to the question of the maximum number of frames one can feed at any time, we note that **our sampling strategy was designed by jointly considering MLLM context length limits with the visual clarity required for medical symptom detection**. In Qwen2.5-VL, total video tokens follow: Total Tokens = ⌈Frames/2⌉ × ⌈Width/28⌉ × ⌈Height/28⌉, with a cap of 24,576 tokens. At 720P this allows ~40 frames; at 480P ~118 frames. Since seizure symptoms involve subtle facial and limb changes demanding adequate spatial resolution, we set resolution between 480P and 720P with a maximum of 60 frames (120 for Qwen3). Compared to standard benchmarks that use sparse sampling (Video-MME: 16–128 frames, MVBench: 8–16, MotionBench: ~0.2 FPS), seizure symptom evolution requires denser temporal coverage, so we adopted 2 FPS with 30-second clips (and 5-second overlap to prevent feature truncation at segment boundaries).
>
> [Q4]:We recognize single-center origin as a limitation. Our concrete generalization plan includes: (1) We are pursuing multi-center data collection and IRB approvals from other tertiary EMUs in order to perform multi-center external validation of both the benchmark and the trained models. (2) our public release plan (dataset under DUA, code, models via Hugging Face) is designed to enable community-driven external validation; (3) the benchmark design is institution-agnostic—any center can apply our 7-task evaluation framework to their own data.
>
> [W1] Our two-stage framework already achieves F1=0.96 on ES vs. NES classification, dramatically outperforming direct MLLM classification (F1=0.70). This has immediate clinical value: over 65 million people worldwide have epilepsy, and in resource-limited regions where access to specialized diagnosis is scarce, NES patients are frequently misdiagnosed as ES and subjected to ineffective antiepileptic medications that fail to relieve their symptoms. Our framework offers a scalable, automated screening pathway that could meaningfully benefit these underserved populations. Beyond near-term utility, we view exposing these failure modes as a contribution of S³. By precisely quantifying where MLLMs fall short, our benchmark charts clear directions for future development: stronger visual perception, spatiotemporal localization, and medical reasoning. As the first fine-grained semiology video dataset, S³ enables cross-modality pathological analysis through temporal alignment with EEG and MRI, accelerating progress toward expert-level multimodal medical intelligence.

---

> > ### Author Rebuttal · Reviewer_HwXc · 2026-04-03
> >
> > Thanks for the authors’ response. I will keep my score unchanged.

---

> > > ### Author Response · Authors · 2026-04-06
> > >
> > > Thank you for your thorough review and for taking the time to evaluate our rebuttal. We greatly appreciate your suggestions, which have helped us improve the quality of our paper.

---

### Decision · Program_Chairs · 2026-04-30

**Decision:**

Accept (spotlight)

**Comment:**

The reviewers have achieved agreement on accepting this paper.